# Exploring and Improving Initialization for Deep Graph Neural Networks: A Signal Propagation Perspective

**Senmiao Wang**[*]                                          *senmiaowang1@link.cuhk.edu.cn*
*The Chinese University of Hong Kong, Shenzhen, China*

**Yupeng Chen**[*]                                          *yupengchen1@link.cuhk.edu.cn*
*The Chinese University of Hong Kong, Shenzhen, China*

**Yushun Zhang**                                          *yushunzhang@link.cuhk.edu.cn*
*The Chinese University of Hong Kong, Shenzhen, China*
*Shenzhen Research Institute of Big Data*

**Ruoyu Sun**                                          *sunruoyu@cuhk.edu.cn*
*The Chinese University of Hong Kong, Shenzhen, China*
*Shenzhen International Center for Industrial and Applied Mathematics*
*Shenzhen Research Institute of Big Data*

**Tian Ding**[†]                                          *dingtian@sribd.cn*
*Shenzhen International Center for Industrial and Applied Mathematics*
*Shenzhen Research Institute of Big Data*

**Reviewed on OpenReview:** *https://openreview.net/forum?id=6AjOaNXfRy*

## Abstract

Graph Neural Networks (GNNs) often suffer from performance degradation as the network depth increases. This paper addresses this issue by introducing initialization methods that enhance signal propagation (SP) within GNNs. We propose three key metrics for effective SP in GNNs: forward propagation, backward propagation, and graph embedding variation (GEV). While the first two metrics derive from classical SP theory, the third is specifically designed for GNNs. We theoretically demonstrate that a broad range of commonly used initialization methods for GNNs, which exhibit performance degradation with increasing depth, fail to control these three metrics simultaneously. To deal with this limitation, a direct exploitation of the SP analysis–searching for weight initialization variances that optimize the three metrics–is shown to significantly enhance the SP in deep GCNs. This approach is called ***Signal Propagation on Graph-guided Initialization (SPoGInit)***. Our experiments demonstrate that SPoGInit outperforms commonly used initialization methods on various tasks and architectures. Notably, SPoGInit enables performance improvements as GNNs deepen, which represents a significant advancement in addressing depth-related challenges and highlights the validity and effectiveness of the SP analysis framework.

## 1 Introduction

Increasing depth has been a prominent trend in the development of neural networks. For instance, from AlexNet (Krizhevsky et al., 2012), VGG19 (Simonyan & Zisserman, 2015) to ResNet (He et al., 2016), the depth of the Convolutional Neural Network (CNN) has increased from 8, 19 to 152, and the corresponding test accuracy on ImageNet has increased from 63.3%, 74.4% to 78.57%. Theoretically, the benefit of depth is

---

[*]Equal contribution.
[†]Corresponding author.

often attributed to strong representation power. Research shows that a shallow network would require an exponential increase in width to match the representational power of a deep network (Telgarsky, 2015; Eldan & Shamir, 2016; Liang & Srikant, 2017).

In graph-related tasks like node classification, graph classification, or link prediction, graph neural networks (GNN) (Wu et al., 2022a; 2020) are also expected to benefit from increased depth. A core concept in GNNs is the message-passing mechanism, where each node aggregates information from its neighboring nodes.[1] Deeper GNNs have larger receptive fields, enabling nodes to gather information from broader local sub-graphs. This is especially beneficial for capturing long-range relationships in complex graph-related tasks. For instance, GNNs have shown great potential in solving optimization problems (Gasse et al., 2019; Nair et al., 2020; Han et al., 2022; Li et al., 2024a). Theoretical studies demonstrate that GNNs possess universal approximation power for solving various optimization problems, including linear programming (Chen et al., 2023) and quadratic programming (Chen et al., 2024), but the network depth need to scale with the problem dimensions (Qian et al., 2024; Li et al., 2024b). Thus, increasing the depth of GNNs is expected to improve their performance in addressing large-scale optimization problems.

However, GNNs often experience performance degradation in practice when their depth increases (Li et al., 2018; Wu et al., 2020; Zhou et al., 2020a). Consequently, most GNNs remain shallow, typically comprising only 2 to 10 layers (Kipf & Welling, 2017; Veličković et al., 2017; Alon & Yahav, 2021). In recent years, over-smoothing has been identified as a primary cause of this issue (Li et al., 2018; Oono & Suzuki, 2019; Cai & Wang, 2020). Over-smoothing refers to a specific issue in GNNs where node embeddings become similar as depth grows, reducing the distinguishability between nodes and impairing task performance. This poses a major obstacle to the development of deeper GNNs and may hinder progress in complex graph-related problems, particularly those involving long-range relationships.

In this paper, we show that over-smoothing can be understood as part of a broader issue known as signal propagation (SP). SP refers to how the input data is transformed as it passes through the layers of a neural network. For traditional neural networks like CNNs, SP analysis plays a critical role in developing initialization strategies to maintain stable signal propagation and prevent gradients from exploding or vanishing during training (Poole et al., 2016; Schoenholz et al., 2017; Pennington et al., 2017; 2018; Hanin, 2018). This work extends the SP analysis framework to GNNs. We introduce a new graph-specific SP metric, graph embedding variation (GEV), which closely relates to over-smoothing. Alongside the standard forward and backward SP (FSP and BSP) metrics, we use GEV to evaluate signal propagation in GNNs.

Building on this framework, we focus on the family of graph convolutional networks (GCNs), one of the most widely-used GNN architectures, to demonstrate the interplay between initialization strategies and signal propagation. While traditional initializations such as Kaiming (He et al., 2015), Xavier (Glorot & Bengio, 2010), and LeCun (Bottou, 1988; LeCun et al., 2002) initializations are commonly used in standard GCN implementations (e.g., the PyTorch Geometric library PyG (Fey & Lenssen, 2019)), we theoretically prove that these initialization methods fail to stabilize all three SP metrics concurrently, leaving deep GCNs vulnerable to performance degradation.

Motivated by the SP framework, we propose a new initialization method, termed ***Signal Propagation on Graph-guided Initialization (SPoGInit)***, for deep GCNs. SPoGInit consolidates the three SP metrics (FSP, BSP, and GEV) into a unified optimization objective, where minimizing the objective leads to more stable SP. Using an iterative algorithm, SPoGInit adjusts the weight initialization variances across layers to minimize this objective. Thus, SPoGInit can simultaneously stabilize all three SP metrics. Furthermore, the design of SPoGInit is independent of specific network architectures, and hence it can adapt effectively across diverse GCN models. The effectiveness of SPoGInit verifies that stabilizing the proposed SP metric can improve the performance of deep GCNs and mitigate the performance degradation problem.

Our contributions are as follows:

- **Theoretical Analysis:** We prove that traditional initialization methods for vanilla GCNs and Residual GCNs (ResGCNs) fail to simultaneously control all three signal propagation metrics. This failure leads to

---

[1]In this paper, we refer specifically to GNNs as *message-passing GNNs*. We note that other network structures not based on message-passing, such as Transformers (Wu et al., 2022b; Kong et al., 2023), are also used in graph-related tasks.

the explosion or vanishing of one or more metrics, ultimately causing performance degradation as network depth increases. We also present experimental evidence to validate our theoretical findings.

- **Empirical Exploration.** Building on the proposed SP framework, we introduce a new initialization design method, ***Signal Propagation on Graph-guided Initialization (SPoGInit)***. SPoGInit employs an optimization algorithm to determine initial weight variances that effectively stabilize all three signal propagation metrics. Experimental results demonstrate that SPoGInit significantly improves signal propagation across various architectures, enhancing the performance of deep GCNs, particularly for graph-based tasks involving long-range relationships. The effectiveness of SPoGInit demonstrates that improving the proposed SP metrics is instrumental in boosting deep GCNs' performance.

## 2 Preliminaries and Background

For any integer $N \in \mathbb{N}$, we define $[N] := \{1, 2, \ldots, N\}$. For brevity, we use $\theta$ to denote the collection of trainable parameters in a GNN model. For additional useful notation, see Appendix A.

### 2.1 Graph convolutional networks

**Featured graph.** Let $\mathcal{G} = (\mathcal{V}, \mathcal{E})$ be an undirected graph, where $\mathcal{V}$ is the set of nodes with $|\mathcal{V}| = n$, and $\mathcal{E}$ is the collection of edges. Assume that each node is associated with a $d_0$-dimensional feature and a label belonging to the set $[C]$, where $C \geq 2$ denotes the number of possible labels. Let $x_i \in \mathbb{R}^{d_0 \times 1}$ and $y_i \in [C]$ denote the feature and the label of node $i$, respectively. Define the node feature matrix as $X = (x_1, x_2, \ldots, x_n)^\top \in \mathbb{R}^{n \times d_0}$. Let $A = (\mathbb{1}_{\{(i,j) \in \mathcal{E}\}})_{i,j \in [n]} \in \mathbb{R}^{n \times n}$ represent the adjacency matrix and $D = \mathrm{diag}(A\mathbf{1}_n) \in \mathbb{R}^{n \times n}$ represent the degree matrix. Further, $\tilde{A} = A + I$ and $\tilde{D} = D + I$ denote the adjacency matrix and the degree matrix of graph $\mathcal{G}$ with self-loop added to each node. Finally, the normalized adjacency matrix is given by $\hat{A} = \tilde{D}^{-\frac{1}{2}} \tilde{A} \tilde{D}^{-\frac{1}{2}}$.

**Vanilla GCN.** Vanilla GCN (Kipf & Welling, 2017) stacks neighborhood aggregations and feature transformations alternately. Specifically, let $H^{(l)}, X^{(l)} \in \mathbb{R}^{n \times d_l}$ denote the pre-activation and the post-activation embedding matrix at the $l$-th layer of the vanilla GCN, respectively. They are defined recursively by

$$H^{(l)} := \hat{A} X^{(l-1)} W^{(l)} + \mathbf{1}_n \cdot b^{(l)}, \quad X^{(l)} := \sigma(H^{(l)}),$$

where $W^{(l)} \in \mathbb{R}^{d_{l-1} \times d_l}$ and $b^{(l)} \in \mathbb{R}^{1 \times d_l}$ are the weight and the bias term at the $l$-th layer, respectively. The input to the first layer is given by $X^{(0)} = X$, and the output matrix of an $L$-layer vanilla GCN is $H^{(L)} \in \mathbb{R}^{n \times C}$, which is then fed into a softmax layer to obtain the predicted labels.

**ResGCN and gatResGCN.** Inspired by He et al. (2016), ResGCN (Kipf & Welling, 2017) combines residual connections with vanilla GCN. An $L$-layer ResGCN adds skip connections to the post-activation embeddings, i.e.,

$$H^{(l)} := \hat{A} X^{(l-1)} W^{(l)} + \mathbf{1}_n \cdot b^{(l)}, \quad X^{(l)} := \alpha \sigma(H^{(l)}) + \beta X^{(l-1)}, \quad \forall l \in [L],$$

where $W^{(l)} \in \mathbb{R}^{d \times d}$ and $b^{(l)} \in \mathbb{R}^{1 \times d}$ are the weight and the bias term at the $l$-th layer, respectively, while $\alpha, \beta \in \mathbb{R}$ are predetermined hyper-parameters.[2] Note that the above formulation requires the hidden dimensions of ResGCN to be equal across all layers. The input of the first layer is given by $X^{(0)} = X W^{(0)}$, and the output of the network is given by $X^{\mathrm{out}} = X^{(L)} W^{(L+1)}$, where $W^{(0)} \in \mathbb{R}^{d_0 \times d}$ and $W^{(L+1)} \in \mathbb{R}^{d \times C}$ are trainable linear transformations to ensure dimension compatability. The output $X^{\mathrm{out}} \in \mathbb{R}^{N \times C}$ is then fed into a softmax layer to obtain the predicted labels. The architecture of a gating ResGCN (gatResGCN) is identical to that of ResGCN, with the exception that the fixed hyper-parameters $\alpha, \beta$ replaced by trainable gating parameters $\alpha^{(l)}, \beta^{(l)}$ for each layer $l \in [L]$.

---

[2]The original version of ResGCN (Kipf & Welling, 2017) focuses on the special case $(\alpha, \beta) = (1, 1)$.

## 2.2 Initialization

We consider the following class of initialization methods. At initialization, all $W_{k'k}^{(l)}$ are i.i.d. and satisfy $\mathbb{E}[W_{k'k}^{(l)}] = 0$, $\text{Var}[W_{k'k}^{(l)}] = \sigma_w^2/d_{l-1}$; all $b_k^{(l)}$ are initialized to be 0 for any $k' \in [d_{l-1}], k \in [d_l], l \in [L]$.

Two widely used random initialization methods, LeCun initialization (Bottou, 1988; LeCun et al., 2002) and Kaiming initialization (He et al., 2015) fit into this framework with $\sigma_w^2 = 1$ and $\sigma_w^2 = 2$ respectively.

- LeCun: $\mathbb{E}[W_{k'k}^{(l)}] = 0$ and $\text{Var}[W_{k'k}^{(l)}] = 1/d_{l-1}$ for any $k' \in [d_{l-1}], k \in [d_l], l \in [L]$.

- Kaiming (usually for ReLU): $\mathbb{E}[W_{k'k}^{(l)}] = 0$ and $\text{Var}[W_{k'k}^{(l)}] = 2/d_{l-1}$ for any $k' \in [d_{l-1}], k \in [d_l], l \in [L]$.

In GCN models, uniform weight distribution with variance $\sigma_w^2 = 1/3$ is also widely used, e.g., in PairNorm (Zhao & Akoglu, 2020), DropEdge (Rong et al., 2020), DropNode (Huang et al., 2020), SkipNode (Lu et al., 2021), GCNII (Chen et al., 2020b). We simply refer to this initialization as "Conventional initialization" in the rest of this paper. Xavier initialization (Glorot & Bengio, 2010) has weight variance $2/(d_{l-1} + d_l) = 1/d$ when hidden layers have the same width $d$.

# 3 Theoretical Analysis of GCN Initializations

In this section, we evaluate the quality of GCN initializations from three aspects based on the signal propagation (SP) theory as follows.

**Forward signal propagation (FSP)** is responsible to extract abstract and higher-level representations from the input data as the information flows through the network. We propose the *FSP metric* $\mathbf{M}_{\text{FSP}}^{(L)}(\sigma_w^2)$, which is the expected output-input norm ratio $\mathbb{E}_\theta[\|H^{(L)}(\theta)\|_F^2/\|X\|_F^2]$. A proper initialization method should prevent $\mathbf{M}_{\text{FSP}}^{(L)}(\sigma_w^2)$ from either vanishing or exploding as $L \to \infty$.

**Backward signal propagation (BSP)** is responsible for updating the weights by utilizing gradients computed via back-propagation. In vanilla GCN, the gradient of $W^{(l)}$ at the $l$-th layer can be decomposed as $\partial\ell/\partial W^{(l)} = \sigma(H^{(l-1)})^T \cdot \hat{A} \cdot [\partial\ell/\partial H^{(l)}]$ where $\ell$ is the training loss. A stable magnitude of $\partial\ell/\partial H^{(l)}$ with respect to the layer $l$ suggests that the gradient is less susceptible to vanishing or exploding. We take $\mathbb{E}_\theta[\|\partial\ell/\partial W^{(1)}\|_F^2]$ at initialization as the *BSP metric* $\mathbf{M}_{\text{BSP}}^{(L)}(\sigma_w^2)$. A proper initialization method should prevent $\mathbf{M}_{\text{BSP}}^{(L)}(\sigma_w^2)$ from vanishing or exploding as $L \to \infty$.

**Graph embedding variation (GEV) propagation** is responsible for tackling the over-smoothing issue, a GCN-specific problem. A number of existing works (Cai & Wang, 2020; Zhou et al., 2021a) measure over-smoothing severity by Dirichlet energy $\text{Dir}(H^{(L)}) = \sum_{(i,j)\in\mathcal{E}} \|h_i/\sqrt{1+d_i} - h_j/\sqrt{1+d_j}\|^2$, where $h_i$ is the output embedding of node $i$. Dirichlet energy $\text{Dir}(H^{(L)})$ reveals the embedding variation with the weighted node pair distance, and a smaller value of $\text{Dir}(H^{(L)})$ is highly related to the over-smoothing. To eliminate the influence of the embedding norm, we propose the *GEV metric* $\mathbf{M}_{\text{GEV}}^{(L)}(\sigma_w^2)$, which is the expected of normalized Dirichlet energy $\mathbb{E}_\theta[\text{Dir}(H^{(L)})/\|H^{(L)}\|_F^2]$ at initialization. A proper initialization method should prevent $\mathbf{M}_{\text{GEV}}^{(L)}(\sigma_w^2)$ from vanishing as $L \to \infty$.

## 3.1 Theoretical results for vanilla GCN

We first theoretically evaluate the signal propagation (SP) quality at initialization in vanilla GCN. Due to the nonlinearity and high dimensionality of neural networks, the SP analysis is challenging. In order to simplify it, we study the infinite-width limit of vanilla GCN using mean field theory (Poole et al., 2016; Schoenholz et al., 2017). Different from traditional NNs, GNN blocks involve interactions across nodes, so we have to consider the SP of $n$ nodes as an integrated whole, rather than that of only one data sample in NNs. Under this approximation, all the channels $\{H_{:,k}^{(l)}\}_{k=1}^{d_l}$ of each embedding at the $l$-th layer are i.i.d., following Gaussian distribution $N(\mathbf{0}_n, \Sigma^{(l)})$. The $n \times n$ covariance matrix $\Sigma^{(l)}$ recursively satisfies

$$\Sigma^{(l)} = \sigma_w^2 \hat{A} G(\Sigma^{(l-1)}) \hat{A}, \quad \Sigma^{(1)} = \sigma_w^2 \hat{A} X X^T \hat{A}/d_0,$$

where $G(\Sigma^{(l)}) = \mathbb{E}_{h \sim N(\mathbf{0}_n, \Sigma)}[\sigma(h)\sigma(h)^T] \in \mathbb{R}^{n \times n}$ (see Appendix C.1 for the details). This theoretical framework is referred to as the neural network Gaussian process (NNGP) correspondence. Under the NNGP correspondence, the forward signal propagation (FSP) metric can be approximated by

$$\mathbf{M}_{\text{FSP}}^{(L)}(\sigma_w^2) \approx \mathbb{E}_{H^{(L)} \sim N(\mathbf{0}_n, \Sigma^{(L)})}\left[\|H^{(L)}\|_{\text{F}}^2 / \|X\|_{\text{F}}^2\right]$$

and the graph embedding variation (GEV) metric can be approximated by

$$\mathbf{M}_{\text{GEV}}^{(L)}(\sigma_w^2) \approx \mathbb{E}_{H^{(L)} \sim N(\mathbf{0}_n, \Sigma^{(L)})}\left[\text{Dir}(H^{(L)}) / \|H^{(L)}\|_{\text{F}}^2\right],$$

where $H^{(L)} \sim N(\mathbf{0}_n, \Sigma^{(L)})$ means all columns (channels) of $H^{(L)} \in \mathbb{R}^{n \times C}$ are i.i.d. $N(\mathbf{0}_n, \Sigma^{(L)})$.

Now we analyze the SP of GCN under various activation functions. We start with ReLU since it is the most commonly used activation in popular GCN models (e.g., Zhao & Akoglu (2020); Rong et al. (2020); Huang et al. (2020); Lu et al. (2021); Chen et al. (2020b)). The following theorem states that under ReLU activation, if the initial weight variance $\sigma_w^2 \leq 2$, which covers Conventional, Kaiming, and LeCun initialization, deep vanilla GCNs suffer from poor FSP and GEV.

**Theorem 3.1.** *Under the NNGP correspondence approximation, when the activation function $\sigma$ is ReLU, we have*

1. *If $\sigma_w^2 = 2$, either the limit graph embedding variation (GEV) metric $\lim_{L \to \infty} \mathbf{M}_{GEV}^{(L)}(\sigma_w^2) = 0$ or the limit forward signal propagation (FSP) metric $\lim_{L \to \infty} \mathbf{M}_{FSP}^{(L)}(\sigma_w^2) = 0$;*

2. *When $\sigma_w^2 < 2$, the forward signal propagation (FSP) metric $\mathbf{M}_{FSP}^{(L)}(\sigma_w^2) \leq \frac{2C}{d_0} \cdot (\sigma_w^2/2)^L$ for any $L \geq 1$.*

Part 1 of Theorem 3.1 shows that under Kaiming initialization in ReLU-activated vanilla GCN, either $\mathbf{M}_{\text{FSP}}^{(L)}$ or $\mathbf{M}_{\text{GEV}}^{(L)}$ vanishes as $L \to \infty$. Part 2 of Theorem 3.1 characterizes the shrinkage of $\mathbf{M}_{\text{FSP}}^{(L)}$ when $\sigma_w^2$ is even less than that of Kaiming initialization. The proof of Theorem 3.1 is provided in Appendix C.3.

**Theorem 3.2.** *Under the NNGP correspondence approximation, when the activation is ReLU, the graph embedding variation (GEV) metric $\mathbf{M}_{GEV}^{(L)}$ is independent of $\sigma_w^2$.*

Theorem 3.2 states that it is impossible to improve the GEV metric, $\mathbf{M}_{\text{GEV}}^{(L)}(\sigma_w^2)$, by simply refining $\sigma_w^2$ for ReLU-activated vanilla GCN. In other words, the over-smoothing issue cannot be resolved by adjusting weight variance $\sigma_w^2$ in ReLU-activated vanilla GCN. The proof of Theorem 3.2 is provided in Appendix C.4.

We now provide numerical evidence for Theorem 3.1 and 3.2. The purple lines in Figure 1(a)-1(c) illustrate the shrinkage of the three SP metrics under Conventional initialization as the network depth $L$ increases. Figure 1(a) when $\sigma_w^2$ presents the vanishing pattern of $\mathbf{M}_{\text{FSP}}^{(L)}(\sigma_w^2)$ is no greater than that of Kaiming initialization, which validates Theorem 3.1. Figure 1(b) shows that $\mathbf{M}_{\text{BSP}}^{(L)}(\sigma_w^2)$ transits from vanishing to stable, and then to exploding as $\sigma_w^2$ increases. Figure 1(c) shows that $\mathbf{M}_{\text{GEV}}^{(L)}(\sigma_w^2)$ cannot be improved via merely changing $\sigma_w^2$, which validate Theorem 3.2.[3]

Different from ReLU-activated GCNs, Figure 1(f) shows that GEV metric transits from vanishing to stable for tanh-activated models as $\sigma_w^2$ increases. With proper $\sigma_w^2$, stable propagation for all three types of signals can be achieved; see the orange lines in Figure 1(d)-1(f). A theoretical result of the FSP for tanh-activated vanilla GCNs is provided in Appendix C.5.

### 3.2 Theoretical results for ResGCN

Similarly to vanilla GCN, performance degradation has also been reported in deeper ResGCN (Huang et al., 2020; Rusch et al., 2023a). In this subsection, we focus on the SP in ResGCN.

---

[3]In all the figures illustrating SP metrics, disappearing nodes and vertical lines are caused by surpassing the machine precision. Specifically, the vanishing FSP result in vertical lines in the plots of the GEV metric, while the exploding FSP leads to node disappearance in the plots of the GEV metric.

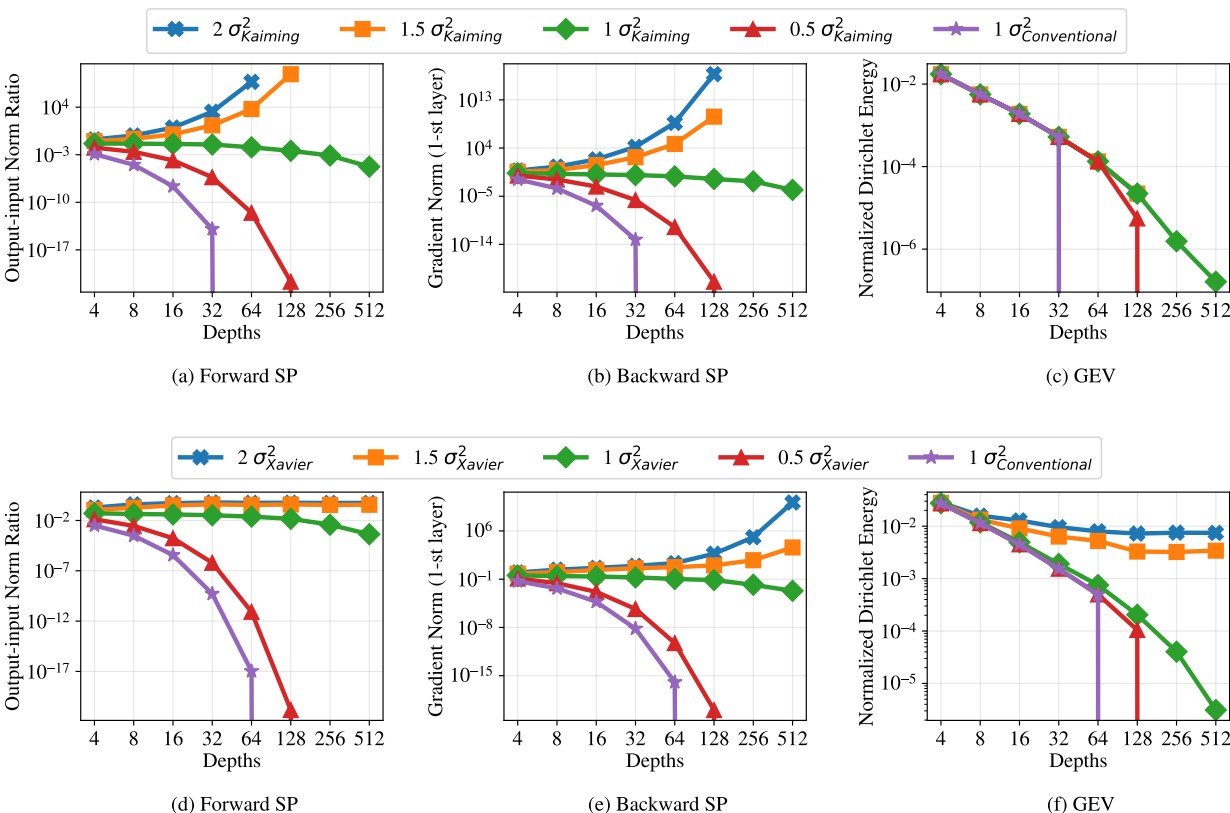

Figure 1: Plots of (a,d) forward metrics, (b,e) backward metrics, and (c,f) graph embedding variation metrics of deep vanilla GCNs with different initialization variances and activations on Cora. (Sub-figures (a)-(c) are for ReLU activation, while sub-figures (d)-(f) are for tanh activation.) We average the results over 20 runs. We see that the choice of initialization variance plays a crucial role in forward and backward propagation. The graph embedding variation propagation can be made stable with proper initialization variance for tanh activation, but not for ReLU activation.

For simplicity, we study linear ResGCN with identity activation in the theoretical analysis. Such a simplification is very common in NN theory (Saxe et al., 2014; Xu et al., 2021). Similar to vanilla GCN, all the channels of $H^{(L)}$ are i.i.d. $N(\mathbf{0}_n, \tilde{\Sigma}^{(L)})$ under the infinite-width limit (a.k.a. NNGP correspondence). The $n \times n$ covariance matrix $\tilde{\Sigma}^{(l)}$ recursively satisfies

$$\tilde{\Sigma}^{(l)} = \sigma_w^2 \hat{A} \tilde{\Sigma}^{(l-1)} \hat{A} + \tilde{\Sigma}^{(l-1)}, \quad \tilde{\Sigma}^{(1)} = \sigma_w^4 \hat{A} X X^T \hat{A}/d_0, \tag{1}$$

See Appendix D.1 for the details.

The following theorem implies that linear ResGCN may suffer from forward signal explosion and over-smoothing under the NNGP approximation at initialization.

**Theorem 3.3.** *Suppose that there exists an eigenvector $u$ of $\hat{A}$ corresponding to the eigenvalue 1, such that the input feature $X \in \mathbb{R}^{n \times d_0}$ satisfies $X^T u \neq \mathbf{0}_{d_0 \times 1}$. Under the NNGP correspondence approximation for linear ResGCN, if $\alpha^2 \sigma_w^2 + \beta^2 > 1$ and $\alpha \neq 0$, then we have*

$$\lim_{L \to \infty} \mathbf{M}_{FSP}^{(L)}(\sigma_w^2) = \infty \quad and \quad \lim_{L \to \infty} \mathbf{M}_{GEV}^{(L)}(\sigma_w^2) = 0.$$

Since $(\alpha, \beta) = (1, 1)$ for the original ResGCN (Kipf & Welling, 2017), $\alpha^2 \sigma_w^2 + \beta^2 > 1$ and $\alpha \neq 0$ always hold for any *nonzero* initialization variance, which indicates exploding $\mathbf{M}_{FSP}^{(L)}(\sigma_w^2)$ and shrinking $\mathbf{M}_{BSP}^{(L)}(\sigma_w^2)$.

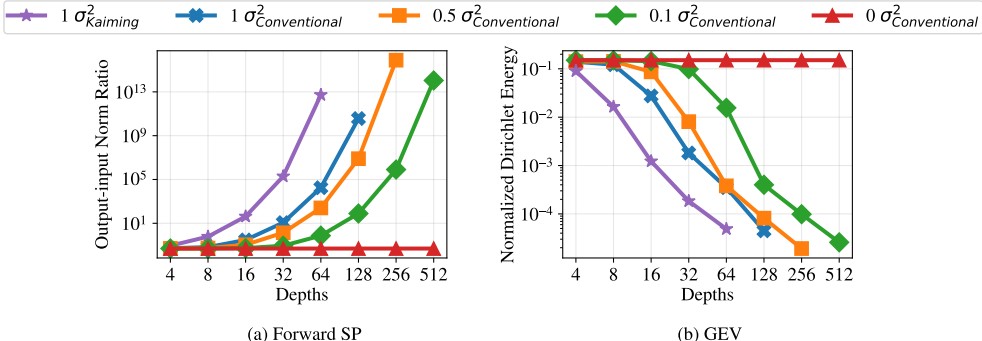

Figure 2: (a) The forward metrics and (b) the graph embedding variation metrics of ReLU-activated deep ResGCN on Cora. We average the results over 20 runs. ResGCNs with non-zero initialization variances always suffer from exploding forward propagation and over-smoothing.

Numerical experiments demonstrate that the consequences of Theorem 3.3 can be observed on ResGCNs with non-linear activations. In Figure 2, we plot the FSP and the GEV of ReLU-activated ResGCN with different initialization variances. We see that the widely used Conventional and Kaiming initialization schemes (Huang et al., 2020; Kipf & Welling, 2017) (and essentially any non-zero initialization variance) lead to exploding forward propagation and over-smoothing.

In summary, the discussions in Section 3.1 and 3.2 provide a theoretical guarantee that the traditional initialization schemes utilized in both vanilla GCN and ResGCN fail to achieve proper SP. To address this challenge, we will introduce new initialization schemes in the subsequent section.

# 4 SPoGInit: Initialization guided by signal propagation on graph

In this section, we propose a new initialization design method, termed **Signal Propagation on Graph-guided Initialization (SPoGInit)**, by enhancing the three types of signal propagation (SP) of GCNs. Through this method, we aim to demonstrate that stabilizing SP can lead to improved performance in deep GCNs and effectively mitigate the performance degradation problem. SPoGInit determines layer-wise initialization variances by solving an optimization problem tailored to the SP of GCNs. To be more specific, given a GCN with $L$ layers, we denote the variance of the $l$-th layer by $\sigma_{w,l}^2$. SPoGInit solves the following optimization problem:

$$\underset{\{\sigma_{w,l}\}_{l=1}^L}{\text{minimize}} \quad w_1 \mathbf{V}_{\text{FSP}} + w_2 \mathbf{V}_{\text{BSP}} - w_3 \mathbf{M}_{\text{GEV}}^{(L)}, \tag{2}$$

where $\mathbf{V}_{\text{FSP}}$ and $\mathbf{V}_{\text{BSP}}$ respectively measure the stability for the FSP and BSP metrics across varying depths.

For vanilla GCNs, the computational graph can often be abstracted as a simple path, suggesting that the stability of SP might be inferred by comparing the SP metrics between very shallow and very deep blocks. Accordingly, we define $\mathbf{V}_{\text{FSP}} := (\mathbf{M}_{\text{FSP}}^{(1)}/\mathbf{M}_{\text{FSP}}^{(L-1)} - 1)^2$ to encourage stable FSP across hidden layers. Similarly, $\mathbf{V}_{\text{BSP}}$ is defined as $(\mathbf{M}_{\text{BSP}}^{(2)}/\mathbf{M}_{\text{BSP}}^{(L-1)} - 1)^2$ to encourage BSP, with the superscript numbers in parentheses indicating the layer indices relevant to the gradient norm. We use $\mathbf{M}_{\text{BSP}}^{(2)}$ rather than $\mathbf{M}_{\text{BSP}}^{(1)}$ to compute $\mathbf{V}_{\text{BSP}}$ to ensure consistent dimensionality across different layers, since the weight parameters of the first layer differ in size from those of the subsequent layers.

For GCNs with skip connections, the computational graph becomes more complex. It is challenging to directly assess the stability of signal propagation by merely examining the SP metrics in very shallow and very deep blocks. Instead, we replace the denominator in vanilla GCN's $\mathbf{V}_{\text{FSP}}$ formula with the smallest FSP metric value, and the numerator with the largest FSP metric value across all the layers. Mathematically, we have

$$\mathbf{V}_{\text{FSP}} = \left( \max_{1 \le l < L} \mathbf{M}_{\text{FSP}}^{(l)} / \min_{1 \le l < L} \mathbf{M}_{\text{FSP}}^{(l)} - 1 \right)^2.$$

Similarly, we introduce the modification on $\mathbf{V}_{\mathrm{BSP}}$ for GCNs with skip connections as follows:

$$\mathbf{V}_{\mathrm{BSP}} = \left( \max_{1 < l < L} \mathbf{M}_{\mathrm{BSP}}^{(l)} / \min_{1 < l < L} \mathbf{M}_{\mathrm{BSP}}^{(l)} - 1 \right)^2 .$$

Besides, in (2), $w_1, w_2, w_3 > 0$ are pre-defined for balancing these three SP metrics. During the implementation of SPoGInit, we adjust the weight initialization variances across layers by gradient descent algorithm. More details about SPoGInit are in Appendix E.

## 5 Numerical Experiments

In this section, we examine the proposed SPoGInit initialization through a series of empirical experiments on various GCN architectures and benchmarks. In Section 5.1, we briefly introduce the experimental settings. In Section 5.2, we demonstrate how SPoGInit improves signal propagation (SP) in different GCN architectures. Finally, in Section 5.3, we showcase the performance of deep GCN models equipped with SPoGInit on mainstream datasets and graph-based tasks involving long-range relationships.

### 5.1 Experiments setting

**Datasets.** We focus on four mainstream datasets and two graph-based tasks involving long-range relationships.

The mainstream datasets include Cora, PubMed (Sen et al., 2008; Yang et al., 2016), OGBN-Arxiv (Hu et al., 2020), and Arxiv-year (Lim et al., 2021). For these mainstream datasets, we use their default training/validation/test splits. Statistics of these datasets are summarized in Table 1.

As for the graph-based tasks involving long-range relationships, we consider 1) the semi-supervised node classification task under missing feature settings, and 2) solving mixed integer linear programming (MILP) problems using GCN-based methods. Further details on these tasks will be provided in Section 5.3.

Table 1: Statistics of the mainstream datasets used in the experiments.

| Dataset | Nodes | Features | Edges | Class | Homophily | Training/Validation/Test |
|---|---|---|---|---|---|---|
| Cora | 2,708 | 1,433 | 10,556 | 7 | 0.81 | 5.2%/18.5%/36.9% |
| PubMed | 19,717 | 500 | 88,648 | 3 | 0.80 | 0.3%/2.5%/5.1% |
| OGBN-Arxiv | 169,343 | 128 | 1,166,243 | 40 | 0.66 | 53.7%/17.6%/28.7% |
| Arxiv-year | 169,343 | 128 | 1,166,243 | 5 | 0.22 | 50%/25%/25% |

**Architectures and Baselines.** For the GCN architectures, we consider the vanilla GCN, and the GCN models with skip-connections: ResGCN (Kipf & Welling, 2017) and the popular MixHop (Abu-El-Haija et al., 2019). Additionally, we examine ResGCN with trainable gating parameters, referred to as gatResGCN. Regarding initialization baselines, we consider standard initialization methods in DNNs and GNNs, including Xavier and Conventional initialization. Besides, we also include VirgoFor and VirgoBack, which are the initialization techniques tailored for GCNs (Li et al., 2023), as part of our baselines. We note that since every layer of MixHop mixes the powers (with different orders) of the adjacency matrix during its information aggregation, VirgoFor and VirgoBack are not directly applicable to MixHop. Thus, our baselines for MixHop only include Conventional and Xavier initializations.

**Implementation.** We conduct all experiments using PyTorch. To prevent out-of-memory issues with deeper models and ensure fair comparisons, we fix the width of all models at 64. In our experiments, we use the tanh activation function for vanilla GCNs, as Theorem 3.2 shows that the graph variation embedding of vanilla GCNs with ReLU activation does not benefit from further optimization. For other GCN architectures, we use the ReLU activation function. All results are averaged over at least three runs. More details of hyperparameters are provided in Appendix G.1.

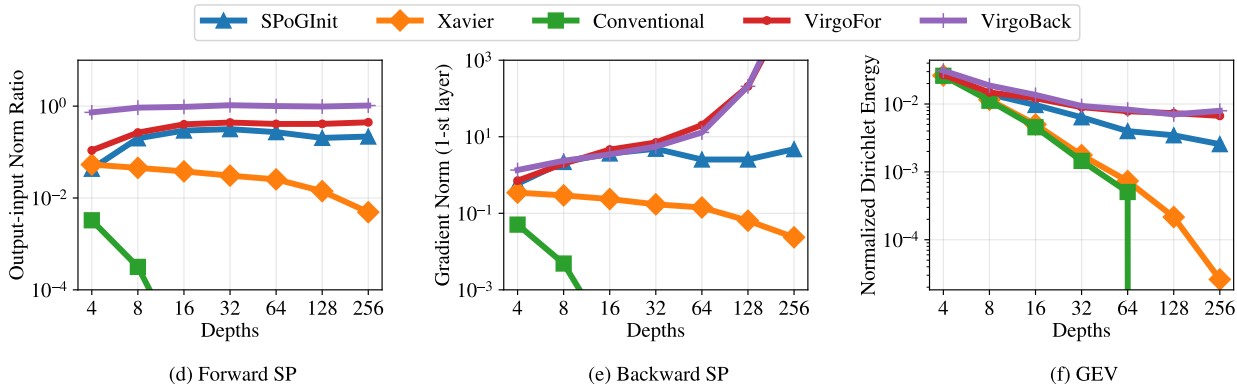

Figure 3: Plots of (a) forward metrics, (b) backward metrics, and (c) graph embedding variation metrics of deep vanilla GCNs with different depths on Cora. SPoGInit is highly effective in stabilizing all three SP metrics.

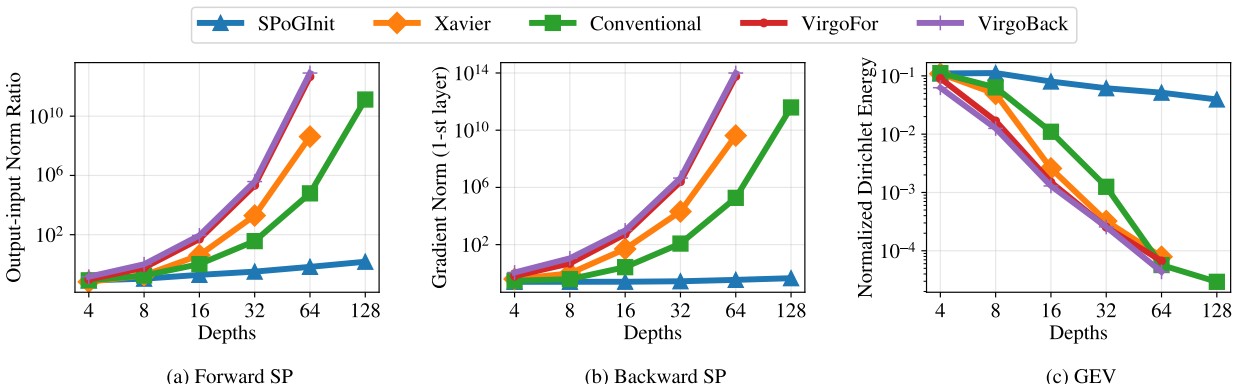

Figure 4: Plots of (a) forward metrics, (b) backward metrics, and (c) graph embedding variation metrics of deep ResGCNs with different depths on Cora. SPoGInit is highly effective in stabilizing all three SP metrics.

## 5.2 Experiments on mainstream datasets

In this section, we illustrate how our proposed SPoGInit improves the performance of deep GCN models on mainstream datasets. Specifically, we address the following questions:

**Q1: Can SPoGInit improve the signal propagation (SP) of GCN models?**

**Q2: Can SPoGInit alleviate the performance degradation in deep GCNs?**

**Can SPoGInit improve the SP of GNN models (Q1)?** On the Cora dataset, we examine the SP across various models, including GCN, ResGCN, and MixHop, under different initializations and depths.

Figures 3, 4, and 5 respectively present the changes in three SP metrics as the depths of GCNs, ResGCN, and MixHop models increase. The results in Figure 3 indicate that on vanilla GCN, both Conventional and Xavier initializations cause diminishing forward SP, backward SP, and graph embedding variations (GEV), which lead to gradient vanishing and over-smoothing issues. The VirgoFor and VirgoBack initialization methods can stabilize the forward SP and maintain the diversity of graph features, but they result in exploding backward SP, leading to gradient explosion. In contrast, SPoGInit is highly effective in stabilizing all three SP metrics.

Figure 4 shows that as the depth increases, ResGCNs with all baseline initializations suffer from exploding forward-backward SP, and diminishing GEV, which lead to gradient explosion and over-smoothing problems. In contrast, the SPoGInit method effectively stabilizes all three SP metrics in deep ResGCNs.

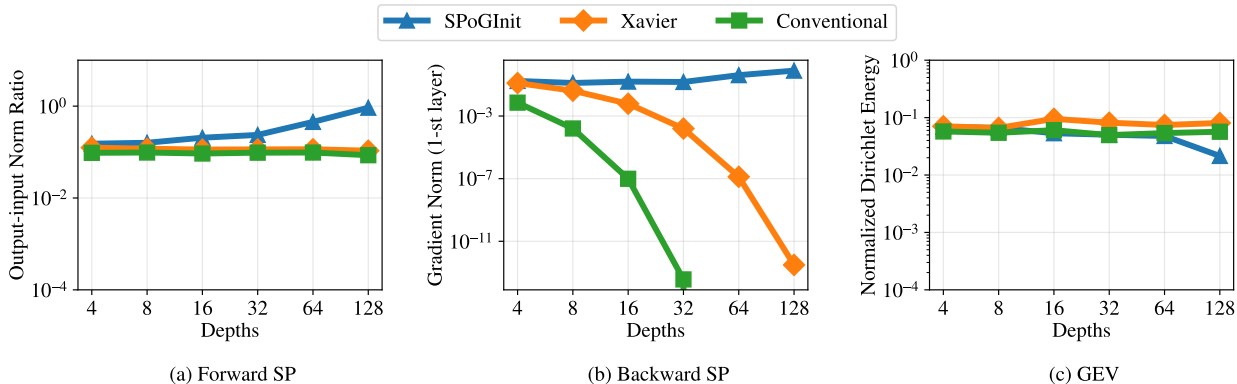

Figure 5: Plots of (a) forward metrics, (b) backward metrics, and (c) graph embedding variation metrics of deep MixHop with different depths on Cora. SPoGInit is highly effective in stabilizing all three SP metrics.

Figure 5 shows that as the depth increases in the MixHop model, Conventional and Xavier initializations succeed in maintaining forward SP and the GEV. However, the backward SP significantly diminishes, leading to severe gradient vanishing in deep models. SPoGInit, on the other hand, effectively stabilizes all three SP metrics.

In summary, these results demonstrate that SPoGInit is capable of finding initializations that stabilize these three SP metrics, thereby alleviating the training difficulties associated with deep models and mitigating the performance degradation as the networks go deeper.

**Can SPoGInit Alleviate Performance Degradation in Deep GCN Models (Q2)?** We present the detailed numerical results for GCN models with varying depths and initializations across four mainstream datasets, shown in Table 2. More details of hyperparameters are provided in Appendix G.1.

Table 2 demonstrates that SPoGInit significantly reduces performance degradation compared to baseline initializations across various GCN models and datasets. For certain models and tasks, SPoGInit can even enhance the performance consistently as network depth increases from 4 to 64, unleashing the potential of deep GCNs. Specifically, in deep ResGCN and gatResGCN models, baseline initializations cause notable test accuracy drops exceeding 30% on the OGBN-Arxiv dataset and 15% on the Arxiv-year dataset compared to their shallow counterparts. In contrast, deep ResGCN and gatResGCN models with SPoGInit achieve performance gains of from 1.2% to 3.0% as depth increases from 4 to 64 across all the tested datasets. For deep vanilla GCNs and MixHop models, SPoGInit exhibits substantially less accuracy decline than baseline initializations on most of the tested datasets. For example, SPoGInit reduces the test accuracy drop for the 64-layer MixHop by around at least 20% on both Cora and OGBN-Arxiv datasets.

Furthermore, we emphasize that SPoGInit exhibits greater robustness over various GCN models. Specifically, we see that Xavier, VirgoFor, and VirgoBack perform well on deep vanilla GCNs, all surpassing Conventional initialization on the four datasets. However, their performance significantly degrades on deep ResGCNs. In contrast, SPoGInit demonstrates excellent versatility, achieving less performance degradation across models of varying depths and architectures.

**Further exploration on SPoGInit.** SPoGInit, as previously introduced, begins with a given starting initialization and searches for a more stable signal propagation (SP) by solving an optimization problem (2). In most cases, we select Xavier initialization as the starting point for SPoGInit in the experiments above. Now we explore the **adaptability of SPoGInit by studying how it performs when starting from different initializations**.

We report the performance of vanilla GCNs on the OGBN-Arxiv dataset, with four baseline initializations, alongside SPoGInit which is applied starting from each of these four initializations, as shown in Table 3. The results indicate that SPoGInit can effectively incorporate different initializations, leading to better

Table 2: Test accuracies of GCN models with varying depths and initializations. The bold figure highlights the best performance among different initializations. "Deg" refers to the test accuracy degradation as the depth increases from 4 to 64 layers. The smallest performance drops are highlighted in orange. The results demonstrate that SPoGInit significantly reduces performance degradation compared to baseline initializations and enhances the performance of deep GNN models across different architectures.

| Model | Init. | Cora | | | | | | Arxiv | | | | | |
|---|---|---|---|---|---|---|---|---|---|---|---|---|---|
| | | 4 | 8 | 16 | 32 | 64 | Deg. | 4 | 8 | 16 | 32 | 64 | Deg. |
| GCN | Conventional | 79.3 | 71.2 | 56.6 | 48.5 | 33.3 | ↓46.0 | 69.2 | 67.5 | 44.9 | 39.1 | 8.0 | ↓61.2 |
| | Xavier | 80.3 | 79.1 | 75.2 | 72.8 | 71.5 | ↓8.8 | 69.6 | 69.3 | 64.1 | 57.9 | 38.1 | ↓31.5 |
| | VirgoFor | **80.4** | **80.0** | 76.7 | 74.3 | 73.4 | ↓7.0 | **69.9** | 69.4 | **69.3** | 67.0 | **61.4** | ↓8.5 |
| | VirgoBack | 80.3 | 77.4 | 74.9 | 74.3 | 73.2 | ↓7.1 | 69.7 | **69.5** | 69.2 | **67.5** | 60.9 | ↓8.8 |
| | SPoGInit | 79.8 | 79.6 | **78.2** | **75.8** | **73.7** | ↓6.1 | 69.8 | 69.1 | 66.8 | 63.8 | 48.4 | ↓21.4 |
| ResGCN | Conventional | 78.0 | 78.5 | 77.5 | 77.6 | 78.2 | ↑0.2 | 70.3 | **71.6** | 72.0 | 70.4 | 34.8 | ↓35.5 |
| | Xavier | 78.0 | 79.1 | 77.7 | 76.8 | 71.6 | ↓6.4 | 70.5 | **71.6** | 70.7 | 53.0 | 16.5 | ↓54.0 |
| | VirgoFor | 78.5 | **78.6** | 77.5 | 74.1 | 54.2 | ↓24.3 | 70.5 | 71.0 | 66.4 | 20.3 | 11.3 | ↓59.2 |
| | VirgoBack | **79.3** | 78.2 | 77.8 | 73.9 | 29.2 | ↓50.1 | **70.6** | 71.1 | 66.5 | 20.1 | 12.5 | ↓58.1 |
| | SPoGInit | 75.7 | 77.9 | **78.5** | **78.5** | **80.1** | ↑4.4 | 70.4 | 71.5 | **72.3** | **71.8** | **71.3** | ↑0.9 |
| gatResGCN | Conventional | 77.4 | 78.2 | 78.0 | 77.9 | 77.0 | ↓0.4 | **70.7** | **71.8** | 71.6 | 70.0 | 27.9 | ↓42.8 |
| | Xavier | 77.9 | **78.5** | 76.6 | 77.4 | 73.2 | ↓4.7 | 70.5 | 71.4 | 70.8 | 45.1 | 16.5 | ↓54.0 |
| | VirgoFor | 78.9 | 78.1 | 77.6 | 70.5 | 35.5 | ↓43.4 | 70.4 | 70.7 | 65.7 | 20.4 | 11.3 | ↓59.1 |
| | VirgoBack | **79.0** | 78.3 | 76.2 | 70.5 | 37.9 | ↓41.1 | 70.5 | 70.7 | 65.9 | 20.1 | 12.5 | ↓58.0 |
| | SPoGInit | 76.3 | 77.8 | **78.2** | **78.1** | **77.6** | ↑1.3 | 70.2 | 71.5 | **72.1** | **72.5** | **72.8** | ↑2.6 |
| MixHop | Conventional | 72.5 | 52.0 | 36.4 | 46.9 | 42.0 | ↓30.5 | 68.0 | 64.1 | 59.8 | 52.6 | 38.2 | ↓29.8 |
| | Xavier | 79.3 | 75.1 | 71.6 | 64.3 | 56.3 | ↓23.0 | 67.9 | 64.0 | 60.1 | 53.0 | 38.0 | ↓29.9 |
| | SPoGInit | **79.6** | **75.0** | **76.8** | **72.0** | **72.2** | ↓7.4 | **69.9** | **70.9** | **70.3** | **68.6** | **61.5** | ↓8.4 |

| Model | Init. | PubMed | | | | | | Arxiv-year | | | | | |
|---|---|---|---|---|---|---|---|---|---|---|---|---|---|
| | | 4 | 8 | 16 | 32 | 64 | Deg. | 4 | 8 | 16 | 32 | 64 | Deg. |
| GCN | Conventional | 75.9 | 68.1 | 67.1 | 68.0 | 60.8 | ↓15.1 | **44.1** | **42.7** | 44.2 | 43.6 | 39.9 | ↓4.2 |
| | Xavier | 78.1 | 76.8 | 76.0 | 77.2 | 75.7 | ↓2.4 | 44.0 | 42.3 | **45.5** | **45.6** | **43.9** | ↓0.1 |
| | VigorFor | **78.9** | **78.5** | **78.2** | 77.8 | 75.9 | ↓3.0 | 43.9 | 30.0 | 45.4 | **45.6** | 41.8 | ↓2.1 |
| | VigorBack | 78.1 | 75.5 | 76.5 | 76.0 | 74.7 | ↓3.4 | 43.7 | 29.8 | **45.5** | 45.4 | 41.8 | ↓1.9 |
| | SPoGInit | 77.4 | 77.5 | 77.0 | **78.4** | **78.1** | ↑0.7 | 43.9 | 41.9 | 45.2 | 44.9 | **43.9** | 0 |
| ResGCN | Conventional | 74.9 | 76.1 | 76.0 | 76.8 | 76.6 | ↑1.7 | 48.2 | 49.6 | 49.7 | 43.6 | 32.6 | ↓15.6 |
| | Xavier | 75.8 | 77.5 | 75.7 | 76.5 | 74.8 | ↓1.0 | **48.4** | 49.0 | 46.0 | 31.9 | 23.0 | ↓25.4 |
| | VigorFor | 76.2 | **77.6** | 76.6 | 76.1 | 74.5 | ↓1.7 | 48.3 | 48.4 | 38.9 | 29.6 | 26.2 | ↓22.1 |
| | VigorBack | **77.0** | **77.6** | **76.9** | **77.0** | 74.9 | ↓2.1 | 48.1 | 48.1 | 38.6 | 29.1 | 24.9 | ↓23.2 |
| | SPoGInit | 75.4 | 76.2 | 76.4 | **77.0** | **77.4** | ↑2.0 | 47.7 | **49.8** | **50.9** | **51.9** | **49.3** | ↑1.6 |
| gatResGCN | Conventional | 74.4 | 76.9 | 76.0 | 76.6 | 76.0 | ↑1.6 | **48.7** | **50.5** | 49.5 | 41.6 | 31.8 | ↓16.9 |
| | Xavier | 76.3 | 77.6 | 75.6 | **77.2** | 75.8 | ↓0.5 | 48.5 | 49.5 | 45.8 | 32.6 | 23.0 | ↓25.5 |
| | VigorFor | 76.2 | 77.6 | **77.1** | 76.1 | 74.3 | ↓1.9 | 48.5 | 48.4 | 38.9 | 31.5 | 26.2 | ↓22.3 |
| | VirgoBack | **76.6** | **77.9** | 76.8 | 75.8 | 75.0 | ↓1.6 | 48.3 | 48.1 | 38.8 | 29.6 | 25.0 | ↓23.3 |
| | SPoGInit | 74.8 | 75.9 | 75.6 | 76.7 | **77.2** | ↑2.4 | 47.9 | 49.3 | **50.0** | **50.6** | **51.0** | ↑3.1 |
| MixHop | Conventional | 73.3 | 65.1 | 68.1 | 65.9 | 56.4 | ↓16.9 | 47.8 | 48.6 | 49.2 | 47.7 | 42.3 | ↓5.5 |
| | Xavier | 76.6 | 76.4 | 72.5 | 72.4 | 71.1 | ↓5.5 | 47.9 | 48.6 | 49.2 | 48.5 | 42.0 | ↓5.9 |
| | SPoGInit | **76.8** | **77.1** | **75.2** | **76.3** | **74.3** | ↓2.5 | **48.3** | **50.2** | **51.3** | **52.0** | **50.5** | ↑2.2 |

performance. Specifically, when starting from the Xavier and Conventional initializations, SPoGInit achieves nearly 10% and 50% improvements in vanilla GCNs with 64 layers compared to these two initializations. Furthermore, when starting from the GNN-based initializations (VirgoFor and VirgoBack), SPoGInit also delivers better performance in deep GCNs. However, it is important to note that when using ResGCN, VirgoFor and VirgoBack exhibit significant performance degradation. Despite this, starting from these GNN-based initializations, SPoGInit still significantly improves performance by more than 58% in deep ResGCNs with 64 layers.

These results demonstrate that although different GNN architectures may favor specific initializations, starting from these initializations, SPoGInit can further enhance the performance. Its strong adaptability allows

Table 3: Test accuracies of GCN models with varying depths and initializations on the OGBN-Arxiv dataset. "Improv." refers to the test accuracy changes after replacing the original initializations with SPoGInit, which starts from those initializations. The results demonstrate that SPoGInit achieves better performance across various models by starting from different initializations.

| Model | Init. | Arxiv | | | | |
|-------|-------|-----|-----|-----|-----|-----|
| | | 4 | 8 | 16 | 32 | 64 |
| GCN | Conventional | 69.2 | 67.5 | 44.9 | 39.1 | 8.0 |
| | +SPoGInit | 69.7 | 69.2 | 68.1 | 63.1 | 56.9 |
| | Improv. | ↑0.5 | ↑1.7 | ↑23.2 | ↑24.0 | ↑48.9 |
| | Xavier | 69.6 | 69.3 | 64.1 | 57.9 | 38.1 |
| | +SPoGInit | 69.8 | 69.1 | 66.8 | 63.8 | 48.4 |
| | Improv. | ↑0.2 | ↓0.2 | ↑2.7 | ↑5.9 | ↑10.3 |
| | VirgoFor | 69.9 | 69.4 | 69.3 | 67.0 | 61.4 |
| | +SPoGInit | 69.6 | 69.6 | 68.7 | 67.2 | 61.4 |
| | Improv. | ↓0.3 | ↑0.2 | ↓0.6 | ↑0.2 | 0 |
| | VirgoBack | 69.7 | 69.5 | 69.2 | 67.5 | 60.9 |
| | +SPoGInit | 69.6 | 69.7 | 69.0 | 67.2 | 61.8 |
| | Improv. | ↓0.1 | ↑0.2 | ↓0.2 | ↓0.3 | ↑0.9 |
| ResGCN | Conventional | 70.3 | 71.6 | 72.0 | 70.4 | 34.8 |
| | +SPoGInit | 70.3 | 71.2 | 71.9 | 71.9 | 71.5 |
| | Improv. | 0 | ↓0.4 | ↓0.1 | ↑1.5 | ↑36.7 |
| | Xavier | 70.5 | 71.6 | 70.7 | 53.0 | 16.5 |
| | +SPoGInit | 70.4 | 71.5 | 72.3 | 71.8 | 71.3 |
| | Improv. | ↓0.1 | ↓0.1 | ↑1.6 | ↑18.8 | ↑54.8 |
| | VirgoFor | 70.5 | 71.0 | 66.4 | 20.3 | 11.3 |
| | +SPoGInit | 70.0 | 71.3 | 72.0 | 72.1 | 71.2 |
| | Improv. | ↓0.5 | ↑0.3 | ↑6.4 | ↑51.8 | ↑59.9 |
| | VirgoBack | 70.6 | 71.1 | 66.5 | 20.1 | 12.5 |
| | +SPoGInit | 70.2 | 71.4 | 72.0 | 72.3 | 71.4 |
| | Improv. | ↓0.4 | ↑0.3 | ↑5.5 | ↑52.2 | ↑58.9 |

SPoGInit to achieve better results across various models by effectively integrating with different initialization techniques.

Additionally, we provide an ablation study on the SP metric components within SPoGInit in Appendix G.2.

### 5.3 Experiments on graph-based tasks involving long-range relationships

**Missing feature settings.** To investigate the performance of SPoGInit on the graph-based tasks involving long-range relationships, we first conduct experiments on the datasets under missing feature settings following (Zhao & Akoglu, 2020). Specifically, we construct graph datasets by zeroing out a designated proportion of node features in the validation and test sets while preserving their corresponding labels. The details of the missing feature settings are provided in Appendix G.1. Within the semi-supervised learning framework, a high proportion of missing features necessitates multiple feature aggregations from the training set for accurate label prediction in the validation and test sets. This approach effectively amplifies the challenge of learning long-range relationships within the dataset.

We adopt the missing feature setting with different missing proportions on the Arxiv-year dataset. We examine the performance of various GCN models with different depths and initializations over these settings. Table 4 presents the best performance of different GCN models, ranging from 8 layers to 64 layers, along with the optimal depth that achieves the best performance. The results indicate that under higher feature missing proportion, the models with SPoGInit tend to achieve their best performance at larger depths. SPoGInit demonstrates a significant improvement over baseline initializations, especially under 100% missing proportion, i.e., where all node features in the validation and test sets are missing. Moreover, deep ResGCN

Table 4: The optimal performance of various deep GCN models, ranging from 8 to 64 layers, with different initializations under missing feature settings. The figures in parentheses denote the depth corresponding to the optimal performance. Bold figures show the best performance for each model and the best performances across all architectures are highlighted in orange. The term "optimal improvement" refers to the maximum performance enhancements achieved by SPoGInit compared to the best performance of baseline initializations across the various models. The results indicate that optimal performance on datasets with long-range relationships is attained at greater depths, and SPoGInit enhances the performance of deep GCN architectures.

| Model | Init. | Missing 50% | Missing 100% |
|---|---|---|---|
| GCN | Conventional | 43.1 (16) | 41.6 (16) |
| | Xavier | **43.3 (16)** | 42.5 (16) |
| | VirgoFor | 40.1 (16) | 41.9 (16) |
| | VirgoBack | 42.0 (16) | 42.0 (16) |
| | SPoGInit | 43.1 (32) | **42.4 (16)** |
| ResGCN | Conventional | 45.8 (16) | 43.9 (16) |
| | Xavier | 45.2 (8) | 43.1 (8) |
| | VirgoFor | 45.4 (8) | 43.0 (8) |
| | VirgoBack | 45.2 (8) | 42.9 (8) |
| | SPoGInit | **46.3 (64)** | **45.1 (64)** |
| gatResGCN | Conventional | **45.5 (16)** | 43.3 (16) |
| | Xavier | 44.7 (8) | 42.2 (8) |
| | VirgoFor | 45.2 (8) | 42.6 (8) |
| | VirgoBack | 45.0 (8) | 42.6 (8) |
| | SPoGInit | 45.3 (16) | **43.4 (32)** |
| MixHop | Conventional | **45.1 (8)** | 42.9 (16) |
| | Xavier | **45.1 (8)** | 42.8 (16) |
| | SPoGInit | 44.9 (8) | **44.6 (16)** |
| Optimal Improvements | | +0.5 | +1.7 |

with SPoGInit achieves the best performance across all the tested architectures and initializations. These findings demonstrate the potential of deep GCNs in graph-based tasks involving long-range relationships and the effectiveness of SPoGInit in unleashing such potential.

**GCNs for Combinatorial Optimization.** In recent years, GNNs have emerged as a powerful tool for addressing Combinatorial Optimization (CO) problems, which are fundamental in areas such as computer science and operations research (Paschos, 2014). Many classical CO problems are extremely difficult to solve due to their NP-hardness (Bomze et al., 1999; Gavish & Graves, 1978; Coffman Jr et al., 1984). To address these difficult CO problems, expert-designed heuristic algorithms (Boussaïd et al., 2013) are developed to find near-optimal solutions within reasonable computational limits. Over the past decade, machine learning-based methods (Bengio et al., 2021) have gained significant interest in tackling CO problems. It has been shown that with sufficient data and proper training, neural networks have the potential of surpassing expert-designed methods in both performance and efficiency (Alvarez et al., 2017; Khalil et al., 2016; 2017). Given that many CO problems naturally have a graph structure, GNNs have become a highly promising approach (Gasse et al., 2019).

In this work, we consider the Maximal Independent Set (MIS) problem, a classic CO problem in graph theory with significant applications in network analysis, wireless communication, etc. Given an undirected graph $\mathcal{G} = (\mathcal{V}, \mathcal{E})$, the objective is to find a subset $\mathcal{S} \subseteq \mathcal{V}$ of vertices such that $\mathcal{S}$ is independent, meaning no two vertices in $\mathcal{S}$ are adjacent, and $\mathcal{S}$ is maximal, implying that no additional vertex can be added to $\mathcal{S}$ without violating its independence.

We note that MIS problems can involve inherent long-range relationships between nodes. The inclusion of a node in the independent set may significantly affect all other nodes in the set. For example, consider a

cycle graph $\mathcal{G}$ with an even number of nodes $V_1, V_2, \ldots, V_{4m}$. In this cycle, each node connects to two others, forming a closed loop. If $V_1$ is included in $\mathcal{S}$, its adjacent nodes $V_{4m}$ and $V_2$ cannot be part of $\mathcal{S}$, allowing nodes $V_{4m-1}$ and $V_3$ to potentially be included in $\mathcal{S}$. This cascading effect continues around the cycle, and it is straightforward to show that the maximal independent set is $\{V_1, V_3, \ldots, V_{2m-1}, V_{2m+1}, \ldots, V_{4m-1}\}$. Similarly, if $V_1$ is excluded from $\mathcal{S}$, the maximal independent set becomes $\{V_2, V_4, \ldots, V_{2m}, \ldots, V_{4m-1}\}$. Notably, the predictions of whether $V_1$ and $V_{2m}$ are in the maximal independent set depend on each other, while the two nodes have a distance of $2m - 1$.

In this work, we follow the setting in Han et al. (2022) and apply GNN to solve MIS problems. The MIS problem is typically formulated as an Integer Linear Programming (ILP) problem:

$$\max_x \sum_{i \in \mathcal{V}} x_i$$
$$\text{s.t.} \quad x_i + x_j \leq 1, \ \forall (i, j) \in \mathcal{E},$$
$$x_i \in \{0, 1\}.$$

Based on this formulation, a variable-constraint bipartite graph representation is constructed. The set of variable nodes $\mathcal{V}$ (each denoted by $i$) is placed on one side of the bipartite graph, while the set of constraint nodes $\mathcal{E}$ (each representing an edge $(i, j)$ in the original graph) is placed on the other side. Each variable node $i$ and $j$ is connected to the corresponding constraint node $(i, j)$. Following Han et al. (2022), we adopt a predict and search framework. The predicting stage utilizes GNN over the bipartite graph representation to find a good initial solution to the MIS problem, while the search stage refines this solution using the traditional ILP solver such as SCIP or Gurobi to obtain the final result.

We investigate the power of SPoGInit in improving the performance of bipartite GCNs for solving MIS problems. We utilize the Independent Set (IS) dataset (Bergman et al., 2016), where each instance comprises 600 constraints and 1500 binary variables. The bipartite GCN models are trained on 80 problem instances and then applied to predict the optimal solution (0 or 1) for each node across 20 test instances. We set the learning rates as 1e-3, 5e-4, and 5e-5, for bipartite GCN models with 2, 8, and 16 layers, respectively.

The experimental results, shown in Table 5, indicate that bipartite GCNs often achieve performance improvements as their depths increase. With Xavier initialization, there is approximately a 4% increase in accuracy from 2 to 16 layers. Moreover, employing SPoGInit yields greater performance enhancements, with an approximate 7% increase in accuracy from 2 to 16 layers, and surpasses the baseline methods on the 16-layer network.

Table 5: Test accuracy of bipartite GNN models on ILP tasks across varying initializations and depths. The results underscore the critical role of depth in solving ILP problems with Bipartite GCNs. Furthermore, SPoGInit significantly boosts the performance of deep Bipartite GCNs, leading to optimal accuracy.

| Model | Init. | 2 layer | 8 layer | 16 layer |
|---|---|---|---|---|
| | Conventional | **78.9** | 82.7 | 84.1 |
| Bipartite GCN | Xavier | 78.5 | 81.0 | 82.2 |
| | SPoGInit | 78.7 | **83.0** | **85.2** |
| | Conventional | **79.4** | **85.4** | 85.5 |
| Bipartite GCN with skip connection | Xavier | **79.4** | 82.2 | 48.6 |
| | SPoGInit | **79.4** | 84.7 | **87.5** |

Further, we adopt skip connection in bipartite GCNs and train the models with various depths using the Adam optimizer at a learning rate of 1e-3. We see that incorporating skip connections boosts the performance of Bipartite GCNs. This enhancement is likely attributed to the reinforcement of the initial embeddings associated with the ILP problem. However, Xavier initialization faces training failure issues and fails to deliver substantial benefits during the deepening process, and Conventional initialization shows minimal gains when the network depth is increased from 8 to 16 layers. Conversely, SPoGInit consistently provides ongoing

benefits as the network depths increase. With SPoGInit, the 16-layer bipartite GCN with skip connections achieves optimal performance across all initializations, models, and depths.

## 6 Related works

**Over-smoothing in GCNs.** The concept of over-smoothing is first introduced in Li et al. (2018) to explain the performance degradation in deeper GCNs. This issue is later explored through both theoretical and empirical studies (Oono & Suzuki, 2019; Cai & Wang, 2020; Yang et al., 2020; Maskey et al., 2024; Yang et al., 2024; Chen et al., 2020a; Nguyen et al., 2023; Rusch et al., 2023b; Roth & Liebig, 2024; Roth, 2024; Luan et al., 2020; Cong et al., 2021; Zhang et al., 2022a). While the smoothing effect of graph convolution may benefit shallow GCNs (Keriven, 2022; Wu et al., 2023), it adversely affect the performance of deep GCNs.

To alleviate over-smoothing, various techniques have been proposed (Chen et al., 2022c; Wang et al., 2021), including node or edge dropping (Srivastava et al., 2014; Zou et al., 2019; Rong et al., 2020; Huang et al., 2020; Lu et al., 2021; Fang et al., 2023; Han et al., 2023a; Finkelshtein et al., 2023), normalization methods (Ioffe & Szegedy, 2015; Zhao & Akoglu, 2020; Zhou et al., 2020b; Yang et al., 2020; Zhou et al., 2021b; Li et al., 2020; Guo et al., 2023), and regularization strategies (Chen et al., 2020a; Yang et al., 2020; Zhou et al., 2021a). In addition to these techniques, substantial efforts have been dedicated to modifying GCN architectures, such as incorporating residual connections (Kipf & Welling, 2017; Jaiswal et al., 2022; Chen et al., 2022b; Scholkemper et al., 2024), jumping connections (Xu et al., 2018; Liu et al., 2020; Zhu et al., 2020), and other architectural modifications (Bose & Das, 2023; Di Giovanni et al., 2022; Chien et al., 2021; Gasteiger et al., 2019; Luan et al., 2019; Chen et al., 2020b; Li et al., 2019; Yan et al., 2022; Guo et al., 2022; Min et al., 2020; Chen et al., 2022a; Jin et al., 2022; Zheng et al., 2021; Yang et al., 2023b; Li et al., 2021; Zhang et al., 2020; Feng et al., 2022; Dong et al., 2021; Wu et al., 2024; Choi et al., 2024; Zhang et al., 2022b; Kelesis et al., 2023). In recent years, a line of work has proposed using negative sampling, which incorporates non-neighboring nodes into the aggregation process, to alleviate over-smoothing in GNNs. Duan et al. (2022) introduce determinantal point process (DPP)–based sampling to encourage diversity among selected negative samples. To improve scalability, Duan et al. (2023) construct candidate sets using shortest-path heuristics, significantly reducing computational cost. Building on this, Duan et al. (2024) propose layer-diverse negative sampling, which dynamically adjusts the sampling space across layers to reduce redundancy and improve the expressiveness of deep GNNs. These methods essentially mitigate over-smoothing by modifying the propagation mechanism of GCNs.

Different from these works, our paper investigates the impact of weight initialization to tackle over-smoothing (as well as gradient pathology) in GCNs, without modifying the network architecture or the message passing mechanism. While a few recent works (Guo et al., 2022; Jaiswal et al., 2022; Li et al., 2023) have explored initialization to improve the training of GNN, they do not explicitly treat over-smoothing as one of their primary concerns. Although Han et al. (2023b) applies analog MLP initialization to GNNs, it does not specifically address the performance degradation of deep GNNs.

**Signal propagation.** Classical signal propagation theory has built up a foundation for understanding how information flows through deep neural networks (DNNs) and guides the random weight initialization. At first, Glorot & Bengio (2010); He et al. (2015) study the forward-backward propagation in linear or ReLU-activated models. Then, the mean-field theory (Neal, 1996; Lee et al., 2018; Matthews et al., 2018) is incorporated to study the signal propagation in models with general non-linear activation. Theoretical analysis on fully-connected neural networks (FCNNs) includes the study of Edge-of-Chaos (EOCs) (Poole et al., 2016; Schoenholz et al., 2017; Hayou et al., 2019; 2022) and dynamical isometry (Saxe et al., 2014; Pennington et al., 2017; 2018). Other works study the signal propagation in deep CNN (Xiao et al., 2018), RNN (Chen et al., 2018), ResNet (Yang & Schoenholz, 2017; Hayou et al., 2022), autoencoder (Li & Nguyen, 2019), and LSTM/GRU (Gilboa et al., 2019).

In the realm of GCNs, several recent works have proposed weight initialization techniques to stabilize signal propagation. For the forward pass, Jaiswal et al. (2022) focus on Topology-Aware Isometry, which differs from our focus on stabilizing the output-input norm ratio. For the backward pass, they rely on gradient-guided dynamic rewiring, modifying the model architecture itself. In contrast, we control backward signal propagation

through the BSP metric. Moreover, their method design does not address over-smoothing, while we explicitly incorporate the graph embedding variation (GEV) metric to do so. Kelesis et al. (2024) propose G-Init, which extends Kaiming initialization to the graph domain by incorporating graph topology-aware scaling. We include in Appendix F.6 a comparison experiment between G-Init and SPoGInit on vanilla GCN. The results show that while both G-Init and SPoGInit alleviate performance degradation in deep GCNs, SPoGInit provides more consistent results and smaller accuracy drops across datasets. In addition, Guo et al. (2022); Li et al. (2023) also build their approaches around forward and backward signal propagation, but do not investigate how initialization affects GEV to mitigate over-smoothing in deep GCNs. Also, we note that Han et al. (2023b) introduce MLPInit, which is not aimed at improving signal propagation or reducing over-smoothing. Instead, it seeks to accelerate training by initializing GNNs with the weights of a fully trained, equivalent MLP. This is a fundamentally different motivation from ours.

**Weight initialization search.** In traditional deep learning, some works have explored weight initialization search to improve training stability (Dauphin & Schoenholz, 2019; Zhu et al., 2021). However, their objectives differ from ours. Dauphin & Schoenholz (2019) aims to minimize the curvature effects around the initial parameters by reducing the gradient quotient, which reflects local curvature sensitivity. Zhu et al. (2021) aims at finding an initialization that minimizes the loss after a single training step. Our proposed SPoGInit, however, primarily targets mitigating signal propagation instability in the initialization of deep GCNs, by addressing both forward and backward SP as well as the over-smoothing issue.

**Infinite-width-limit regime.** Our analysis relies on an infinite-width assumption—a common approach in traditional theoretical analyses of neural networks (see, e.g., Lee et al. (2018); Sohl-Dickstein et al. (2020); Yang et al. (2024)). This assumption offers the primary benefit of ensuring that the feature embeddings at each layer follow a Gaussian distribution under random initialization, thereby simplifying the theoretical treatment. Such a Gaussian property is not strictly guaranteed for practical, finite-width networks.

Nonetheless, we empirically demonstrate that the embeddings of a finite-width network are approximately Gaussian, implying that the infinite-width approximation remains reasonable in practice. Specifically, we randomly selected three node embeddings from the final convolutional layer of a 4-layer, 64-width GCN network trained on the OGBN-Arxiv dataset. We then applied the Kolmogorov-Smirnov hypothesis test to evaluate their distribution. In this test, a p-value below 0.05 would indicate a significant deviation from a Gaussian distribution. The observed p-values—0.75, 0.90, and 0.73—suggest that the embeddings closely follow a Gaussian distribution. Thus, despite the theoretical limitation, our empirical findings support the validity of using the infinite-width approximation.

**Relationship with GNTK.** NNGP (Neural Network Gaussian Process) for GNNs and GNTK (Graph Neural Tangent Kernel) are two theoretical tools used to analyze the behavior of graph neural networks. While they share the common goal of characterizing GNNs in the infinite-width limit, they differ in focus: NNGP primarily describes the properties of feature embeddings at initialization, whereas GNTK captures the training dynamics of GNNs, particularly how the model converges under gradient descent. Some existing works study graph neural tangent kernel (GNTK) (Bayer et al., 2022; Du et al., 2019; Huang et al., 2022; Jiang et al., 2022; Sabanayagam et al., 2021; 2022; Zhou & Wang, 2022; Gebhart, 2022; Krishnagopal & Ruiz, 2023; Yang et al., 2023a). They analyze the training dynamics of GCNs under the infinite-width limit.

## 7 Limitations and future works

SPoGInit has been specifically designed and tested for node classification tasks within GNNs. While the method has shown promising results in this context, its applicability to other tasks, such as edge prediction or graph classification, remains unexplored. Adapting SPoGInit for a wider range of graph-based learning tasks is as an important direction for future work.

Although SPoGInit is designed as a general method for GNNs, our current implementation is not entirely plug-and-play for all GNN architectures. The integration of SPoGInit into novel or non-standard GCN variants (e.g., GCNII (Chen et al., 2020b), GraphTransformer (Shehzad et al., 2024)) will require additional

engineering efforts. Future work could focus on creating more robust and flexible integration methods for a broader range of GNN models, enhancing SPoGInit's adaptability.

As highlighted in our analysis, SPoGInit introduces an approximate 10%-20% computational overhead. While we believe this overhead is acceptable in many practical scenarios, it could become a limiting factor in situations where computational resources are severely constrained. Future work could explore methods to reduce the computational cost of SPoGInit, either through more efficient optimization techniques or by leveraging hardware acceleration.

## 8 Conclusion

We attempt to address the performance degradation of deep GCNs from the lens of signal propagation. We consider three metrics: forward propagation, backward propagation, and graph embedding variation propagation. Our theoretical analysis and empirical studies revealed that widely used initialization methods in GCNs fail to control these metrics simultaneously, resulting in undesirable performance degradation as depth increases. Motivated by our SP framework, a new initialization method, termed SPoGInit, is proposed. The experiment results demonstrate that SPoGInit enhances the signal propagation of various deep GCN architectures. Moreover, SPoGInit significantly mitigates performance degradation or enables performance enhancement as depths increase, especially in graph-based tasks involving long-range relationships. Both our theoretical and empirical findings underscore the importance of stabilizing these three SP metrics for boosting the performance of deep GCNs.

## Acknowledgement

The authors would like to express our sincere gratitude to Kenta Oono and the anonymous reviewers for their insightful feedback during the discussion phase. The authors also thank Bingheng Li and Haitao Mao for their helpful suggestions during the early stages of revision. This paper is supported in part by the National Key Research and Development Project under grant 2022YFA1003900; Hetao Shenzhen-Hong Kong Science and Technology Innovation Cooperation Zone Project (No.HZQSWS-KCCYB-2024016); University Development Fund UDF01001491, the Chinese University of Hong Kong, Shenzhen; Guangdong Provincial Key Laboratory of Mathematical Foundations for Artificial Intelligence (2023B1212010001); the Guangdong Major Project of Basic and Applied Basic Research (2023B0303000001).

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

# Appendix

## A  Supplemental notation

For any integer $n \in \mathbb{N}$, we define $[n] \triangleq \{1, 2, \dots, n\}$. We may denote a matrix $X \in \mathbb{R}^{m \times n}$ by $(x_{ij})_{i \in [m], j \in [n]}$, where $x_{ij}$ is the entry in the $i$-the row and the $j$-th column. We further use $X_{i,:} \in \mathbb{R}^{1 \times n}$ and $X_{:,j} \in \mathbb{R}^{m \times 1}$ to denote the $i$-th row and the $j$-th column of $X$, respectively. $\| \cdot \|_{\mathrm{F}}$ denotes the Frobenius norm. Given any function $f : \mathbb{R}^{m \times n} \to \mathbb{R}$, its derivative $\partial f / \partial X$ with respect to $X \in \mathbb{R}^{m \times n}$ is the $m \times n$ matrix with $(\partial f / \partial X)_{ij} = \partial f(X) / \partial x_{ij}$. For any activation function $\sigma : \mathbb{R} \to \mathbb{R}$, we use $\sigma(X) \in \mathbb{R}^{m \times n}$ to denote the output of applying $\sigma$ entry-wise to the matrix $X$, i.e., $(\sigma(X))_{ij} = \sigma(x_{ij})$. We denote ReLU activation by $\mathrm{ReLU}(x) \triangleq \max(0, x)$ and tanh activation by $\tanh(x) \triangleq (e^x - e^{-x})/(e^x + e^{-x})$. For brevity, we use $\theta$ to denote the collection of all trainable parameters in a GCN model.

For any matrix $X = (x_{ij}) \in \mathbb{R}^{m \times n}$, the vectorizaion of $X$ is defined by

$$\mathrm{vec}(X) := (x_{11}, \dots, x_{m1}, x_{12}, \dots, x_{m2}, \dots, x_{1n}, \dots, x_{mn})^T \in \mathbb{R}^{mn \times 1}.$$

For any matrix $X = (x_{ij}) \in \mathbb{R}^{m \times n}$ and $Y = (y_{ij}) \in \mathbb{R}^{p \times q}$, the Kronecker product of $X$ and $Y$ is a $mp \times nq$ block matrix defined by

$$X \otimes Y := \begin{pmatrix} x_{11}Y & \dots & x_{1n}Y \\ \vdots & \ddots & \vdots \\ x_{m1}Y & \dots & x_{mn}Y \end{pmatrix}.$$

For a matrix $X = (x_{ij}) \in \mathbb{R}^{m \times n}$, if $x_{ij} = 0$ for all $i \in [m]$ and $j \in [n]$, we denote $X = \mathbf{0}_{m \times n}$; if $x_{ij} = 1$ for all $i \in [m]$ and $j \in [n]$, we denote $X = \mathbf{1}_{m \times n}$. For a vector $Z = (z_i) \in \mathbb{R}^n$, if $z_i = 0$ for all $i \in [n]$, we denote $Z = \mathbf{0}_n$; if $z_i = 1$ for all $i \in [n]$, we denote $Z = \mathbf{1}_n$.

# B Convolutional kernel

Suppose that graph $\mathcal{G}$ has $M$ connected components. The $m$-th component is a subgraph denoted by $\mathcal{G}_m = (\mathcal{V}_m, \mathcal{E}_m)$ for $m \in [M]$. We present a well-known result characterizing the eigenvalues and the eigenvectors of $\hat{A}$ without giving proof, see, e.g., Proposition 1 in Oono & Suzuki (2019).

**Proposition B.1.** *Suppose that $\mathcal{G} = (\mathcal{V}, \mathcal{E})$ has $M$ connected components $\{\mathcal{G}_m = (\mathcal{V}_m, \mathcal{E}_m)\}_{m=1}^{M}$ and the eigenvalues of $\hat{A}$ are $\lambda_1 \geq \lambda_2 \geq \cdots \geq \lambda_n$. Then we have*

- $\lambda_i = 1$, *for any $1 \leq i \leq M$.*

- $\lambda_i \in (-1, 1)$, *for any $M + 1 \leq i \leq n$.*

*Moreover, the set $\{v^{(m)} = \tilde{D}^{\frac{1}{2}} u^{(m)} : m \in [M]\}$ is a basis of the $m$-dimensional eigenspace of $\hat{A}$ corresponding to the eigenvalue 1, where $u^{(m)} = (\mathbb{1}_{\{i \in \mathcal{V}_m\}})_{i \in [n]} \in \mathbb{R}^{n \times 1}$ is the indicator vector of the $m$-th connected component $\mathcal{G}_m$.*

**Lemma B.2.** *Given any $H \in \mathbb{R}^{n \times C}$ and $H \neq \mathbf{0}_{n \times C}$, we have $0 \leq \mathrm{Dir}(H)/\|H\|_{\mathrm{F}}^2 \leq 2$.*

*Proof.* Recall that $\hat{L} = I - \hat{A}$ is the normalized Laplacian of graph $\mathcal{G}$. By Proposition B.1, all the eigenvalues of $\hat{L}$ belong to $[0, 2)$.

Given any $H \in \mathbb{R}^{n \times C}$, we have

$$\mathrm{Dir}(H) = \mathrm{tr}(H^T \hat{L} H) = \sum_{k=1}^{C} H_{:,k}^T \hat{L} H_{:,k} \leq \sum_{k=1}^{C} 2 \cdot H_{:,k}^T H_{:,k} = 2\|H\|_{\mathrm{F}}^2.$$

Similarly, we have

$$\mathrm{Dir}(H) = \mathrm{tr}(H^T \hat{L} H) = \sum_{k=1}^{C} H_{:,k}^T \hat{L} H_{:,k} \geq \sum_{k=1}^{C} 0 \cdot H_{:,k}^T H_{:,k} = 0.$$

Therefore, we conclude that

$$0 \leq \mathrm{Dir}(H)/\|H\|_{\mathrm{F}}^2 \leq 2.$$

$\square$

## C Signal propagation theory for vanilla GCN

### C.1 NNGP correspondence for vanilla GCN

**Proposition C.1** (NNGP correspondence for vanilla GCN). *As the network widths $d_1, d_2, \ldots, d_{L-1}$ sequentially go to infinity, the l-th layer's pre-activation embedding channels $\{H_{:,k}^{(l)}\}_{k \in [d_l]}$ converge to i.i.d. n-dimensional Gaussian random variables $N(\mathbf{0}_n, \Sigma^{(l)})$ in distribution for any $l \geq 2$. The covariance matrices are*

$$\Sigma^{(1)} = \frac{\sigma_w^2}{d_0} \hat{A} X X^T \hat{A},$$
$$\Sigma^{(l+1)} = \sigma_w^2 \hat{A} G(\Sigma^{(l)}) \hat{A}, \tag{3}$$

*where $G(\Sigma) = \mathbb{E}_{h \sim N(\mathbf{0}_n, \Sigma)}[\sigma(h)\sigma(h)^T]$ for any $n \times n$ positive semi-definite matrix $\Sigma$.*

*Proof of Proposition C.1.* We will prove that $\{H_{:,k}^{(l)}\}_{k \in [d_l]}$ are asymptotically i.i.d. $n$-dimensional random variables with mean $\mathbf{0}_n$ and covariance matrix $\Sigma^{(l)}$ for any $l \geq 1$ under the infinite width limit by mathematical induction. Proposition C.1, which contains a stronger claim that $\{H_{:,k}^{(l)}\}$ are asymptotically Gaussian for any $l \geq 2$, will be shown during the induction steps.

**Base case.** Since the bias terms are initialized to be zero, when $l = 1$, the $k$-th channel of the embedding is

$$H_{:,k}^{(1)} = \hat{A} X W_{:,k}^{(1)} + \mathbf{1}_n \cdot b_k^{(1)} = \hat{A} X W_{:,k}^{(1)}. \tag{4}$$

Since $\{W_{:,k}^{(1)}\}_{k \in [d_1]}$ are i.i.d. random variables, so $\{H_{:,k}^{(1)}\}_{k \in [d_1]}$ are also i.i.d. random variables. Taking the expectation of (4), we get

$$\mathbb{E}[H_{:,k}^{(1)}] = \hat{A} X \cdot \mathbb{E}[W_{:,k}^{(1)}] = \mathbf{0}_n.$$

Calculating the covariance matrix of (4), we have

$$\mathrm{Cov}[H_{:,k}^{(1)}, H_{:,k}^{(1)}] = \mathbb{E}[H_{:,k}^{(1)} \cdot H_{:,k}^{(1)T}] = \mathbb{E}[\hat{A} X W_{:,k}^{(1)} W_{:,k}^{(1)T} X^T \hat{A}]$$
$$= \hat{A} X \cdot \mathbb{E}[W_{:,k}^{(1)} W_{:,k}^{(1)T}] \cdot X^T \hat{A} = \hat{A} X \cdot \left( \frac{\sigma_w^2}{d_0} \cdot I_{d_0} \right) \cdot X^T \hat{A}$$
$$= \frac{\sigma_w^2}{d_0} \hat{A} X X^T \hat{A}.$$

Thus, if we define $\Sigma^{(1)} = \sigma_w^2 \hat{A} X X^T \hat{A}/d_0$, then $\{H_{:,k}^{(1)}\}_{k \in [d_1]}$ are exactly i.i.d with mean $\mathbf{0}_n$ and covariance matrix $\Sigma^{(1)}$.

**Induction step.** Suppose that $\{H_{:,k}^{(l)}\}_{k \in [d_l]}$ converge to i.i.d. $n$-dimensional random variables with mean $\mathbf{0}_n$ and covariance matrix $\Sigma^{(l)}$ in distribution as $d_1, \ldots, d_{l-1}$ sequentially go to infinity, we look at the $(l+1)$-th layer. Recall from the formation of the $l$-th layer in vanilla GCN, we have

$$H^{(l+1)} = \hat{A} X^{(l)} W^{(l+1)} + \mathbf{1}_n \cdot b^{(l+1)},$$
$$X^{(l)} = \sigma(H^{(l)}),$$

for any $l \geq 1$. We vectorize the first equation and get

$$\mathrm{vec}(H^{(l+1)}) = \mathrm{vec}(\hat{A} X^{(l)} W^{(l+1)}) + \mathrm{vec}(\mathbf{1}_n \cdot b^{(l+1)})$$
$$= \sum_{k=1}^{d_l} \mathrm{vec} \left( \underbrace{[\hat{A} X_{:,k}^{(l)}]}_{n \times 1} \cdot \underbrace{W_{k,:}^{(l+1)}}_{1 \times d_{l+1}} \right), \tag{5}$$

because $b^{(l+1)}$ is initialized to be $\mathbf{0}_{d_{l+1}}$. Suppose that $\Sigma^{(l+1)} = \sigma_w^2 \hat{A} G(\Sigma^{(l)}) \hat{A}$, we are going to show that $\mathrm{vec}(H^{(l+1)})$ converges to a Gaussian random variable $N(\mathbf{0}_{nd_{l+1}}, I_{d_{l+1}} \otimes \Sigma^{(l+1)})$ in distribution as

$d_1, d_2, \ldots, d_{l-1}, d_l$ sequentially go to infinity. If this claim holds, $\{H_{:,k}^{(l+1)}\}$ are not only asymptotically i.i.d., but also asymptotically Gaussian i.i.d. with $N(\mathbf{0}_n, \Sigma^{(l+1)})$, which corresponds to the statement of this proposition.

For brevity, we define

$$\omega_{kk'}^{(l+1)} := \sqrt{d_l} \cdot W_{kk'}^{(l+1)}, \quad \text{for all } k \in [d_l] \text{ and } k' \in [d_{l+1}],$$

and

$$Z_k^{(l+1)} := \text{vec}\left([\hat{A}X_{:,k}^{(l)}] \cdot \omega_{k,:}^{(l+1)}\right), \quad \text{for all } k \in [d_l]. \tag{6}$$

Then we get that $\{\omega_{kk'}^{(l+1)}\}_{k \in [d_l], k' \in [d_{l+1}]}$ are i.i.d. from $N(0, \sigma_w^2)$ and

$$\text{RHS of (5)} = \frac{1}{\sqrt{d_l}} \sum_{k=1}^{d_l} Z_k^{(l+1)}. \tag{7}$$

By the induction hypothesis, as $d_1, d_2, \ldots, d_{l-1}$ sequentially go to infinity, $\{X_{:,k}^{(l)}\}_{k \in [d_l]} = \{\sigma(H_{:,k}^{(l)})\}_{k \in [d_l]}$ converge to i.i.d. $n$-dimensional random vectors in distribution. Because $X^{(l)}$ can be regarded as a function of $\{W^{(l')}\}_{l'=1}^{l}$ at initialization, we get that $X^{(l)}$ and $W^{(l+1)}$ are independent. Thus, as $d_1, d_2, \ldots, d_{l-1}$ sequentially go to infinity, $\{Z_k^{(l+1)}\}_{k \in [d_l]}$ converge to i.i.d. random vectors in distribution. Moreover, in this limiting case, by taking the expectation of (6), we have

$$\mathbb{E}[Z_1^{(l+1)}] = \text{vec}\left(\left[\hat{A}\mathbb{E}[X_{:,k}^{(l)}]\right] \cdot \mathbb{E}[\omega_{k,:}^{(l+1)}]\right) = \text{vec}\left(\mathbf{0}_{n \times 1} \cdot \mathbf{0}_{1 \times d_{l+1}}\right) = \mathbf{0}_{nd_{l+1}}.$$

Calculating the covariance matrix of (6), we have

$$\begin{aligned}
\text{Cov}[Z_1^{(l+1)}, Z_1^{(l+1)}] &= \mathbb{E}[Z_1^{(l+1)} \cdot Z_1^{(l+1)T}] \\
&= \mathbb{E}\left[\text{vec}\left([\hat{A}X_{:,1}^{(l)}] \cdot \omega_{1,:}^{(l+1)}\right) \cdot \text{vec}\left([\hat{A}X_{:,1}^{(l)}] \cdot \omega_{1,:}^{(l+1)}\right)^T\right] \\
&= \mathbb{E}\left[(\omega_{1,:}^{(l+1)T} \otimes \hat{A}X_{:,1}^{(l)}) \cdot (\omega_{1,:}^{(l+1)} \otimes X_{:,1}^{(l)T}\hat{A})\right] \\
&= \mathbb{E}\left[\omega_{1,:}^{(l+1)T}\omega_{1,:}^{(l+1)} \otimes \hat{A}X_{:,1}^{(l)}X_{:,1}^{(l)T}\hat{A}\right] \\
&= \mathbb{E}\left[\omega_{1,:}^{(l+1)T}\omega_{1,:}^{(l+1)}\right] \otimes \left\{\hat{A} \cdot \mathbb{E}\left[X_{:,1}^{(l)}X_{:,1}^{(l)T}\right] \cdot \hat{A}\right\} \\
&= \sigma_w^2 I_{d_{l+1}} \otimes \hat{A}G(\Sigma^{(l)})\hat{A} \\
&= I_{d_{l+1}} \otimes \sigma_w^2 \hat{A}G(\Sigma^{(l)})\hat{A} = I_{d_{l+1}} \otimes \Sigma^{(l+1)}.
\end{aligned}$$

Here $X_{:,1}^{(l)}$ actually stands for the limit of true $X_{:,1}^{(l)}$ as $d_1, \ldots, d_{l-1}$ sequentially go to infinity without bringing any confusion.

By multivariate central limit theorem, $\frac{1}{\sqrt{d_l}} \sum_{k=1}^{d_l} Z_k^{(l+1)}$ converges to a Gaussian random variable $N(\mathbf{0}_{nd_{l+1}}, I_{d_{l+1}} \otimes \Sigma^{(l+1)})$ in distribution as $d_l \to \infty$. Recalling (5) and (7), we conclude that $\text{vec}(H^{(l+1)})$ converges to a Gaussian random variable $N(\mathbf{0}_{nd_{l+1}}, I_{d_{l+1}} \otimes \Sigma^{(l+1)})$ as $d_1, \ldots, d_l$ sequentially go to infinity.

**Conclusion.** By the principle of mathematical induction, we have proven this proposition. □

## C.2 Some discussion w.r.t. $G$

We claim that the function $G$ is well-defined in Proposition C.1 on the collection of positive semi-definite matrices

$$\mathcal{S} = \{\Sigma \in \mathbb{R}^{n \times n} : x^T \Sigma x \geq 0 \text{ for all } x \in \mathbb{R}^{n \times 1}\}. \tag{8}$$

*Remark* C.2. To show that $G(\Sigma) = \mathbb{E}_{h \sim N(\mathbf{0}_n, \Sigma)}[\sigma(h)\sigma(h)^T]$ is well-defined at any $\Sigma \in \mathcal{S}$, we only need to show that such $\Sigma$ is always a feasible covariance matrix of Gaussian distribution. For any $\Sigma \in \mathcal{S}$, there exists $P \in \mathbb{R}^{n \times n}$, such that $PP^T = \Sigma$. Let $\xi \sim N(\mathbf{0}_n, I_n)$ be an $n$-dimensional standard normal random variable, then the random variable $P\xi \sim N(\mathbf{0}_n, \Sigma)$. Thus, all positive semi-definite matrices are feasible covariance matrices for Gaussian distributions.

**Definition C.3.** Given any positive semi-definite matrix $\Sigma \in \mathcal{S}$, we define

$$G_1(\Sigma) := q(\Sigma)q(\Sigma)^T, \tag{9}$$

where $q(\Sigma) \in \mathbb{R}^{n \times 1}$ is defined by

$$q(\Sigma)_i := \sqrt{G(\Sigma)_{ii}}, \quad \text{for all } i \in [n]. \tag{10}$$

**Lemma C.4.** *Given any positive semi-definite matrix $\Sigma \in \mathcal{S}$, it holds that*

$$G_1(\Sigma)_{ij} \geq G(\Sigma)_{ij} \quad \text{for any } i, j \in [n]. \tag{11}$$

*Proof.* Recalling the formation of function $G$ in Proposition C.1 (NNGP correspondence for vanilla GCN), for any $i, j \in [n]$, we have

$$G(\Sigma)_{ij} = \mathbb{E}_{h \sim N(\mathbf{0}_n, \Sigma)}[\sigma(h_i) \cdot \sigma(h_j)].$$

Recalling (9) and (10) in Definition C.3, we get

$$\begin{aligned} G_1(\Sigma)_{ij} &:= q(\Sigma)_i \cdot q(\Sigma)_j = \sqrt{G(\Sigma)_{ii}} \cdot \sqrt{G(\Sigma)_{jj}} \\ &= \mathbb{E}_{h \sim N(\mathbf{0}_n, \Sigma)}[\sigma(h_i)^2]^{\frac{1}{2}} \cdot \mathbb{E}_{h \sim N(\mathbf{0}_n, \Sigma)}[\sigma(h_j)^2]^{\frac{1}{2}} \end{aligned} \tag{12}$$

From Hölder's inequality (Hardy et al., 1952), we get

$$\begin{aligned} \text{RHS of (12)} &\geq \mathbb{E}_{h \sim N(\mathbf{0}_n, \Sigma)}\left[|\sigma(h_i) \cdot \sigma(h_j)|\right] \\ &\geq \mathbb{E}_{h \sim N(\mathbf{0}_n, \Sigma)}\left[\sigma(h_i) \cdot \sigma(h_j)\right] = G(\Sigma)_{ij}. \end{aligned}$$

$\square$

**Lemma C.5.** *Given the NNGP covariance matrices $\{\Sigma^{(l)}\}_{l=1}^{\infty}$ defined by (3), it holds that*

$$\text{tr}(\Sigma^{(l+1)}) \leq \sigma_w^2 \, \text{tr}(G(\Sigma^{(l)})).$$

*Proof.* Recalling the NNGP correspondence formula for vanilla GCN (3) in Proposition C.1, we have

$$\text{tr}(\Sigma^{(l+1)}) = \text{tr}(\sigma_w^2(\hat{A}G(\Sigma^{(l)})\hat{A})) = \sigma_w^2 \, \text{tr}(\hat{A}G(\Sigma^{(l)})\hat{A}). \tag{13}$$

Since all entries of $\hat{A}$ are non-negative, by Lemma C.4, we have

$$(\hat{A}G(\Sigma^{(l)})\hat{A})_{ii} \leq (\hat{A}G_1(\Sigma^{(l)})\hat{A})_{ii}, \quad \text{for any } i \in [n].$$

Taking the summation of w.r.t $i \in [n]$, we get

$$\text{tr}(\hat{A}G(\Sigma^{(l)})\hat{A}) \leq \text{tr}(\hat{A}G_1(\Sigma^{(l)})\hat{A}). \tag{14}$$

Recalling the definition of function $G_1$ in (9), we get

$$\text{tr}(\hat{A}G_1(\Sigma^{(l)})\hat{A}) = \text{tr}(\hat{A}q(\Sigma^{(l)})q(\Sigma^{(l)})^T\hat{A}) = \|\hat{A}q(\Sigma^{(l)})\|^2. \tag{15}$$

By Proposition B.1, all the eigenvalues of $\hat{A}$ belong to $(-1, 1]$. Recalling the definition of function $q$ in (10), we get

$$\|\hat{A}q(\Sigma^{(l)})\|^2 \leq \|q(\Sigma^{(l)})\|^2 = \sum_{i=1}^{n} q(\Sigma^{(l)})_i^2 = \text{tr}(G(\Sigma^{(l)})). \tag{16}$$

Finally, combining (13), (14), (15), and (16), we complete the proof. $\square$

## C.3  Proof of Theorem 3.1 (Signal propagation on ReLU-like-activated vanilla GCN)

To facilitate reading and understanding, we restate Theorem 3.1 here.

**Theorem 3.1.** *Under the NNGP correspondence approximation, when the activation function $\sigma$ is ReLU, we have*

*1. If $\sigma_w^2 = 2$, either the limit graph embedding variation (GEV) metric $\lim_{L\to\infty} \mathbf{M}_{GEV}^{(L)}(\sigma_w^2) = 0$ or the limit forward signal propagation (FSP) metric $\lim_{L\to\infty} \mathbf{M}_{FSP}^{(L)}(\sigma_w^2) = 0$;*

*2. When $\sigma_w^2 < 2$, the forward signal propagation (FSP) metric $\mathbf{M}_{FSP}^{(L)}(\sigma_w^2) \leq \frac{2C}{d_0} \cdot (\sigma_w^2/2)^L$ for any $L \geq 1$.*

We now provide a more general signal propagation analysis on vanilla GCN with ReLU-like activation.

**Definition C.6** (ReLU-like activation)**.** An activation function $\sigma : \mathbb{R} \to \mathbb{R}$ is $(\alpha, \beta)$-ReLU if it has the form

$$\sigma(x) = \begin{cases} \alpha x, & x \geq 0, \\ \beta x, & x < 0, \end{cases} \tag{17}$$

where $\alpha, \beta \in \mathbb{R}_+$ and not both of them are 0. We also call such $\sigma$ a ReLU-like activation function.

Next, to prove Theorem 3.1, we generalize our analysis from the standard ReLU activation to the more general $(\alpha, \beta)$-ReLU activation in Definition C.6. Since $(\alpha, \beta) = (1, 0)$ recovers the standard ReLU, proving the following Theorem C.7 implies Theorem 3.1.

**Theorem C.7** (The generalized version of Theorem 3.1)**.** *Under the NNGP correspondence approximation, when the activation function $\sigma$ is $(\alpha, \beta)$-ReLU in Definition C.6, we have*

*1. When $\sigma_w^2 = 2/(\alpha^2 + \beta^2)$, either the graph embedding variation metric*

$$\lim_{L\to\infty} \mathbf{M}_{GEV}^{(L)}(\sigma_w^2) = \lim_{L\to\infty} \mathbb{E}_{H\sim N(\mathbf{0}_n, \Sigma^{(L)})} \left[ \mathrm{Dir}(H)/\|H\|_{\mathrm{F}}^2 \right] = 0,$$

*or the forward propagation metric*

$$\lim_{L\to\infty} \mathbf{M}_{FSP}^{(L)}(\sigma_w^2) = \lim_{L\to\infty} \mathbb{E}_{H\sim N(\mathbf{0}_n, \Sigma^{(L)})} [\|H\|_{\mathrm{F}}^2/\|X\|_{\mathrm{F}}^2] = 0.$$

*2. When $\sigma_w^2 < 2/(\alpha^2 + \beta^2)$, for any $L \geq 1$, the forward propagation metric satisfies*

$$\mathbf{M}_{FSP}^{(L)}(\sigma_w^2) = \mathbb{E}_{H\sim N(\mathbf{0}_n, \Sigma^{(L)})} [\|H\|_{\mathrm{F}}^2/\|X\|_{\mathrm{F}}^2] \leq \frac{2C}{(\alpha^2 + \beta^2)d_0} \cdot \left( \frac{\sigma_w^2(\alpha^2 + \beta^2)}{2} \right)^L.$$

**Lemma C.8.** *For any $x \in \mathbb{R}^n$, it holds that*

$$\mathrm{Dir}(\hat{A}x) \leq \lambda^2 \mathrm{Dir}(x), \tag{18}$$

*where $\lambda$ is the second largest absolute eigenvalue of $\hat{A}$, i.e.,*

$$\lambda = \max_{i\in[n], \lambda_i \neq 1} |\lambda_i|.$$

*Proof.* Since $\hat{A}$ is a symmetric real matrix, by Proposition B.1, it can be decomposed as $\hat{A} = U\Lambda U^T$, where $\Lambda = \mathrm{diag}(\lambda_1, \lambda_2, \ldots, \lambda_n)$ and $U \in \mathbb{R}^{n\times n}$ is an orthogonal matrix. The $i$-th column $u_i$ of $U$ is the eigenvector corresponding to $\lambda_i$.

By Proposition B.1, we have $\lambda_i \in (-1, 1]$ for all $i \in [n]$. Since $\hat{L} = I - \hat{A}$, we conclude that

$$\mathrm{Dir}(\hat{A}x) = (\hat{A}x)^T \hat{L} \hat{A}x = x^T \hat{A}\hat{L}\hat{A}x = z^T U^T (U\Lambda U^{-1})(U(I - \Lambda)U^{-1})(U\Lambda U^{-1})z$$

$$= z^T \Lambda(I - \Lambda)\Lambda z = \sum_{i=1}^{n}(1 - \lambda_i)\lambda_i^2 z_i^2 \leq \lambda^2 \sum_{i=1}^{n}(1 - \lambda_i)z_i^2$$

$$= \lambda^2 z^T (I - \Lambda)z = \lambda^2 \mathrm{Dir}(x).$$

$\square$

**Lemma C.9.** *When the activation function $\sigma$ is $(\alpha, \beta)$-ReLU, it holds that*

$$(\sigma(x) - \sigma(y))^2 + (\sigma(-x) - \sigma(-y))^2 \leq (\alpha^2 + \beta^2)(x - y)^2, \tag{19}$$

*for any $x, y \in \mathbb{R}$. Moreover, the inequality becomes an equality if and only if $xy \geq 0$.*

*Proof.* When $x, y \geq 0$, it holds that

$$\text{LHS of (19)} = (\alpha x - \alpha y)^2 + (-\beta x + \beta y)^2 = \text{RHS of (19)}.$$

Similarly, the equality holds when $x, y \leq 0$. When $xy < 0$,

$$\begin{aligned}
\text{LHS of (19)} &= (\alpha x - \beta y)^2 + (-\beta x + \alpha y)^2 \\
&= (\alpha^2 + \beta^2)(x^2 + y^2) - 4\alpha\beta xy \\
&= (\alpha^2 + \beta^2)(x - y)^2 + 2(\alpha - \beta)^2 xy \\
&< \text{RHS of (19)}.
\end{aligned}$$

$\square$

**Lemma C.10.** *When the activation function $\sigma$ is $(\alpha, \beta)$-ReLU, it holds that*

$$\text{Dir}(\sigma(h)) + \text{Dir}(\sigma(-h)) \leq (\alpha^2 + \beta^2)\text{Dir}(h). \tag{20}$$

*Proof.* Since the activation function $\sigma$ is $(\alpha, \beta)$-ReLU, we have

$$\sigma(cx) = c\sigma(x), \quad \text{for any } c \in \mathbb{R}_+, x \in \mathbb{R}.$$

Then we get

$$\begin{aligned}
\text{LHS of (20)} &= \sum_{(i,j)\in\mathcal{E}} \left[\frac{\sigma(h_i)}{\sqrt{1+d_i}} - \frac{\sigma(h_j)}{\sqrt{1+d_j}}\right]^2 + \left[\frac{\sigma(-h_i)}{\sqrt{1+d_i}} - \frac{\sigma(-h_j)}{\sqrt{1+d_j}}\right]^2 \\
&= \sum_{(i,j)\in\mathcal{E}} \left[\sigma\left(\frac{h_i}{\sqrt{1+d_i}}\right) - \sigma\left(\frac{h_j}{\sqrt{1+d_j}}\right)\right]^2 + \left[\sigma\left(\frac{-h_i}{\sqrt{1+d_i}}\right) - \sigma\left(\frac{-h_j}{\sqrt{1+d_j}}\right)\right]^2.
\end{aligned}$$

By Lemma C.9, we have

$$\text{LHS of (20)} \leq (\alpha^2 + \beta^2) \sum_{(i,j)\in\mathcal{E}} \left[\frac{h_i}{\sqrt{1+d_i}} - \frac{h_j}{\sqrt{1+d_j}}\right]^2 = \text{RHS of (20)}.$$

$\square$

**Lemma C.11.** *When the activation function $\sigma$ is $(\alpha, \beta)$-ReLU, for any feasible covariance matrix $\Sigma \in \mathbb{R}^{n \times n}$, it holds that*

$$\mathbb{E}_{h \sim N(\mathbf{0}_n, \Sigma)}[\text{Dir}(\sigma(h))] \leq \frac{\alpha^2 + \beta^2}{2} \cdot \mathbb{E}_{h \sim N(\mathbf{0}_n, \Sigma)}[\text{Dir}(h)].$$

*Proof.* By symmetry, for any $n$-dimensional random variable $h \sim N(\mathbf{0}_n, \Sigma)$, it holds that $-h \sim N(\mathbf{0}_n, \Sigma)$. By Lemma C.10, we have

$$\begin{aligned}
2\mathbb{E}_{h \sim N(\mathbf{0}_n, \Sigma)}[\text{Dir}(\sigma(h))] &= \mathbb{E}_{h \sim N(\mathbf{0}_n, \Sigma)}[\text{Dir}(\sigma(h)) + \text{Dir}(\sigma(-h))] \\
&\leq (\alpha^2 + \beta^2)\mathbb{E}_{h \sim N(\mathbf{0}_n, \Sigma)}[\text{Dir}(h)].
\end{aligned}$$

$\square$

**Lemma C.12.** *Under the NNGP correspondence approximation, suppose that the activation function $\sigma$ is $(\alpha, \beta)$-ReLU in Definition C.6. If*

$$\sigma_w^2 < \frac{2}{\lambda^2(\alpha^2 + \beta^2)},$$

*then we have*

$$\mathbb{E}_{h \sim N(\mathbf{0}_n, \Sigma^{(l)})}[\mathrm{Dir}(h)] = O\left(\left(\frac{\lambda^2 \sigma_w^2(\alpha^2 + \beta^2)}{2}\right)^l\right), \quad as \ l \to \infty,$$

*where $\lambda$ is the second largest non-one absolute eigenvalue of $\hat{A}$, i.e.,*

$$\lambda = \max_{i \in [n], \lambda_i \neq 1} |\lambda_i|.$$

*Proof.* For any positive semi-definite matrix $\Sigma \in \mathcal{S}$ and any $n$-dimensional Gaussian random variable $h \sim N(\mathbf{0}_n, \Sigma)$, we have

$$\mathbb{E}_{h \sim N(\mathbf{0}_n, \Sigma)}[\mathrm{Dir}(h)] = \mathbb{E}_{h \sim N(\mathbf{0}_n, \Sigma)}[\mathrm{tr}(h^T \hat{L} h)] = \mathbb{E}_{h \sim N(\mathbf{0}_n, \Sigma)}[\mathrm{tr}(\hat{L} h h^T)] = \mathrm{tr}(\hat{L}\Sigma).$$

Then according the NNGP correspondence formula (3) in Proposition C.1, for any $l \in \mathbb{N}$, we have

$$
\begin{aligned}
&\mathbb{E}_{h \sim N(\mathbf{0}_n, \Sigma^{(l+1)})}[\mathrm{Dir}(h)] = \mathrm{tr}(\hat{L}\Sigma^{(l+1)}) \\
&= \sigma_w^2 \, \mathrm{tr}(\hat{L}\hat{A}G(\Sigma^{(l)})\hat{A}) = \sigma_w^2 \, \mathrm{tr}\left(\hat{L}\hat{A} \cdot \mathbb{E}_{h \sim N(\mathbf{0}_n, \Sigma^{(l)})}[\sigma(h)\sigma(h)^T] \cdot \hat{A}\right) \\
&= \sigma_w^2 \mathbb{E}_{h \sim N(\mathbf{0}_n, \Sigma^{(l)})}\left[\mathrm{tr}\left(\hat{L}\hat{A}\sigma(h)\sigma(h)^T\hat{A}\right)\right] = \sigma_w^2 \mathbb{E}_{h \sim N(\mathbf{0}_n, \Sigma^{(l)})}\left[\mathrm{tr}\left(\sigma(h)^T\hat{A}\hat{L}\hat{A}\sigma(h)\right)\right] \\
&= \sigma_w^2 \mathbb{E}_{h \sim N(\mathbf{0}_n, \Sigma^{(l)})}\left[\mathrm{Dir}\left(\hat{A}\sigma(h)\right)\right].
\end{aligned}
\tag{21}
$$

By Lemma C.8 and Lemma C.11, we get

$$\text{RHS of } (21) \leq \lambda^2 \sigma_w^2 \cdot \mathbb{E}_{h \sim N(\mathbf{0}_n, \Sigma^{(l)})}[\mathrm{Dir}(\sigma(h))] \leq \frac{\lambda^2 \sigma_w^2(\alpha^2 + \beta^2)}{2} \cdot \mathbb{E}_{h \sim N(\mathbf{0}_n, \Sigma^{(l)})}[\mathrm{Dir}(h)]. \tag{22}$$

Thus, combining (21) and (22), by induction, we have

$$\mathbb{E}_{h \sim N(\mathbf{0}_n, \Sigma^{(l)})}[\mathrm{Dir}(h)] = O\left(\left(\frac{\lambda^2 \sigma_w^2(\alpha^2 + \beta^2)}{2}\right)^l\right), \quad as \ l \to \infty.$$

$\square$

*Proof of Theorem C.7 (the generalized version of Theorem 3.1).* First of all, we will prove **part 2** of this theorem. For any positive semi-definite matrix $\Sigma \in \mathcal{S}$, we have

$$\mathbb{E}_{h \sim N(\mathbf{0}_n, \Sigma)}[\|h\|^2] = \mathbb{E}_{h \sim N(\mathbf{0}_n, \Sigma)}[\mathrm{tr}(h^T h)] = \mathbb{E}_{h \sim N(\mathbf{0}_n, \Sigma)}[\mathrm{tr}(hh^T)] = \mathrm{tr}(\Sigma).$$

For this reason, we only need to focus on $\{\mathrm{tr}(\Sigma^{(l)})\}_{l=1}^\infty$ in the following proof.

We will show that $\{\mathrm{tr}(\Sigma^{(l)})\}_{l=1}^\infty$ is a decreasing sequence if $\sigma_w \leq 2/(\alpha^2 + \beta^2)$. By Lemma C.5, we have

$$\mathrm{tr}(\Sigma^{(l+1)}) \leq \sigma_w^2 \, \mathrm{tr}(G(\Sigma^{(l)})). \tag{23}$$

When the activation function $\sigma$ is $(\alpha, \beta)$-ReLU, for any $c \in \mathbb{R}_+$, it holds that

$$
\begin{aligned}
\mathbb{E}_{Z \sim N(0,1)}[\sigma(cZ)^2] &= \mathbb{E}_{Z \sim N(0,1)}[\alpha^2 c^2 Z^2 \mathbb{1}_{\{Z > 0\}}] + \mathbb{E}_{Z \sim N(0,1)}[\beta^2 c^2 Z^2 \mathbb{1}_{\{Z \leq 0\}}] \\
&= \frac{\alpha^2 + \beta^2}{2} \cdot \mathbb{E}_{Z \sim N(0,1)}[c^2 Z^2].
\end{aligned}
$$

Accordingly, for any positive semi-definite matrix $\Sigma \in \mathcal{S}$ and $i \in [n]$, we have

$$
\begin{aligned}
G(\Sigma)_{ii} &= \mathbb{E}_{h \sim N(\mathbf{0}_n, \Sigma)}[\sigma(h_i)^2] = \mathbb{E}_{Z \sim N(0,1)}\left[\sigma(\sqrt{\Sigma_{ii}}Z)^2\right] \\
&= \frac{\alpha^2 + \beta^2}{2} \cdot \mathbb{E}_{Z \sim N(0,1)}\left[\Sigma_{ii}Z^2\right] = \frac{\alpha^2 + \beta^2}{2} \cdot \Sigma_{ii}.
\end{aligned}
\tag{24}
$$

Combining (23) and (24), we get

$$
\operatorname{tr}(\Sigma^{(l+1)}) \leq \frac{\sigma_w^2(\alpha^2 + \beta^2)}{2} \operatorname{tr}(\Sigma^{(l)}).
$$

Thus, we have shown that $\{\operatorname{tr}(\Sigma^{(l)})\}_{l=1}^{\infty}$ is a decreasing sequence if $\sigma_w \leq 2/(\alpha^2 + \beta^2)$. In addition, if $\sigma_w < 2/(\alpha^2 + \beta^2)$, we get

$$
\operatorname{tr}(\Sigma^{(L)}) \leq \left(\frac{\sigma_w^2(\alpha^2 + \beta^2)}{2}\right)^{L-1} \operatorname{tr}(\Sigma^{(1)}).
\tag{25}
$$

By Proposition C.1, we have

$$
\operatorname{tr}(\Sigma^{(1)}) = \frac{\sigma_w^2}{d_0} \operatorname{tr}(\hat{A}XX^T\hat{A}) = \frac{\sigma_w^2}{d_0} \sum_{k=1}^{d_0} \operatorname{tr}(\hat{A}X_{:,k}X_{:,k}^T\hat{A}) = \frac{\sigma_w^2}{d_0} \sum_{k=1}^{d_0} \|\hat{A}X_{:,k}\|^2
\tag{26}
$$

Since all the eigenvalues of $\hat{A}$ belong to $(-1, 1]$ by Propositon B.1, we get

$$
\text{RHS of (26)} \leq \frac{\sigma_w^2}{d_0} \sum_{k=1}^{d_0} \|X_{:,k}\|^2 = \frac{\sigma_w^2}{d_0} \operatorname{tr}(XX^T).
\tag{27}
$$

Combining (25), (26), and (27), we have

$$
\operatorname{tr}(\Sigma^{(L)}) \leq \frac{\sigma_w^2}{d_0} \cdot \left(\frac{\sigma_w^2(\alpha^2 + \beta^2)}{2}\right)^{L-1} \operatorname{tr}(XX^T).
$$

Thus, the forward propagation metric at the $L$-th layer satisfies

$$
\begin{aligned}
\mathbb{E}_{H \sim N(\mathbf{0}_n, \Sigma^{(L)})}\left[\frac{\|H\|_{\mathrm{F}}^2}{\|X\|_{\mathrm{F}}^2}\right] &= \frac{C}{\operatorname{tr}(XX^T)} \cdot \mathbb{E}_{h \sim N(\mathbf{0}_n, \Sigma^{(L)})}[\|h\|^2] = \frac{C}{\operatorname{tr}(XX^T)} \operatorname{tr}(\Sigma^{(L)}) \\
&\leq \frac{C\sigma_w^2}{d_0} \cdot \left(\frac{\sigma_w^2(\alpha^2 + \beta^2)}{2}\right)^{L-1} \\
&= \frac{2C}{(\alpha^2 + \beta^2)d_0} \cdot \left(\frac{\sigma_w^2(\alpha^2 + \beta^2)}{2}\right)^{L}.
\end{aligned}
$$

Then we have completed part 2 of this theorem. If $\sigma$ is ReLU activation function, i.e., $(1,0)$-ReLU. If $\sigma < 2 = \frac{2}{1^2 + 0^2}$, we have

$$
\mathbb{E}_{H \sim N(\mathbf{0}_n, \Sigma^{(L)})}\left[\frac{\|H\|_{\mathrm{F}}^2}{\|X\|_{\mathrm{F}}^2}\right] = \frac{2C}{(1^2 + 0^2)d_0} \cdot \left(\frac{\sigma_w^2(1^2 + 0^2)}{2}\right)^{L} = \frac{2C}{d_0} \cdot \left(\frac{\sigma_w^2}{2}\right)^{L},
$$

which coincides with **part 2 in Theorem 3.1**.

Next, we will prove **part 1** of this theorem. Let's study the case when $\sigma_w^2 = 2/(\alpha^2 + \beta^2)$. Suppose that

$$
\lim_{l \to \infty} \operatorname{tr}(\Sigma^{(l)}) = \delta_0.
$$

If $\delta_0 = 0$, then we have completed the first part of this theorem by getting

$$
\begin{aligned}
\lim_{L \to \infty} \mathbb{E}_{H \sim N(\mathbf{0}_n, \Sigma^{(L)})}[\|H\|_{\mathrm{F}}^2/\|X\|_{\mathrm{F}}^2] &= \lim_{L \to \infty} \frac{C}{\|X\|_{\mathrm{F}}^2} \cdot \mathbb{E}_{h \sim N(\mathbf{0}_n, \Sigma^{(L)})}[\|h\|^2] \\
&= \frac{C}{\|X\|_{\mathrm{F}}^2} \cdot \lim_{L \to \infty} \operatorname{tr}(\Sigma^{(L)}) = 0.
\end{aligned}
$$

Now we study the case when $\delta_0 > 0$. In order to show part 1 of the theorem, we only need to demonstrate that

$$\lim_{L\to\infty} \mathbb{E}_{H\sim N(\mathbf{0}_n,\Sigma^{(L)})}\left[\frac{\mathrm{Dir}(H)}{\|H\|_{\mathrm{F}}^2}\right] = 0.$$

Given any fixed $\epsilon > 0$, we have

$$\begin{aligned}
&\mathbb{E}_{H\sim N(\mathbf{0}_n,\Sigma^{(L)})}\left[\frac{\mathrm{Dir}(H)}{\|H\|_{\mathrm{F}}^2}\right]\\
&= \mathbb{E}_{H\sim N(\mathbf{0}_n,\Sigma^{(L)})}\left[\frac{\mathrm{Dir}(H)}{\|H\|_{\mathrm{F}}^2}\mathbb{1}_{\{\|H\|_{\mathrm{F}}\geq\epsilon\}}\right] + \mathbb{E}_{H\sim N(\mathbf{0}_n,\Sigma^{(L)})}\left[\frac{\mathrm{Dir}(H)}{\|H\|_{\mathrm{F}}^2}\mathbb{1}_{\{\|H\|_{\mathrm{F}}\leq\epsilon\}}\right].
\end{aligned} \tag{28}$$

From Lemma B.2, it holds that $\mathrm{Dir}(H)/\|H\|_{\mathrm{F}}^2 \leq 2$, so we get

$$\begin{aligned}
\text{RHS of (28)} &\leq \frac{1}{\epsilon^2}\cdot\mathbb{E}_{H\sim N(\mathbf{0}_n,\Sigma^{(L)})}\left[\mathrm{Dir}(H)\mathbb{1}_{\{\|H\|_{\mathrm{F}}\geq\epsilon\}}\right] + 2\cdot\mathbb{P}_{H\sim N(\mathbf{0}_n,\Sigma^{(L)})}\left[\|H\|_{\mathrm{F}}\leq\epsilon\right]\\
&\leq \frac{1}{\epsilon^2}\cdot\mathbb{E}_{H\sim N(\mathbf{0}_n,\Sigma^{(L)})}\left[\mathrm{Dir}(H)\right] + 2\cdot\mathbb{P}_{H\sim N(\mathbf{0}_n,\Sigma^{(L)})}\left[\|H\|_{\mathrm{F}}\leq\epsilon\right].
\end{aligned} \tag{29}$$

For any $L \geq 1$, there exists $i \in [n]$, such that $\Sigma_{ii}^{(L)} \geq \mathrm{tr}(\Sigma^{(L)})/n$. Then for any $n \times C$ random matrix $H \sim N(\mathbf{0}_n, \Sigma^{(L)})$, we have $H_{i,1} \sim N(0, \Sigma_{ii}^{(L)})$. For this reason, we have

$$\begin{aligned}
\mathbb{P}_{H\sim N(\mathbf{0}_n,\Sigma^{(L)})}\left[\|H\|_{\mathrm{F}}\leq\epsilon\right] &\leq \mathbb{P}_{H\sim N(\mathbf{0}_n,\Sigma^{(L)})}\left[|H_{i,1}|\leq\epsilon\right] = \mathbb{P}_{Z\sim N(0,1)}\left[|Z|\leq\frac{\epsilon}{\sqrt{\Sigma_{ii}}}\right]\\
&\leq \mathbb{P}_{Z\sim N(0,1)}\left[|Z|\leq\epsilon\cdot\sqrt{\frac{n}{\mathrm{tr}(\Sigma^{(L)})}}\right]\\
&= 2\Phi\left(\epsilon\cdot\sqrt{\frac{n}{\mathrm{tr}(\Sigma^{(L)})}}\right) - 1,
\end{aligned} \tag{30}$$

where $\Phi(x) = \mathbb{P}_{Z\sim N(0,1)}[Z \leq x]$ denotes the cumulative distribution function of the standard normal distribution $N(0,1)$.

Combining (28), (29), and (30), we get

$$\mathbb{E}_{H\sim N(\mathbf{0}_n,\Sigma^{(L)})}\left[\frac{\mathrm{Dir}(H)}{\|H\|_{\mathrm{F}}^2}\right] \leq \frac{1}{\epsilon^2}\cdot\mathbb{E}_{H\sim N(\mathbf{0}_n,\Sigma^{(L)})}\left[\mathrm{Dir}(H)\right] + 4\Phi\left(\epsilon\cdot\sqrt{\frac{n}{\mathrm{tr}(\Sigma^{(L)})}}\right) - 2,$$

for any $L \geq 1$.

Since

$$\sigma_w^2 = \frac{2}{\alpha^2 + \beta^2} < \frac{2}{\lambda^2(\alpha^2 + \beta^2)},$$

by Lemma C.12, we have

$$\lim_{L\to\infty} \mathbb{E}_{H\sim N(\mathbf{0}_n,\Sigma^{(L)})}[\mathrm{Dir}(H)] = 0.$$

We let $L \to \infty$ in (28) and get

$$\begin{aligned}
&\limsup_{L\to\infty}\mathbb{E}_{H\sim N(\mathbf{0}_n,\Sigma^{(L)})}\left[\frac{\mathrm{Dir}(H)}{\|H\|_{\mathrm{F}}^2}\right]\\
&\leq \frac{1}{\epsilon^2}\cdot\limsup_{L\to\infty}\mathbb{E}_{H\sim N(\mathbf{0}_n,\Sigma^{(L)})}\left[\mathrm{Dir}(H)\right] + 4\cdot\limsup_{L\to\infty}\Phi\left(\epsilon\cdot\sqrt{\frac{n}{\mathrm{tr}(\Sigma^{(L)})}}\right) - 2\\
&= \frac{1}{\epsilon^2}\cdot 0 + 4\Phi\left(\epsilon\cdot\sqrt{\frac{n}{\delta_0}}\right) - 2 = 4\Phi\left(\epsilon\cdot\sqrt{\frac{n}{\delta_0}}\right) - 2.
\end{aligned} \tag{31}$$

Notice that the left hand side of (31) is independent of the choice of $\epsilon$. Since $\Phi$ is a continuous map, we let $\epsilon \to 0^+$ and get

$$\limsup_{L \to \infty} \mathbb{E}_{H \sim N(\mathbf{0}_n, \Sigma^{(L)})} \left[ \frac{\text{Dir}(H)}{\|H\|_F^2} \right] \le 4\Phi(0) - 2 = 0.$$

Therefore, we have

$$\lim_{L \to \infty} \mathbb{E}_{H \sim N(\mathbf{0}_n, \Sigma^{(L)})} \left[ \frac{\text{Dir}(H)}{\|H\|_F^2} \right] = 0.$$

$\square$

### C.4 Proof of Theorem 3.2 (Signal propagation on ReLU-activated vanilla GCN)

To facilitate reading and understanding, we restate Theorem 3.2 here.

**Theorem 3.2.** *Under the NNGP correspondence approximation, when the activation is ReLU, the graph embedding variation (GEV) metric $\mathbf{M}_{GEV}^{(L)}$ is independent of $\sigma_w^2$.*

Similar to Appendix C.3, to prove Theorem 3.2, we generalize our analysis from the standard ReLU activation to the more general $(\alpha, \beta)$-ReLU activation in Definition C.6. Since $(\alpha, \beta) = (1, 0)$ recovers the standard ReLU, proving the following Theorem C.13 implies Theorem 3.2.

**Theorem C.13** (The generalized version of Theorem 3.2)**.** *Under the NNGP correspondence approximation, when the activation is $(\alpha, \beta)$-ReLU, the graph embedding variation metric $\mathbf{M}_{GEV}^{(L)}(\sigma_w^2) = \mathbb{E}_{H \sim N(\mathbf{0}_n, \Sigma^{(L)})}[\text{Dir}(H)/\|H\|_F^2]$ is independent of the choice of $\sigma_w^2$.*

*Proof of Theorem C.13 (the generalized version of Theorem 3.2).* Under the NNGP correspondence approximation, we only need to prove that

$$\frac{\Sigma^{(l)}(\sigma_w^2)}{\sigma_w^{2l}} = \frac{\Sigma^{(l)}(\tilde{\sigma}_w^2)}{\tilde{\sigma}_w^{2l}}, \quad \text{for any } l \ge 1 \text{ and } \sigma_w^2, \tilde{\sigma}_w^2 > 0. \tag{32}$$

If (32) holds, then $H \sim N(\mathbf{0}_n, \Sigma^{(L)}(\sigma_w^2))$ implies $\tilde{\sigma}_w^L H/\sigma_w^L \sim N(\mathbf{0}_n, \Sigma^{(L)}(\tilde{\sigma}_w^2))$. In this way, we have

$$\mathbb{E}_{H \sim N(\mathbf{0}_n, \Sigma^{(L)}(\sigma_w^2))} \left[ \frac{\text{Dir}(H)}{\|H\|_F^2} \right] = \mathbb{E}_{H \sim N(\mathbf{0}_n, \Sigma^{(L)}(\sigma_w^2))} \left[ \frac{\text{Dir}(\tilde{\sigma}_w^L H/\sigma_w^L)}{\|\tilde{\sigma}_w^L H/\sigma_w^L\|_F^2} \right]$$
$$= \mathbb{E}_{H \sim N(\mathbf{0}_n, \Sigma^{(L)}(\tilde{\sigma}_w^2))} \left[ \frac{\text{Dir}(H)}{\|H\|_F^2} \right].$$

Now we prove (32) by mathematical induction. When $l = 1$, by Proposition C.1, we have

$$\frac{\Sigma^{(1)}(\sigma_w^2)}{\sigma_w^2} = \frac{1}{d_0} \hat{A} X X^T \hat{A} = \frac{\Sigma^{(1)}(\tilde{\sigma}_w^2)}{\tilde{\sigma}_w^2}, \quad \text{for any } \sigma_w^2, \tilde{\sigma}_w^2 > 0.$$

If (32) holds for $L$, we look at the case for $L + 1$. Since the activation $\sigma$ is $(\alpha, \beta)$-ReLU, for any $c \in \mathbb{R}_+$, we have $\sigma(cx) = c\sigma(x)$. Recalling the definition of $G$ in Proposition C.1, for any positive semi-definite matrix $\Sigma \in \mathcal{S}$, we have

$$G(c^2\Sigma)_{ij} = \mathbb{E}_{h \sim N(\mathbf{0}_n, c^2\Sigma)}[\sigma(h_i) \cdot \sigma(h_j)] = \mathbb{E}_{h \sim N(\mathbf{0}_n, \Sigma)}[\sigma(ch_i) \cdot \sigma(ch_j)]$$
$$= c^2 \mathbb{E}_{h \sim N(\mathbf{0}_n, \Sigma)}[\sigma(h_i) \cdot \sigma(h_j)] = c^2 G(\Sigma)_{ij},$$

for any $i, j \in [n]$ and $c \in \mathbb{R}_+$. Thus, by Proposition C.1, we have

$$
\begin{aligned}
\left(\frac{\tilde{\sigma}_w^2}{\sigma_w^2}\right)^{L+1} \cdot \Sigma^{(L+1)}(\sigma_w^2) &\overset{(a)}{=} \left(\frac{\tilde{\sigma}_w^2}{\sigma_w^2}\right)^{L+1} \cdot \sigma_w^2 \hat{A} G\left(\Sigma^{(L)}(\sigma_w^2)\right) \hat{A} \\
&= \tilde{\sigma}_w^2 \cdot \left(\frac{\tilde{\sigma}_w^2}{\sigma_w^2}\right)^{L} \cdot \hat{A} G\left(\Sigma^{(L)}(\sigma_w^2)\right) \hat{A} \\
&\overset{(b)}{=} \tilde{\sigma}_w^2 \cdot \hat{A} G\left(\Sigma^{(L)}(\tilde{\sigma}_w^2)\right) \\
&\overset{(c)}{=} \Sigma^{(L+1)}(\tilde{\sigma}_w^2),
\end{aligned}
$$

where $(a)$ and $(c)$ are due to Proposition C.1 and $(b)$ are from the induction hypothesis.

Therefore, (32) holds for all $L \geq 1$ and we have completed the proof. $\qquad\square$

### C.5 Signal propagation on tanh-activated vanilla GCN

**Theorem C.14.** *Under the NNGP correspondence approximation, when the activation function $\sigma$ is tanh, we have*

*1. When $\sigma_w^2 = 1$, we have $\lim_{L \to \infty} \mathbf{M}_{FSP}^{(L)}(\sigma_w^2) = \lim_{L \to \infty} \mathbb{E}_{H \sim N(\mathbf{0}_n, \Sigma^{(L)})}[\|H\|_{\mathrm{F}}^2 / \|X\|_{\mathrm{F}}^2] = 0$.*

*2. When $\sigma_w^2 < 1$, we have $\mathbf{M}_{FSP}^{(L)}(\sigma_w^2) = \mathbb{E}_{H \sim N(\mathbf{0}_n, \Sigma^{(L)})}[\|H\|_{\mathrm{F}}^2 / \|X\|_{\mathrm{F}}^2] \leq \frac{C}{d_0} \cdot \sigma_w^{2L}$ for any $L \geq 1$.*

**Lemma C.15.** *The collection of positive semi-definite matrices $\mathcal{S}$ defined by (8) is a closed subset of $\mathbb{R}^{n \times n}$.*

*Proof.* We only need to show that given any convergent sequence $\{Q^{(k)}\}_{k=1}^{\infty} \subset \mathcal{S}$, its limit also belongs to $\mathcal{S}$. Suppose that

$$
\lim_{k \to \infty} Q^{(k)} = Q^*.
$$

Since all $\{Q^{(k)}\}_{k=1}^{\infty}$ are positive semi-definite matrices, so given any $x \in \mathbb{R}^{n \times 1}$, we have

$$
x^T Q^{(k)} x \geq 0, \quad \text{for all } k \in \mathbb{N}.
$$

Then we get

$$
x^T Q^* x = \lim_{k \to \infty} x^T Q^{(k)} x \geq 0.
$$

Thus, $Q^*$ also belongs to $\mathcal{S}$. $\qquad\square$

**Lemma C.16.** *When the activation function $\sigma$ is tanh, i.e., $\sigma(x) = (e^x - e^{-x})/(e^x + e^{-x})$, then we have $|\sigma(x)| \leq |x|$ for any $x \in \mathbb{R}$. Moreover, the equality holds if and only if $x = 0$.*

*Proof.* It is easy to verify that $\sigma(0) = 0$. Given any $x \geq 0$, we have

$$
\sigma(-x) = \frac{e^{-x} - e^x}{e^{-x} + e^x} = -\frac{e^x - e^{-x}}{e^{-x} + e^x} = -\sigma(x).
$$

For this reason, we only need to prove that $|\sigma(x)| < |x|$ for any $x > 0$. In the following part, we will show that $0 < \sigma(x) < x$ when $x > 0$.

We define $f(x) := \sigma(x) - x$ for any $x \geq 0$. Let's consider the derivative of $f$:

$$
\begin{aligned}
f'(x) &= \frac{d}{dx}\left(\frac{e^x - e^{-x}}{e^x + e^{-x}} - x\right) \\
&= \frac{1}{(e^x + e^{-x})^2}\left[(e^x + e^{-x})\cdot\frac{d}{dx}(e^x - e^{-x}) - (e^x - e^{-x})\cdot\frac{d}{dx}(e^x + e^{-x})\right] - 1 \\
&= \frac{(e^x + e^{-x})^2 - (e^x - e^{-x})^2}{(e^x + e^{-x})^2} - 1 \\
&= \frac{-(e^x - e^{-x})^2}{(e^x + e^{-x})^2}.
\end{aligned}
$$

Then if $x > 0$, we have $f'(x) < 0$; if $x = 0$, we have $f'(x) = 0$. Thus, $f(x) = \sigma(x) - x$ is a strictly decreasing function in $[0, +\infty)$. Since $f(0) = \sigma(0) - 0 = 0$, we have

$$
f(x) = \sigma(x) - x < 0, \quad \text{for any } x > 0.
$$

Since $0 < e^x - e^{-x} < e^x + e^{-x}$ for any $x > 0$, it holds that

$$
\sigma(x) = (e^x - e^{-x})/(e^x + e^{-x}) > 0, \quad \text{for any } x > 0.
$$

Therefore, we get that $0 < \sigma(x) < x$ for any $x > 0$ and have completed the proof of this lemma. $\qquad\square$

Now it is time for Theorem C.14.

*Proof of Theorem C.14.* First of all, we will prove **part 2** of this theorem. For any positive semi-definite matrix $\Sigma \in \mathcal{S}$, we have

$$
\mathbb{E}_{h\sim N(\mathbf{0}_n, \Sigma)}[\|h\|^2] = \mathbb{E}_{h\sim N(\mathbf{0}_n, \Sigma)}[\mathrm{tr}(h^T h)] = \mathbb{E}_{h\sim N(\mathbf{0}_n, \Sigma)}[\mathrm{tr}(hh^T)] = \mathrm{tr}(\Sigma).
$$

For this reason, we only need to focus on $\{\mathrm{tr}(\Sigma^{(l)})\}_{l=1}^{\infty}$ in the following proof.

We will show that $\{\mathrm{tr}(\Sigma^{(l)})\}_{l=1}^{\infty}$ is a decreasing sequence if $\sigma_w \leq 1$. By Lemma C.5, we have

$$
\mathrm{tr}(\Sigma^{(l+1)}) \leq \sigma_w^2\,\mathrm{tr}(G(\Sigma^{(l)})). \tag{33}
$$

By Lemma C.16, we have $|\sigma(x)| \leq |x|$ for any $x \in \mathbb{R}$. Moreover, the equality holds if and only if $x = 0$. For this reason, given any positive semi-definite matrix $\Sigma \in \mathcal{S}$, we have

$$
\begin{aligned}
\mathrm{tr}(G(\Sigma)) &= \sum_{i=1}^{n}\mathbb{E}_{h\sim N(\mathbf{0}_n, \Sigma)}[\sigma(h_i)^2] = \sum_{i=1}^{n}\mathbb{E}_{Z\sim N(0,1)}\left[\sigma(\sqrt{\Sigma_{ii}}Z)^2\right] \\
&\leq \sum_{i=1}^{n}\mathbb{E}_{Z\sim N(0,1)}\left[(\sqrt{\Sigma_{ii}}Z)^2\right] = \sum_{i=1}^{n}\mathbb{E}_{h\sim N(\mathbf{0}_n, \Sigma)}[h_i^2] = \mathrm{tr}(\Sigma),
\end{aligned} \tag{34}
$$

and the inequality becomes an equality if and only if $\sqrt{\Sigma_{ii}}Z = 0$ holds $\mathbb{P}$-a.s. for all $i \in [n]$. Since $Z \sim N(0,1)$ follows a standard normal distribution, it is equivalent to $\Sigma_{ii} = 0$ for all $i \in [n]$, i.e., $\mathrm{tr}(\Sigma) = 0$.

Combining (33) and (34), we get

$$
\mathrm{tr}(\Sigma^{(l+1)}) \leq \sigma_w^2\,\mathrm{tr}(\Sigma^{(l)}).
$$

Thus, we have shown that $\{\mathrm{tr}(\Sigma^{(l)})\}_{l=1}^{\infty}$ is a decreasing sequence if $\sigma_w \leq 1$. In addition, if $\sigma_w < 1$, we get

$$
\mathrm{tr}(\Sigma^{(L)}) \leq \sigma_w^{2(L-1)}\,\mathrm{tr}(\Sigma^{(1)}). \tag{35}
$$

Analogous to the proof of part 2 in Theorem C.7 for ReLU-activated model, by Proposition C.1 and Proposition B.1, we have

$$
\begin{aligned}
\operatorname{tr}(\Sigma^{(1)}) = \frac{\sigma_w^2}{d_0} \operatorname{tr}(\hat{A} X X^T \hat{A}) &= \frac{\sigma_w^2}{d_0} \sum_{k=1}^{d_0} \operatorname{tr}(\hat{A} X_{:,k} X_{:,k}^T \hat{A}) \\
&= \frac{\sigma_w^2}{d_0} \sum_{k=1}^{d_0} \|\hat{A} X_{:,k}\|^2 \le \frac{\sigma_w^2}{d_0} \sum_{k=1}^{d_0} \|X_{:,k}\|^2 = \frac{\sigma_w^2}{d_0} \|X\|_{\mathrm{F}}^2.
\end{aligned}
\tag{36}
$$

Combining (35) and (36), we have

$$
\operatorname{tr}(\Sigma^{(1)}) \le \frac{\sigma_w^{2L}}{d_0} \|X\|_{\mathrm{F}}^2.
$$

Then we have completed part 2 of the theorem by getting

$$
\begin{aligned}
\mathbb{E}_{H \sim N(\mathbf{0}_n, \Sigma^{(L)})} \left[ \frac{\|H\|_{\mathrm{F}}^2}{\|X\|_{\mathrm{F}}^2} \right] &= \frac{C}{\|X\|_{\mathrm{F}}^2} \mathbb{E}_{h \sim N(\mathbf{0}_n, \Sigma^{(L)})}[\|h\|^2] = \frac{C}{\|X\|_{\mathrm{F}}^2} \operatorname{tr}(\Sigma^{(L)}) \\
&\le \frac{C}{\|X\|_{\mathrm{F}}^2} \cdot \frac{\sigma_w^{2L}}{d_0} \cdot \|X\|_{\mathrm{F}}^2 \le \frac{C}{d_0} \cdot \sigma_w^{2L}.
\end{aligned}
$$

Next, we will prove **part 1** of this theorem. Let's study the case when $\sigma_w = 1$.

Since $\Sigma^{(l)}$ is a positive semi-definite matrix for any $l \in \mathbb{N}$, we have

$$
|\Sigma_{ij}^{(l)}|^2 \le \Sigma_{ii}^{(l)} \Sigma_{jj}^{(l)} \le \operatorname{tr}(\Sigma^{(l)})^2 \le \operatorname{tr}(\Sigma^{(1)})^2, \quad \text{for all } i, j \in [n].
$$

Taking the summation of both sides w.r.t. $i$ and $j$, we get

$$
\|\Sigma^{(l)}\|_F^2 = \sum_{i,j=1}^{n} |\Sigma_{ij}^{(l)}|^2 \le n^2 \operatorname{tr}(\Sigma^{(1)})^2 < \infty.
$$

Thus, the matrix sequence $\{\Sigma^{(l)}\}_{l=1}^{\infty}$ lies in

$$
\mathcal{S}' = \mathcal{S} \cap \{\Sigma \in \mathbb{R}^{n \times n} : \|\Sigma\|_F \le n \operatorname{tr}(\Sigma^{(1)})\}.
$$

By Lemma C.15, $\mathcal{S}'$ is a bounded and closed subset, i.e., a compact subset, of $\mathbb{R}^{n \times n}$. By the Bolzano–Weierstrass theorem, there exists a subsequence $\{\Sigma^{(l_k)}\}_{k=1}^{\infty}$ of $\{\Sigma^{(l)}\}_{l=1}^{\infty}$ and $\Sigma^* \in \mathcal{S}'$ such that

$$
\lim_{k \to \infty} \Sigma^{(l_k)} = \Sigma^*.
$$

Recalling (33) and that $\{\operatorname{tr}(\Sigma^{(l)})\}_{l=1}^{\infty}$ is a decreasing sequence, we have

$$
\operatorname{tr}(\Sigma^{(l_{k+1})}) \le \operatorname{tr}(\Sigma^{(l_k+1)}) \le \operatorname{tr}(G(\Sigma^{(l_k)})).
$$

Since $G$ is a continuous function, we let $k \to \infty$ and get

$$
\operatorname{tr}(\Sigma^*) = \lim_{k \to \infty} \operatorname{tr}(\Sigma^{(l_{k+1})}) \le \lim_{k \to \infty} \operatorname{tr}(G(\Sigma^{(l_k)})) = \operatorname{tr}(G(\Sigma^*)).
$$

According to (34), we have

$$
\operatorname{tr}(G(\Sigma^*)) = \operatorname{tr}(\Sigma^*).
$$

This implies $\operatorname{tr}(\Sigma^*) = 0$ by (34).

Then, since $\{\operatorname{tr}(\Sigma^{(l)})\}_{l=1}^{\infty}$ is a decreasing sequence, we have

$$
\lim_{l \to \infty} \mathbb{E}_{h \sim N(\mathbf{0}_n, \Sigma^{(l)})}[\|h\|^2] = \lim_{l \to \infty} \operatorname{tr}(\Sigma^{(l)}) = \lim_{k \to \infty} \operatorname{tr}(\Sigma^{(l_k)}) = \operatorname{tr}(\Sigma^*) = 0.
$$

Consequently, we have

$$
\lim_{L \to \infty} \mathbb{E}_{H \sim N(\mathbf{0}_n, \Sigma^{(L)})} \left[ \frac{\|H\|_{\mathrm{F}}^2}{\|X\|_{\mathrm{F}}^2} \right] = \frac{C}{\|X\|_{\mathrm{F}}^2} \lim_{L \to \infty} \mathbb{E}_{h \sim N(\mathbf{0}_n, \Sigma^{(L)})}[\|h\|^2] = 0.
$$

$\square$

# D Signal propagation theory for linear ResGCN

## D.1 NNGP correspondence for linear ResGCN

**Proposition D.1** (NNGP correspondence for linear ResGCN)**.** *As the width of the hidden layers $d \to \infty$, the l-th layer's pre-activation embedding channels $\{H_{:,k}^{(l)}\}_{k \in [d]}$ converge to i.i.d. Gaussian random variables $N(\mathbf{0}_n, \tilde{\Sigma}^{(l)})$ in distribution. The covariance matrices are*

$$\tilde{\Sigma}^{(1)} = \frac{\sigma_w^4}{d_0} \hat{A} X X^T \hat{A},$$
$$\tilde{\Sigma}^{(l+1)} = \alpha^2 \sigma_w^2 \hat{A} \tilde{\Sigma}^{(l)} \hat{A} + \beta^2 \tilde{\Sigma}^{(l)}. \tag{37}$$

*Moreover, as $d \to \infty$, the l-th layer's post-activation embedding channels $\{X_{:,k}^{(l)}\}_{k \in [d]}$ converge to i.i.d. random variables in distribution. The random variables have mean $\mathbf{0}_n$ and their covariance matrices $\Phi^{(l)}$, which satisfy*

$$\Phi^{(0)} = \frac{\sigma_w^2}{d_0} X X^T,$$
$$\Phi^{(l)} = \alpha^2 \sigma_w^2 \hat{A} \Phi^{(l-1)} \hat{A} + \beta^2 \Phi^{(l-1)}. \tag{38}$$

*Proof of Proposition D.1.* For $\Phi^{(l)}$, $\tilde{\Sigma}^{(l+1)}$ defined by (38) and (37), it is easy to show that $\tilde{\Sigma}^{(l+1)} = \sigma_w^2 \hat{A} \Phi^{(l)} \hat{A}$. Similar to the proof of Proposition C.1, We will prove this proposition by mathematical induction.

**Base case.** When $l = 0$, the $k$-th channel of $X^{(0)}$ is

$$X_{:,k}^{(0)} = X W_{:,k}^{(0)} + \mathbf{1}_n \cdot b_k^{(0)} = X W_{:,k}^{(0)}. \tag{39}$$

According to our initialization, the weights $\{W_{:,k}^{(0)}\}_{k \in [d]}$ are i.i.d. random variables, so $\{X_{:,k}^{(0)}\}_{k \in [d]}$ are also i.i.d. random variables. Taking the expectation of (39), we get

$$\mathbb{E}[X_{:,k}^{(0)}] = X \cdot \mathbb{E}[W_{:,k}^{(0)}] = \mathbf{0}_n.$$

Calculating the covariance matrix of (39), we have

$$\begin{aligned}
\text{Cov}[X_{:,k}^{(0)}] &= \mathbb{E}[X_{:,k}^{(0)} \cdot X_{:,k}^{(0)T}] = \mathbb{E}[X W_{:,k}^{(0)} W_{:,k}^{(0)T} X^T] \\
&= X \cdot \mathbb{E}[W_{:,k}^{(0)} W_{:,k}^{(0)T}] \cdot X^T = X \left( \frac{\sigma_w^2}{d_0} \cdot I_{d_0} \right) X^T \\
&= \frac{\sigma_w^2}{d_0} X X^T.
\end{aligned}$$

Thus, if we let $\Phi^{(0)} = \sigma_w^2 X X^T / d_0$, then we have $\{X_{:,k}^{(0)}\}_{k \in [d]}$ are i.i.d. with mean $\mathbf{0}_n$ and covariance matrix $\Phi^{(0)}$.

Now we study the pre-activation embedding $H^{(1)}$. Since the bias term $b^{(1)}$ is initialized to be $\mathbf{0}_d$, we have

$$H^{(1)} = \hat{A} X^{(0)} W^{(1)} + \mathbf{1}_n \cdot b^{(1)} = \hat{A} X^{(0)} W^{(1)}.$$

Similar to the proof of Proposition C.1 for vanilla GCN, we vectorize the equation and get

$$\text{vec}(H^{(1)}) = \sum_{k=1}^{d} \text{vec} \left( \underbrace{[\hat{A} X_{:,k}^{(0)}]}_{n \times 1} \cdot \underbrace{W_{k,:}^{(1)}}_{1 \times d} \right).$$

For brevity, we define

$$\omega_{kk'}^{(1)} := \sqrt{d} \cdot W_{kk'}^{(1)}, \quad \text{for all } k, k' \in [d]$$

and

$$Z_k^{(1)} := \text{vec}\left([\hat{A}X_{:,k}^{(0)}] \cdot w_{k,:}^{(l+1)}\right), \text{for all } k \in [d].$$

Then we get that $\{\omega_{kk'}^{(1)}\}_{k,k' \in [d]}$ are i.i.d. with mean 0 and variance $\sigma_w^2$, and

$$\text{vec}(H^{(1)}) = \frac{1}{\sqrt{d}} \sum_{k=1}^{d} Z_k^{(1)}.$$

Analogous to the proof of Proposition C.1, $\{Z_k^{(1)}\}_{k \in [d]}$ are i.i.d., $\mathbb{E}[Z_1^{(1)}] = \mathbf{0}_{nd}$, and

$$\begin{aligned}
\text{Cov}[Z_1^{(1)}] &= \mathbb{E}\left[\omega_{1,:}^{(1)T}\omega_{1,:}^{(1)}\right] \otimes \left\{\hat{A} \cdot \mathbb{E}\left[X_{:,1}^{(0)}X_{:,1}^{(0)T}\right] \cdot \hat{A}\right\} \\
&= \sigma_w^2 I_d \otimes \hat{A}\Phi^{(0)}\hat{A} \\
&= I_d \otimes \sigma_w^2 \hat{A}\Phi^{(0)}\hat{A}.
\end{aligned}$$

Since $\tilde{\Sigma}^{(1)} = \sigma_w^4 \hat{A}XX^T\hat{A}/d_0 = \sigma_w^2 \hat{A}\Phi^{(0)}\hat{A}$, applying the central limit theorem, $\text{vec}(H^{(1)})$ converges to a Gaussian random variable $N(\mathbf{0}_{nd}, I_d \otimes \tilde{\Sigma}^{(1)})$ as $d \to \infty$. Consequently, $\{H_{:,k}^{(1)}\}$ converge to i.i.d. Gaussian random variables $N(\mathbf{0}_n, \tilde{\Sigma}^{(1)})$ in distribution.

**Induction step.** Suppose that $\{X_{:,k}^{(l-1)}\}_{k \in [d]}$ converge to i.i.d. random variables with mean $\mathbf{0}_n$ and covariance matrix $\Phi^{(l-1)}$ in distribution. Suppose that $\{H_{:,k}^{(l)}\}_{k \in [d]}$ converge to i.i.d. Gaussian random variables $N(\mathbf{0}_n, \tilde{\Sigma}^{(l)})$ in distribution. Now we look at $X^{(l)}$ first.

For the linear ResGCN at initialization, the post-activation embeddings satisfy

$$X^{(l)} = \alpha H^{(l)} + \beta X^{(l-1)} = \alpha \hat{A}X^{(l-1)}W^{(l)} + \beta X^{(l-1)}$$

We take any $k$-th channel $X_{:,k}^{(l)}$ of $X^{(l)}$:

$$\begin{aligned}
X_{:,k}^{(l)} &= \alpha H_{:,k}^{(l)} + \beta X_{:,k}^{(l-1)} \\
&= \alpha \hat{A}X^{(l-1)}W_{:,k}^{(l)} + \beta X_{:,k}^{(l-1)} \\
&= \alpha \left(\sum_{k'=1}^{d} \hat{A}X_{:,k'}^{(l-1)}W_{k'k}^{(l)}\right) + \beta X_{:,k}^{(l-1)} \\
&= \underbrace{\frac{\alpha}{\sqrt{d}}\left(\sum_{k'=1}^{d} \hat{A}X_{:,k'}^{(l-1)}\omega_{k'k}^{(l)}\right)}_{(i)} + \underbrace{\beta X_{:,k}^{(l-1)}}_{(ii)},
\end{aligned}$$

where $\omega_{k'k}^{(l)} := W_{k'k}^{(l)}/\sqrt{d}$ has mean 0 and variance $\sigma_w^2$, which does not rely on $d$. By the induction hypothesis, $X_{:,k'}^{(l-1)}$ and $X_{:,k}^{(l-1)}$ are independent when $k' \neq k$. Then $(ii)$ is independent of the $k'$-th term $\alpha\hat{A}X_{:,k'}^{(l-1)}\omega_{k'k}^{(l)}/\sqrt{d}$ in $(i)$ when $k' \neq k$. We notice that the correlation between $(i)$'s $k$-th term $\alpha\hat{A}X_{:,k}^{(l-1)}\omega_{kk}^{(l)}/\sqrt{d}$ and $\beta X_{:,k}^{(l-1)}$ goes to 0 as $d \to \infty$. Thus, we get that $(i)$ and $(ii)$ are asymptotically independent, the expectation

$$\mathbb{E}[X_{:,k}^{(l)}] = \alpha\mathbb{E}[H_{:,k}^{(l)}] + \beta\mathbb{E}[X_{:,k}^{(l-1)}] = \mathbf{0}_n,$$

and the covariance matrix

$$\begin{aligned}
\text{Cov}[X_{:,k}^{(l)}] &= \alpha^2\text{Cov}[H_{:,k}^{(l)}] + \beta^2\text{Cov}[X_{:,k}^{(l-1)}] = \alpha^2\tilde{\Sigma}^{(l)} + \beta^2\Phi^{(l-1)} \\
&= \alpha^2\sigma_w^2\hat{A}\Phi^{(l-1)}\hat{A} + \beta^2\Phi^{(l-1)} \\
&= \Phi^{(l)}.
\end{aligned}$$

By the induction hypothesis, $\{H^{(l)}_{:,k}\}$ are i.i.d. and $\{X^{(l-1)}_{:,k}\}$ are i.i.d. as $d \to \infty$, so $\{X^{(l)}_{:,k}\}$ are i.i.d..

Next, we look at the pre-activation embedding $H^{(l+1)}$. We have

$$H^{(l+1)} = \hat{A}X^{(l)}W^{(l+1)} + \mathbf{1}_n \cdot b^{(l+1)} = \hat{A}X^{(l)}W^{(l+1)}.$$

We also vectorize it and get

$$\text{vec}(H^{(l+1)}) = \sum_{k=1}^{d} \text{vec}\big([\hat{A}X^{(l)}_{:,k}] \cdot W^{(l+1)}_{k,:}\big).$$

Analogous to the proof of base case (or the proof of Proposition C.1), we can conclude that $\{H^{(l+1)}_{:,k}\}$ converge i.i.d. to $N(\mathbf{0}_n, \sigma_w^2 \hat{A}\Phi^{(l)}\hat{A})$, i.e. $N(\mathbf{0}_n, \tilde{\Sigma}^{(l+1)})$.

**Conclusion.** By the principle of mathematical induction, we have proven this proposition. $\qquad \square$

### D.2 Proof of Theorem 3.3 (signal propagation on linear ResGCN)

To facilitate reading and understanding, we restate Theorem 3.3 here.

**Theorem 3.3.** *Suppose that there exists an eigenvector $u$ of $\hat{A}$ corresponding to the eigenvalue $1$, such that the input feature $X \in \mathbb{R}^{n \times d_0}$ satisfies $X^T u \neq \mathbf{0}_{d_0 \times 1}$. Under the NNGP correspondence approximation for linear ResGCN, if $\alpha^2 \sigma_w^2 + \beta^2 > 1$ and $\alpha \neq 0$, then we have*

$$\lim_{L \to \infty} \mathbf{M}^{(L)}_{FSP}(\sigma_w^2) = \infty \quad and \quad \lim_{L \to \infty} \mathbf{M}^{(L)}_{GEV}(\sigma_w^2) = 0.$$

*Proof of part $1$ in Theorem 3.3.* For any positive semi-definite matrix $\Sigma \in \mathcal{S}$, we have

$$\mathbb{E}_{h \sim N(\mathbf{0}_n, \Sigma)}[\|h\|^2] = \mathbb{E}_{h \sim N(\mathbf{0}_n, \Sigma)}[\text{tr}(h^T h)] = \mathbb{E}_{h \sim N(\mathbf{0}_n, \Sigma)}[\text{tr}(h h^T)] = \text{tr}(\Sigma).$$

Recalling the NNGP correspondence formula for linear ResGCN (37) in Proposition D.1, we have

$$\begin{aligned} \tilde{\Sigma}^{(1)} &= \frac{\sigma_w^4}{d_0} \hat{A} X X^T \hat{A}, \\ \tilde{\Sigma}^{(l+1)} &= \sigma_w^2 \alpha^2 \hat{A}\tilde{\Sigma}^{(l)}\hat{A} + \beta^2 \tilde{\Sigma}^{(l)}. \end{aligned} \tag{40}$$

By Proposition B.1, we can assume that $A = U\Lambda U^T$, where $\Lambda = \text{diag}(\lambda_1, \ldots, \lambda_n)$ with $1 = \lambda_1 \geq \cdots \geq \lambda_n > -1$ and $U \in \mathbb{R}^{n \times n}$ is an orthogonal matrix, i.e., $UU^T = U^T U = I_n$. Then from (40), we get

$$\begin{aligned} U^T \tilde{\Sigma}^{(l+1)} U &= \sigma_w^2 \alpha^2 \cdot U^T \hat{A}\tilde{\Sigma}^{(l)}\hat{A}U + \beta^2 \cdot U^T \tilde{\Sigma}^{(l)} U \\ &= \sigma_w^2 \alpha^2 \cdot \Lambda U^T \tilde{\Sigma}^{(l)} U \Lambda + \beta^2 \cdot U^T \tilde{\Sigma}^{(l)} U. \end{aligned} \tag{41}$$

So for any $i \in [n]$ and $l \in \mathbb{N}$, we have

$$\begin{aligned} (U^T \tilde{\Sigma}^{(l+1)} U)_{ii} &= \sigma_w^2 \alpha^2 \cdot \lambda_i (U^T \tilde{\Sigma}^{(l)} U)_{ii} \lambda_i + \beta^2 (U^T \tilde{\Sigma}^{(l)} U)_{ii} \\ &= (\alpha^2 \lambda_i^2 \sigma_w^2 + \beta^2) \cdot (U^T \tilde{\Sigma}^{(l)} U)_{ii}. \end{aligned}$$

Thus, for any $i \in [n]$ and $L \in \mathbb{N}$, we have

$$(U^T \tilde{\Sigma}^{(L)} U)_{ii} = (\alpha^2 \lambda_i^2 \sigma_w^2 + \beta^2)^{L-1} \cdot (U^T \tilde{\Sigma}^{(1)} U)_{ii}.$$

According to the assumption on input feature $X$, there exists an eigenvector $u$ of $\hat{A}$ corresponding to the eigenvalue $1$, such that $X^T u \neq \mathbf{0}_{d_0 \times 1}$. Suppose that $u_1, u_2, \ldots, u_n \in \mathbb{R}^{n \times 1}$ are the columns of $U$, then there exists $i \in [n]$ such that $X^T u_i \neq 0$. Otherwise, suppose that $u = \sum_{j=1}^{n} c_j u_j$ and $X^T u_j = 0$ for any $j \in [n]$, then $X^T u = \sum_{j=1}^{n} c_j X^T u_j = 0$. Contradiction!

Without loss of generality, we suppose that $Au_1 = u_1$ and $X^T u_1 \neq \mathbf{0}_{d_0 \times 1}$. Then we have

$$(U^T \tilde{\Sigma}^{(1)} U)_{11} = \frac{\sigma_w^4}{d_0} \cdot u_1^T \hat{A} X X^T \hat{A} u_1 = \frac{\sigma_w^4}{d_0} \cdot u_1^T X X^T u_1 = \frac{\sigma_w^4}{d_0} \cdot \|X^T u_1\|^2 > 0.$$

It results in

$$\mathrm{tr}(\tilde{\Sigma}^{(L)}) = \mathrm{tr}(U^T \tilde{\Sigma}^{(L)} U) \geq (U^T \tilde{\Sigma}^{(L)} U)_{11} = (\alpha^2 \sigma_w^2 + \beta^2)^{L-1} \cdot \frac{\sigma_w^4}{d_0} \|X^T u_1\|^2. \tag{42}$$

Therefore, if $\alpha^2 \sigma_w^2 + \beta^2 > 1$, we have

$$\begin{aligned}
\lim_{L \to \infty} \mathbf{M}_{\mathrm{FSP}}^{(L)}(\sigma_w^2) &= \lim_{L \to \infty} \mathbb{E}_{H^{(L)} \sim N(\mathbf{0}_n, \tilde{\Sigma}^{(L)})}[\|H^{(L)}\|_{\mathrm{F}}^2 / \|X\|_{\mathrm{F}}^2] \\
&= \frac{C}{\|X\|_{\mathrm{F}}^2} \lim_{L \to \infty} \mathbb{E}_{h \sim N(\mathbf{0}_n, \tilde{\Sigma}^{(L)})}[\|h\|^2] \\
&= \frac{C}{\|X\|_{\mathrm{F}}^2} \lim_{L \to \infty} \mathrm{tr}(\tilde{\Sigma}^{(L)}) = +\infty.
\end{aligned}$$

$\square$

*Proof of part* 2 *in Theorem 3.3.* For any positive semi-definite matrix $\Sigma \in \mathcal{S}$, we have

$$\mathbb{E}_{h \sim N(\mathbf{0}_n, \Sigma)}[\mathrm{Dir}(h)] = \mathbb{E}_{h \sim N(\mathbf{0}_n, \Sigma)}[\mathrm{tr}(h^T \hat{L} h)] = \mathbb{E}_{h \sim N(\mathbf{0}_n, \Sigma)}[\mathrm{tr}(\hat{L} h h^T)] = \mathrm{tr}(\hat{L}\Sigma).$$

So when we want to study $\mathbb{E}_{h \sim N(\mathbf{0}_n, \Sigma)}[\mathrm{Dir}(h)]$, we only need to look at $\mathrm{tr}(\hat{L}\Sigma)$ in the following of the proof.

Since $\hat{A}\hat{L} = \hat{A}(I_n - \hat{A}) = \hat{A} - \hat{A}^2 = (I_n - \hat{A})\hat{A} = \hat{L}\hat{A}$, we multiply $\hat{L}$ on both sides of the second equation in (40) and get

$$\begin{aligned}
\hat{L}\tilde{\Sigma}^{(l+1)} &= \sigma_w^2 \alpha^2 \cdot \hat{L}\hat{A}\tilde{\Sigma}^{(l)}\hat{A} + \beta^2 \hat{L}\tilde{\Sigma}^{(l)} \\
&= \sigma_w^2 \alpha^2 \cdot \hat{A}\hat{L}\tilde{\Sigma}^{(l)}\hat{A} + \beta^2 \hat{L}\tilde{\Sigma}^{(l)}.
\end{aligned}$$

Then for any $i \in [n]$ and $l \in \mathbb{N}$, we have

$$\begin{aligned}
(U^T \hat{L}\tilde{\Sigma}^{(l+1)} U)_{ii} &= \sigma_w^2 \alpha^2 \cdot \lambda_i (U^T \hat{L}\tilde{\Sigma}^{(l)} U)_{ii} \lambda_i + \beta^2 \cdot (U^T \hat{L}\tilde{\Sigma}^{(l)} U)_{ii} \\
&= (\alpha^2 \lambda_i^2 \sigma_w^2 + \beta^2) \cdot (U^T \hat{L}\tilde{\Sigma}^{(l)} U)_{ii}.
\end{aligned}$$

Thus, for any $i \in [n]$ and $L \in \mathbb{N}$, we have

$$(U^T \hat{L}\tilde{\Sigma}^{(L)} U)_{ii} = (\alpha^2 \sigma_w^2 \lambda_i^2 + \beta^2)^{L-1} \cdot (U^T \hat{L}\tilde{\Sigma}^{(1)} U)_{ii} \tag{43}$$

Since $U^T \hat{L} U = U^T(I_n - \hat{A})U = I_n - \Lambda$, we get

$$U^T \hat{L}\tilde{\Sigma}^{(1)} U = (I_n - \Lambda)U^T \tilde{\Sigma}^{(1)} U$$

We denote

$$r_i = (U^T \tilde{\Sigma}^{(1)} U)_{ii}, \quad \text{for any } i \in [n].$$

Then by (43), we have

$$(U^T \hat{L}\tilde{\Sigma}^{(L)} U)_{ii} = (\alpha^2 \sigma_w^2 \lambda_i^2 + \beta^2)^{L-1} \cdot (1 - \lambda_i) r_i,$$

From Proposition B.1, we have

$$\begin{aligned}
(U^T \hat{L}\tilde{\Sigma}^{(L)} U)_{ii} &\leq (\alpha^2 \sigma_w^2 \lambda^2 + \beta^2)^L \cdot (1 - \lambda_i) r_i, \quad \text{if } \lambda_i \in (-1, 1); \\
(U^T \hat{L}\tilde{\Sigma}^{(L)} U)_{ii} &= 0 = (\alpha^2 \sigma_w^2 \lambda^2 + \beta^2)^L \cdot (1 - \lambda_i) r_i, \quad \text{if } \lambda_i = 1,
\end{aligned}$$

where $\lambda = \max_{\lambda_i \neq 1} |\lambda_i| \in [0, 1)$. Thus, we get

$$\mathrm{tr}(\hat{L}\tilde{\Sigma}^{(L)}) = \mathrm{tr}(U^T \hat{L}\tilde{\Sigma}^{(L)} U) \leq (\alpha^2 \sigma_w^2 \lambda^2 + \beta^2)^{L-1} \cdot \sum_{i=1}^{n} (1 - \lambda_i) r_i.$$

We conclude that

$$\mathbb{E}_{H^{(L)} \sim N(\mathbf{0}_n, \tilde{\Sigma}^{(L)})}[\mathrm{Dir}(H^{(L)})] = C \cdot \mathbb{E}_{h \sim N(\mathbf{0}_n, \tilde{\Sigma}^{(L)})}[\mathrm{Dir}(h)] = C \cdot \mathrm{tr}(\hat{L}\tilde{\Sigma}^{(L)})$$

$$\leq C(\alpha^2\sigma_w^2\lambda^2 + \beta^2)^{L-1} \cdot \sum_{i=1}^{n}(1-\lambda_i)r_i.$$

Then we have

$$\frac{\mathbb{E}_{H^{(L)} \sim N(\mathbf{0}_n, \tilde{\Sigma}^{(L)})}[\mathrm{Dir}(H^{(L)})]}{(\alpha^2\sigma_w^2 + \beta^2)^{L-1}} \leq \left(C\sum_{i=1}^{n}(1-\lambda_i)r_i\right) \cdot \left(\frac{\alpha^2\sigma_w^2\lambda^2 + \beta^2}{\alpha^2\sigma_w^2 + \beta^2}\right)^{L-1}.$$

Since $\alpha^2\sigma_w^2 + \beta^2 > 1$ and $\alpha \neq 0$ as assumed in the statement of this theorem, we have $(\alpha^2\sigma_w^2\lambda^2 + \beta^2)/(\alpha^2\sigma_w^2 + \beta^2) \in [0, 1)$. So we get that

$$\lim_{L \to \infty} \frac{\mathbb{E}_{H^{(L)} \sim N(\mathbf{0}_n, \tilde{\Sigma}^{(L)})}[\mathrm{Dir}(H^{(L)})]}{(\alpha^2\sigma_w^2 + \beta^2)^L} = 0. \tag{44}$$

Recalling (42) in the proof of part 1 for Theorem 3.3, if we define

$$\delta_0 = \frac{\sigma_w^4}{d_0}\|X^T u_1\|^2 \quad \text{and} \quad K = \alpha^2\sigma_w^2 + \beta^2,$$

then given any $L \in \mathbb{N}$, we have

$$\frac{1}{K^{L-1}} \cdot \mathrm{tr}(\tilde{\Sigma}^{(L)}) \geq \delta_0 > 0. \tag{45}$$

Similar to the proof of part 2 in Theorem C.7, we have

$$\mathbb{E}_{H \sim N(\mathbf{0}_n, \tilde{\Sigma}^{(L)})}\left[\frac{\mathrm{Dir}(H)}{\|H\|_{\mathrm{F}}^2}\right]$$

$$= \mathbb{E}_{H \sim N(\mathbf{0}_n, \tilde{\Sigma}^{(L)})}\left[\frac{\mathrm{Dir}(H)}{\|H\|_{\mathrm{F}}^2}\mathbb{1}_{\{\|H\|_{\mathrm{F}}^2 > \epsilon K^{L-1}\}}\right] + \mathbb{E}_{H \sim N(\mathbf{0}_n, \tilde{\Sigma}^{(L)})}\left[\frac{\mathrm{Dir}(H)}{\|H\|_{\mathrm{F}}^2}\mathbb{1}_{\{\|H\|_{\mathrm{F}}^2 \leq \epsilon K^{L-1}\}}\right] \tag{46}$$

$$\leq \frac{\mathbb{E}_{H \sim N(\mathbf{0}_n, \tilde{\Sigma}^{(L)})}[\mathrm{Dir}(H)]}{\epsilon K^{L-1}} + 2 \cdot \mathbb{P}_{H \sim N(\mathbf{0}_n, \tilde{\Sigma}^{(L)})}[\|H\|_{\mathrm{F}}^2 \leq \epsilon K^{L-1}].$$

For any $L \geq 1$, there exists $i \in [n]$, such that $\tilde{\Sigma}_{ii}^{(L)} \geq \mathrm{tr}(\tilde{\Sigma}^{(L)})/n$. For any $n \times C$ random matrix $H \sim N(\mathbf{0}_n, \tilde{\Sigma}^{(L)})$, it holds that $H_{i,1} \sim N(0, \tilde{\Sigma}_{ii}^{(L)})$. By (45), we have

$$\mathbb{P}_{H \sim N(\mathbf{0}_n, \tilde{\Sigma}^{(L)})}\left[\|H\|_{\mathrm{F}}^2 \leq \epsilon K^{L-1}\right] \leq \mathbb{P}_{H \sim N(\mathbf{0}_n, \tilde{\Sigma}^{(L)})}\left[H_{i,1}^2 \leq \epsilon K^{L-1}\right]$$

$$= \mathbb{P}_{Z \sim N(0,1)}\left[Z^2 \leq \frac{\epsilon K^{L-1}}{\tilde{\Sigma}_{ii}^{(L)}}\right] \leq \mathbb{P}_{Z \sim N(0,1)}\left[Z^2 \leq \frac{\epsilon n K^{L-1}}{\mathrm{tr}(\tilde{\Sigma}^{(L)})}\right] \tag{47}$$

$$\leq \mathbb{P}_{Z \sim N(0,1)}\left[Z^2 \leq \frac{\epsilon n}{\delta_0}\right] = 2\Phi\left(\sqrt{\frac{\epsilon n}{\delta_0}}\right) - 1,$$

where $\Phi(x) = \mathbb{P}_{Z \sim N(0,1)}[Z \leq x]$ denotes the cumulative distribution function of the standard normal distribution $N(0, 1)$.

Combining (46) and (47), we get

$$\mathbb{E}_{H \sim N(\mathbf{0}_n, \tilde{\Sigma}^{(L)})}\left[\frac{\mathrm{Dir}(H)}{\|H\|_{\mathrm{F}}^2}\right] \leq \frac{1}{\epsilon K^{L-1}} \cdot \mathbb{E}_{H \sim N(\mathbf{0}_n, \tilde{\Sigma}^{(L)})}[\mathrm{Dir}(H)] + 4\Phi\left(\sqrt{\frac{\epsilon n}{\delta_0}}\right) - 2.$$

By (44) , we let $L \to \infty$ and get

$$\limsup_{L\to\infty} \mathbb{E}_{H\sim N(\mathbf{0}_n, \tilde{\Sigma}^{(L)})} \left[ \frac{\text{Dir}(H)}{\|H\|_\text{F}^2} \right]$$

$$\leq \frac{1}{\epsilon K^{L-1}} \cdot \limsup_{L\to\infty} \mathbb{E}_{H\sim N(\mathbf{0}_n, \tilde{\Sigma}^{(L)})}[\text{Dir}(H)] + 4\Phi\left(\sqrt{\frac{\epsilon n}{\delta_0}}\right) - 2 \tag{48}$$

$$= \frac{1}{\epsilon} \cdot 0 + 4\Phi\left(\sqrt{\frac{\epsilon n}{\delta_0}}\right) - 2 = 4\Phi\left(\sqrt{\frac{\epsilon n}{\delta_0}}\right) - 2.$$

Notice that the left hand side of (48) is independent of the choice of $\epsilon$. Since $\Phi$ is a continuous map, we let $\epsilon \to 0^+$ and get

$$\limsup_{L\to\infty} \mathbb{E}_{H\sim N(\mathbf{0}_n, \tilde{\Sigma}^{(L)})} \left[ \frac{\text{Dir}(H)}{\|H\|_\text{F}^2} \right] \leq 4\Phi(0) - 2 = 0.$$

Therefore, we have

$$\lim_{L\to\infty} \mathbf{M}_{\text{GEV}}^{(L)}(\sigma_w^2) = \lim_{L\to\infty} \mathbb{E}_{H\sim N(\mathbf{0}_n, \tilde{\Sigma}^{(L)})} \left[ \frac{\text{Dir}(H)}{\|H\|_\text{F}^2} \right] = 0.$$

$\square$

# E  SPoGInit algorithm

In Section 4, SPoGInit aims to find a better initialization by minimizing

$$w_1 \mathbf{V}_{\text{FSP}} + w_2 \mathbf{V}_{\text{BSP}} - w_3 \mathbf{M}_{\text{GEV}}^{(L)}.$$

In the implementation of SPoGInit algorithm, we always use one random weight sample to get point estimates $\hat{\mathbf{V}}_{\text{FSP}}, \hat{\mathbf{V}}_{\text{BSP}}, \hat{\mathbf{M}}_{\text{GEV}}^{(L)}$ of $\mathbf{V}_{\text{FSP}}, \mathbf{V}_{\text{BSP}}, \mathbf{M}_{\text{GEV}}^{(L)}$, respectively. We take deep vanilla GCNs as an example to showcase the SPoGInit methodology. Given any Xavier-initialized weight $\{\hat{W}^{(l)}\}_{l=1}^L$, SPoGInit scales the weights layer-wise by $\gamma = (\gamma^{(l)})_{l \in [L]} \in \mathbb{R}_{>0}^L$ to yield new initialization $\theta(\gamma) = \{W^{(l)}\}_{l=1}^L = \{\gamma^{(l)}\hat{W}^{(l)}\}_{l=1}^L$ that achieves proper signal propagation. To be more specific, SPoGInit algorithm solves the optimization problem

$$\min_\gamma F(\theta(\gamma)) := w_1 \hat{\mathbf{V}}_{\text{FSP}}(\gamma) + w_2 \hat{\mathbf{V}}_{\text{BSP}}(\gamma) - w_3 \hat{\mathbf{M}}_{\text{GEV}}^{(L)}(\gamma), \tag{49}$$

where

$$\hat{\mathbf{V}}_{\text{FSP}} := (\hat{\mathbf{M}}_{\text{FSP}}^{(1)}/\hat{\mathbf{M}}_{\text{FSP}}^{(L-1)} - 1)^2 = \left[ \frac{\|H^{(1)}(\theta(\gamma))\|_{\text{F}}}{\|H^{(L-1)}(\theta(\gamma))\|_{\text{F}}} - 1 \right]^2,$$

$$\hat{\mathbf{V}}_{\text{BSP}} := (\hat{\mathbf{M}}_{\text{BSP}}^{(2)}/\hat{\mathbf{M}}_{\text{BSP}}^{(L-1)} - 1)^2 = \left[ \frac{\|g^{(2)}(\theta(\gamma))\|_{\text{F}}}{\|g^{(L-1)}(\theta(\gamma))\|_{\text{F}}} - 1 \right]^2,$$

$$\hat{\mathbf{M}}_{\text{GEV}}^{(L)} := \frac{\text{Dir}(H^{(L)}(\theta(\gamma))}{\|H^{(L)}(\theta(\gamma)\|_{\text{F}}^2},$$

with $g^{(l)}(\theta(\gamma)) := \partial\ell/\partial W^{(l)}$.

## E.1  Implementation details

Now we explain SPoGInit (Algorithm 1) in detail.

In lines 2-3, we initialize the weight parameters and weight scales $\gamma^{(l)}(0) = 1$. We iteratively update $\theta(\gamma)$ as follows.

In line 5, we calculate the objective function $F(\theta(\gamma(t)))$ as defined in (49).

In lines 6-10, we update the weight parameters $\theta(\gamma(t))$ by optimizing the objective function through the projected gradient descent method to the scales $\{\gamma^{(l)}(t)\}_{l=1}^L$ for each layer. We adopt the projected gradient descent method to ensure the scales $\{\gamma^{(l)}(t)\}_{l=1}^L$ remain positive.

---

**Algorithm 1** *SPoGInit*: Searching for weight initialization with better Signal Propagation on Graph

---

1: normalized adjacency matrix $\hat{A}$, input $X(t)$, label $y(t)$, network depth $L$, hidden dimension $d$, learning rate $\eta$, total iterations $T$, metric weights $w_1, w_2, w_3$.
2: **initialize** $\gamma^{(l)}(0) = 1$ and sample $\{\hat{W}^{(l)}\}_{l=1}^L$ by Xavier initialization.
3: **initialize** $\theta(\gamma(0)) \triangleq \{W^{(l)}(0)\}_{l=1}^L$ by $W^{(l)}(0) \leftarrow \gamma^{(l)}(0) \cdot \hat{W}^{(l)}$.
4: **for** $t = 0, 1, \cdots, T-1$ **do**
5:     calculate the objective function $F(\theta(\gamma(t)))$ by $\hat{A}$, $X(t)$, $y(t)$ and $\theta(\gamma(t))$.
6:     **for** layers $l = 1, 2, \ldots, L$ **do**
7:         $\gamma^{(l)}(t+1) \leftarrow \gamma^{(l)}(t) - \eta\nabla_{\gamma^{(l)}} F(\theta(\gamma(t)))$.
8:         $\gamma^{(l)}(t+1) \leftarrow \text{Proj}_{[10^{-6}, \infty)}(\gamma^{(l)}(t+1))$.
9:         $W^{(l)}(t+1) \leftarrow \gamma^{(l)}(t+1) \cdot \hat{W}^{(l)}$.
10:     $\theta(\gamma(t+1)) \triangleq \{W^{(l)}(t+1)\}_{l=1}^L$.
11: **return** $\theta(\gamma(T))$.

---

In line 7 of Algorithm 1, unless otherwise specified, we employ the random direction finite difference method (Equation (31) in (Nesterov & Spokoiny, 2017)) to compute the derivative of the objective function with

respect to the scaling factor $\gamma$. Specifically, we use

$$\hat{\mathbb{E}}_\mu \left\{ \frac{F(\theta(\gamma(t) + \mu\delta)) - F(\theta(\gamma(t)))}{\delta} \mu \right\}$$

to approximate the gradient $\nabla_\gamma F(\theta(\gamma(t)))$, where $\delta$ is a small scalar and $\mu$ follows the standard Gaussian distribution $N(0, I_L)$. Here, $\hat{\mathbb{E}}_\mu$ denotes the Monte Carlo approximation of the expectation, computed by averaging over 3 independent and identically distributed (i.i.d.) samples of $N(0, I_L)$.

For the GCNs with skip connections, we replace $\hat{\mathbf{V}}_{\text{FSP}}$ and $\hat{\mathbf{V}}_{\text{BSP}}$ with $(\max_{1 \leq l < L} \hat{\mathbf{M}}_{\text{FSP}}^{(l)} / \min_{1 \leq l < L} \hat{\mathbf{M}}_{\text{FSP}}^{(l)} - 1)^2$ and $(\max_{1 < l < L} \hat{\mathbf{M}}_{\text{BSP}}^{(l)} / \min_{1 < l < L} \hat{\mathbf{M}}_{\text{BSP}}^{(l)} - 1)^2$, respectively. Also, in these models, we use the same scaling factor across all layers. We empirically find that these modifications more effectively minimize the objective function in GCNs with skip connections.

### E.2 Analysis on the coefficient $w_2$ in SPoGInit

We analyze the effect of the coefficient $w_2$ in SPoGInit, which controls the contribution of the backward signal propagation (BSP) term $\hat{V}_{BSP}$ in the objective function (Formula (49)). Empirically, we observe that the initial value of $\hat{V}_{BSP}$ is substantially lower than that of $\hat{V}_{FSP}$, leading an imbalance between the two objectives. To mitigate this discrepancy, in practice, we set a larger value for $w_2$ than $w_1$ in order to amplify the role of $\hat{V}_{BSP}$ during optimization.

We conduct experiments on a 32-layer GCN with tanh activation, trained on the Cora dataset. Table 7 reports the values of $\hat{V}_{FSP}$ and $\hat{V}_{BSP}$ under different settings of $w_2 \in \{1, 10, 100\}$. We fix the learning rate at 0.1, use 100 optimization iterations, and set $w_1 = w_3 = 1$. We report the average result over 50 runs. The results reveal two key trends regarding the impact of $w_2$:

- **Gradual reduction of $\hat{V}_{BSP}$ as $w_2$ increases**: As $w_2$ increases from 1 to 10 and then to 100, $\hat{V}_{BSP}$ decreases from approximately $3 \times 10^{-3}$ to $1.1 \times 10^{-3}$ and finally to $1.0 \times 10^{-3}$. This consistent but diminishing improvement suggests that increasing $w_2$ enhances backward signal propagation, though the marginal gain becomes smaller at higher values.

- **Forward signal propagation deteriorates at large $w_2$**: While $\hat{V}_{FSP}$ remains stable between $w_2 = 1$ and $w_2 = 10$, it increases sharply when $w_2$ reaches 100, indicating that an overly large $w_2$ harms forward signal preservation.

These findings suggest that $w_2 = 10$ achieves a favorable trade-off between forward and backward signal preservation, yielding low values for both $\hat{V}_{FSP}$ and $\hat{V}_{BSP}$. Based on this analysis, we adopt $w_2 = 10$ as the default setting in the subsequent experiments.

Table 6: The $\hat{V}_{FSP}$ and $\hat{V}_{BSP}$ of 32-layer tanh-activated GCN using SPoGInit with different $w_2$ on Cora dataset. In SPoGInit, we set the learning rate as 0.1, total iteration as 100, $w_1$ and $w_3$ as 1. We report the average result over 50 runs.

|  | w/o SPoG | SPoG using $w_2 = 1$ | SPoG using $w_2 = 10$ | SPoG using $w_2 = 100$ |
|---|---|---|---|---|
| $\hat{V}_{FSP}$ | 31.2 | $5.4 \times 10^{-3}$ | $6.4 \times 10^{-3}$ | $7.5 \times 10^{-2}$ |
| $\hat{V}_{BSP}$ | $1.3 \times 10^{-1}$ | $3.0 \times 10^{-3}$ | $1.1 \times 10^{-3}$ | $1.0 \times 10^{-3}$ |

### E.3 Computational efficiency of GEV in SPoGInit

In this subsection, we analyze the extra computational cost brought by introducing GEV metric in SPoGInit. We claim that the additional cost introduced by adding the GEV term to SPoGInit's objective is minimal, and we believe the performance gains justify its inclusion.

**Technical explanation:**

Recall that the GEV metric is computed as $\hat{M}_{GEV}^{(L)} := \frac{Dir(H^{(L)})}{\|H^{(L)}\|_F^2}$, where the Dirichlet energy is given by $Dir(H) = \mathrm{tr}(H^\top \hat{L} H)$, and $\hat{L}$ denotes the normalized Laplacian of the input graph $\mathcal{G}$ (see Appendix B for mathematical details). Importantly, the GEV metric depends solely on the final-layer node embeddings $H^{(L)}$, which are computed in a single forward pass of the network. The additional cost of computing the norm and Dirichlet energy is negligible compared to the forward pass itself.

We note that the computation of FSP metric already requires a forward pass. Thus, the final-layer embeddings for GEV computation can be directly obtained from this same forward pass, meaning no additional computation is needed.

**Empirical evaluation:**

To evaluate the computational efficiency of incorporating GEV, we conduct experiments using a 64-layer GCN with tanh activation on the Cora and PubMed datasets. As shown in Table 7, the runtime of the SPoGInit algorithm with GEV is only slightly higher than that without GEV, with an increase of just about 1%–3%. This indicates that adding GEV introduces minimal overhead to the initialization process. In practice, this additional cost is negligible and acceptable, given the consistent performance improvements observed when GEV is used alongside the FSP and BSP metrics.

Table 7: Initialization time (in seconds) of a 64-layer tanh-activated GCN using SPoGInit with and without GEV on the Cora and PubMed datasets. For each case, the total number of initialization iterations is set to 40, and the results are averaged over 3 runs.

|  | **Cora** | **PubMed** |
|---|---|---|
| SPoGInit without GEV | 6.46 | 9.90 |
| SPoGInit with GEV | 6.66 | 10.0 |
| Relative Overhead of GEV | 3.1% | 1.1% |

In summary, since the GEV computation utilizes existing outputs from the forward pass with minimal overhead, its integration into SPoGInit's objective is both computationally efficient and beneficial in practice.

### E.4 Comparison of the per-layer scaling factors across different baseline initializations

By lines 9–10 of Algorithm 1, SPoGInit's adjustment of the scaling factor $\gamma^{(l)}$ is equivalent to modifying the initialization standard deviation $\sigma_{w,i}$. We have plotted the values of $\sigma_{w,i}$ for each layer of our SPoGInit, as well as for several baseline initialization methods, including Conventional initialization, Xavier, VirgoFor, and VirgoBack. As shown in Figure 6, $\sigma_{w,i}$ differs significantly across different initialization methods. Notably, SPoGInit performs an adaptive variance search, which leads to a fluctuating pattern of $\sigma_{w,i}$ across layers.

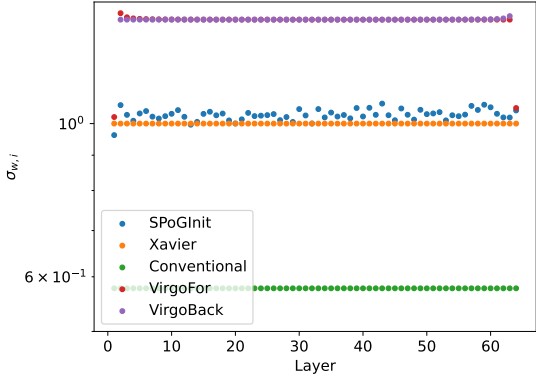

Figure 6: Layer-wise $\sigma_{w,i}$ of a 64-layer tanh-activated GCN using different initialization methods on the Cora datasets. In SPoGInit, we set the learning rate as 0.02, the early stop step $\delta$ as 10, and the number of total step is set as 40.

## F  Additional experiments

### F.1  Experiments on additional datasets

Building upon the experimental results in Section 5, we further evaluate the performance of SPoGInit on additional homophily and non-homophily datasets. For homophily datasets, we consider Amazon-photo and Amazon-computers (Shchur et al., 2018), following the data splits used in Luo et al. (2024). For heterophily datasets, in addition to the Arxiv-year dataset examined in Section 5, we incorporate Roman-empire and Amazon-ratings (Platonov et al., 2023), using their default splits. Table 8 summarizes the dataset statistics. Further implementation details are provided in Appendix G.1.

Table 8: Statistics of the additional homophily datasets used in the experiments.

| Dataset | Nodes | Features | Edges | Class | Homophily | Training/Validation/Test |
|---|---|---|---|---|---|---|
| Amazon-photo | 7,650 | 745 | 238,162 | 8 | 0.83 | 60%/20%/20% |
| Amazon-computers | 13,752 | 767 | 491,722 | 10 | 0.78 | 60%/20%/20% |
| Roman-empire | 22,662 | 300 | 32,927 | 18 | 0.05 | 50%/25%/25% |
| Amazon-ratings | 24,492 | 300 | 93,050 | 5 | 0.38 | 50%/25%/25% |

**Evaluation on homophily datasets**

We first evaluate the performance of GCN models with varying depths and initialization schemes on the two homophily datasets, Amazon-photo and Amazon-computers. The results are summarized in Table 9. To limit computational overhead, we enforce a maximum depth of 32 layers.

Table 9: Test accuracies of GCN models with varying depths and initializations on Amazon-photo and Amazon-computers. The bold figure highlights the best performance among different initializations. "Deg" refers to the test accuracy degradation as the depth increases from 4 to 32 layers. The smallest performance drops are highlighted in orange. The results demonstrate that SPoGInit significantly reduces performance degradation compared to baseline initializations and enhances the performance of deep GNN models across different architectures.

| Model | Init. | Amazon-photo | | | | | Amazon-computers | | | | |
|---|---|---|---|---|---|---|---|---|---|---|---|
| | | 4 | 8 | 16 | 32 | Deg. | 4 | 8 | 16 | 32 | Deg. |
| GCN | Conventional | 94.1 | 93.4 | 68.3 | 55.0 | ↓ 39.1 | **91.9** | 84.5 | 56.5 | 55.8 | ↓ 36.1 |
| | Xavier | 94.1 | 93.6 | 92.1 | 85.9 | ↓ 8.2 | 91.6 | 89.2 | 82.4 | 74.7 | ↓ 16.9 |
| | VirgoFor | **94.5** | **93.7** | **92.5** | 85.1 | ↓ 9.4 | **91.9** | **89.4** | 82.1 | **79.7** | ↓ 12.2 |
| | VirgoBack | **94.5** | 93.0 | 92.1 | 85.2 | ↓ 9.3 | 91.7 | **89.4** | **84.3** | 79.3 | ↓ 12.4 |
| | SPoGInit | 94.3 | 93.9 | 92.0 | **86.5** | ↓ 7.8 | 91.8 | 89.3 | 83.9 | **79.7** | ↓ 12.1 |
| ResGCN | Conventional | 96.2 | 96.4 | 96.2 | **96.5** | ↑ 0.3 | 92.8 | 93.4 | **93.9** | 93.8 | ↑ 1.0 |
| | Xavier | 96.6 | **96.6** | **96.3** | 95.7 | ↓ 0.9 | 93.2 | 93.6 | 93.5 | 93.1 | ↓ 0.1 |
| | VirgoFor | 96.4 | 96.2 | 95.5 | 94.1 | ↓ 2.3 | **93.4** | 93.4 | 92.5 | 90.5 | ↓ 2.9 |
| | VirgoBack | **96.7** | 96.1 | 95.6 | 93.4 | ↓ 3.3 | 93.3 | 93.2 | 92.5 | 90.1 | ↓ 3.2 |
| | SPoGInit | 96.4 | 96.5 | **96.3** | 96.4 | 0 | 92.7 | 93.2 | 93.8 | **93.9** | ↑ 1.2 |
| gatResGCN | Conventional | 96.0 | 96.4 | 96.4 | **96.6** | ↑ 0.6 | 92.8 | 92.7 | 93.3 | 93.4 | ↑ 0.6 |
| | Xavier | 96.4 | **96.6** | **96.5** | 95.4 | ↓ 1.0 | 92.8 | 93.0 | **93.7** | 93.3 | ↑ 0.5 |
| | VirgoFor | 96.4 | 96.4 | 95.3 | 91.7 | ↓ 4.7 | **92.9** | **93.4** | 92.3 | 87.5 | ↓ 5.4 |
| | VirgoBack | **96.6** | 96.3 | 95.2 | 91.2 | ↓ 5.4 | **92.9** | 93.3 | 92.2 | 86.3 | ↓ 6.6 |
| | SPoGInit | 96.1 | 96.2 | 96.4 | 96.5 | ↑ 0.4 | 92.5 | 92.7 | 93.3 | **93.7** | ↑ 1.2 |
| MixHop | Conventional | 96.1 | 95.7 | 92.9 | 85.6 | ↓ 10.4 | 92.7 | 90.6 | 89.3 | 79.7 | ↓ 13.0 |
| | Xavier | **96.4** | **95.8** | 94.6 | 91.4 | ↓ 5.0 | 93.1 | **92.9** | 91.0 | 83.8 | ↓ 9.3 |
| | SPoGInit | 96.3 | 95.7 | **95.5** | **92.5** | ↓ 3.9 | **93.3** | 92.6 | **91.6** | **88.4** | ↓ 4.9 |

Table 9 indicates that SPoGInit enables deep vanilla GCN and MixHop models to achieve better performance than other initialization baselines. Despite the general trend that increasing depth leads to some degree of performance degradation, models initialized with SPoGInit exhibit a more gradual decline in accuracy. Specifically, performance degradation is most severe under the Conventional initialization, where test accuracy can drop by over 30% when increasing the depth from 4 to 32 layers. Alternative initialization methods such as Xavier, VirgoFor, and VirgoBack mitigate this degradation to some extent. Among all initializations, SPoGInit consistently demonstrates the best performance in deeper models. Notably, on the Amazon-computers dataset, SPoGInit reduces the performance drop in MixHop by 4.4% compared to the best-performing baseline, Xavier initialization.

Additionally, Table 9 indicates that SPoGInit benefits from increased depth in ResGCN and gatResGCN, leading to superior performance for 32-layer networks. On the Amazon-Computers dataset, it outperforms other initialization baselines, while on the Amazon-Photo dataset, it achieves very competitive performance. We observe that deep ResGCN and gatResGCN maintain strong performance, with test accuracy exceeding 90% under widely used initialization schemes. We hypothesize that these two datasets are relatively simple, allowing residual connections to facilitate stable optimization during training, which in turn enhances model performance.

These findings underscore the critical role of initialization in deep networks and demonstrate the advantages of SPoGInit in mitigating performance degradation in deep GNN models.

### Evaluation on non-homophily datasets

### Amazon-ratings dataset

We evaluate SPoGInit on the heterophily dataset, Amazon-ratings, with results presented in Table 10.

As presented in Table 10, for ResGCN, gatResGCN, and MixHop models, SPoGInit outperforms other initialization baseline methods when increasing the network depth. Conventional and Xavier initialization lead to performance degradation as network depth increases, while VirgoFor and VirgoBack suffer even more severe degradation. In contrast, SPoGInit significantly mitigates performance degradation of deeper networks. Moreover, on ResGCN and gatResGCN, SPoGInit consistently improves performance as depth increases, demonstrating its effectiveness in deep models.

For the vanilla GCN, we observe that both VirgoFor and VirgoBack successfully mitigate performance degradation. Notably, VirgoBack achieves a 2% performance improvement as depth increases. However, this improvement is largely due to its relatively weaker performance on shallow GCNs (4 layers). In contrast, SPoGInit outperforms other initializations on deeper GCNs, highlighting its robustness in handling depth-related challenges.

### Roman-empire dataset

Beyond Amazon-ratings, we extend our experiments to the heterophily dataset Roman-empire. The original paper (Platonov et al., 2023) adopts skip connections for all the tested GCN models. Empirically, we indeed find that without skip connections, models perform severely worse. Thus, for this dataset we focus on the performance of ResGCNs across varying depths and initializations.

Table 11 shows that all baseline initializations suffer from severe performance degradation in deep models. In contrast, SPoGInit consistently achieves superior performance on deep GCN models and substantially reduces performance degradation as the network depths increase.

### Summary

In summary, across these additional homophily and heterophily datasets, SPoGInit consistently outperforms baseline initializations on deep GCN models, and substantially alleviating performance degradation in deep GCN models. These results provide robust empirical evidence for the effectiveness of SPoGInit in enhancing the stability and performance of deep GCNs.

Table 10: Test accuracies of GCN models on Amazon-ratings dataset with varying depths and initializations. The bold figure highlights the best performance among different initializations. "Deg" refers to the test accuracy degradation as the depth increases from 4 to 32 layers. The smallest performance drops are highlighted in orange. The results demonstrate that SPoGInit significantly reduces performance degradation compared to baseline initializations and enhances the performance of deep GNN models across different architectures.

| Model | Init. | Amazon-ratings | | | | |
|---|---|---|---|---|---|---|
| | | 4 | 8 | 16 | 32 | Deg. |
| GCN | Conventional | 44.7 | 44.9 | 42.5 | 38.2 | ↓ 6.5 |
| | Xavier | 44.4 | 45.6 | 45.1 | 44.0 | ↓ 0.4 |
| | VirgoFor | **45.1** | **45.9** | **45.8** | 45.3 | ↑ 0.2 |
| | VirgoBack | 43.1 | 44.5 | 45.5 | 45.1 | ↑ 2.0 |
| | SPoGInit | 45.0 | **45.9** | 45.7 | **45.4** | ↑ 0.4 |
| ResGCN | Conventional | 48.5 | 48.9 | 48.9 | 47.6 | ↓ 0.9 |
| | Xavier | **48.9** | **49.2** | 48.5 | 42.1 | ↓ 6.8 |
| | VirgoFor | 48.6 | 48.8 | 46.5 | 36.8 | ↓ 11.8 |
| | VirgoBack | 48.6 | 49.0 | 45.9 | 36.7 | ↓ 11.6 |
| | SPoGInit | 48.6 | 49.1 | **49.3** | **49.5** | ↑ 0.9 |
| gatResGCN | Conventional | 48.8 | 48.8 | **49.3** | 47.4 | ↓ 1.4 |
| | Xavier | **49.0** | **49.2** | 48.9 | 42.3 | ↓ 6.7 |
| | VirgoFor | 48.4 | 48.6 | 46.3 | 36.8 | ↓ 11.6 |
| | VirgoBack | 48.3 | 48.5 | 45.6 | 36.7 | ↓ 11.6 |
| | SPoGInit | 48.4 | 48.9 | 49.1 | **49.2** | ↑ 0.8 |
| MixHop | Conventional | 45.8 | 46.1 | 44.8 | 44.8 | ↓ 1.0 |
| | Xavier | 45.8 | 46.0 | 44.9 | 44.8 | ↓ 1.0 |
| | SPoGInit | **46.9** | **47.7** | **47.2** | **46.1** | ↓ 0.8 |

Table 11: Test accuracies of ResGCN model with varying depths and initializations on the Roman-empire dataset. The bold figure highlights the best performance among different initializations. "Deg" refers to the test accuracy degradation as the depth increases from 4 to 32 layers. The smallest performance drops are highlighted in orange. The results demonstrate that SPoGInit significantly reduces performance degradation compared to baseline initializations and enhances the performance of deep networks.

| Model | Init. | Amazon-ratings | | | | |
|---|---|---|---|---|---|---|
| | | 4 | 8 | 16 | 32 | Deg. |
| ResGCN | Conventional | **78.9** | **79.4** | 79.2 | 76.9 | ↓ 2.0 |
| | Xavier | 78.7 | 79.0 | 78.2 | 42.3 | ↓ 36.4 |
| | VirgoFor | 78.3 | 78.9 | 64.6 | 15.6 | ↓ 62.7 |
| | VirgoBack | 78.3 | 78.9 | 66.5 | 15.9 | ↓ 62.4 |
| | SPoGInit | **78.9** | **79.4** | **79.5** | **79.1** | ↑ 0.2 |

### F.2 Comparison and integration of SPoGInit with additional methods

In this subsection, we evaluate SPoGInit by comparing it with DGN (Zhou et al., 2020b), a normalization technique designed to mitigate oversmoothing, and CO-GNN (Finkelshtein et al., 2023), a dynamic message-passing framework. Additionally, we examine the effects of integrating SPoGInit with these methods to assess potential performance improvements.

**Comparison and integration of SPoGInit with DGN**

We first evaluate SPoGInit in comparison with DGN and assess their combined effects. Since DGN (Zhou et al., 2020b) was originally designed for GCN models without skip connections, we perform experiments on vanilla GCN using Cora, PubMed, and the large-scale OGBN-Arxiv dataset. The detailed results are presented in Table 12 and Table 13.

Table 12: Test accuracies of GCN models with SPoGInit or DGN technique on Cora and PubMed datasets. The bold figure highlights the best performance among different initializations. "Deg" refers to the test accuracy degradation as the depth increases from 4 to 32 layers. The smallest performance drops are highlighted in orange. The results demonstrate that SPoGInit significantly reduces performance degradation compared to DGN.

| Model | Method. | Cora | | | | | PubMed | | | | |
|---|---|---|---|---|---|---|---|---|---|---|---|
| | | 4 | 8 | 16 | 32 | Deg. | 4 | 8 | 16 | 32 | Deg. |
| | DGN | **80.6** | **79.8** | 75.8 | 72.6 | ↓ 8.0 | **77.7** | **78.8** | **78.0** | 77.6 | ↓ 0.2 |
| GCN | SPoGInit | 79.8 | 79.6 | **78.2** | **75.8** | ↓ 4.0 | 77.4 | 77.5 | 77.0 | **78.4** | ↑ 1.0 |
| | SPoGInit + DGN | 80.4 | 79.2 | 76.8 | 75.0 | ↓ 5.4 | 77.5 | 77.9 | **78.0** | 77.5 | 0 |

Table 13: Test accuracies of GCN models with SPoGInit or DGN technique on the OGBN-Arxiv dataset. The bold figure highlights the best performance among different initializations. "Deg" refers to the test accuracy degradation as the depth increases from 4 to 32 layers. The smallest performance drops are highlighted in orange. The results demonstrate that SPoGInit combines effectively with DGN and significantly reduces its performance degradation as the depth increases.

| Model | Method | OGBN-Arxiv | | | | |
|---|---|---|---|---|---|---|
| | | 4 | 8 | 16 | 32 | Deg. |
| | DGN | 69.7 | 68.2 | 63.7 | 61.7 | ↓ 8.0 |
| GCN | SPoGInit | 69.7 | 69.2 | 68.1 | 63.1 | ↓ 6.6 |
| | SPoGInit+DGN | **69.9** | **69.3** | **69.6** | **68.3** | ↓ 1.6 |

Experimental results in Table 12 demonstrate that, compared to DGN, SPoGInit more effectively mitigates performance degradation as GCN depth increases on small-graph datasets such as Cora and PubMed. Moreover, on the Cora dataset, initializing the model with SPoGInit before applying DGN improves DGN's effectiveness, reducing performance degradation in deep GCNs by over 2% compared to using DGN alone, though it remains slightly less effective than using SPoGInit alone.

Table 13 highlights the effectiveness of SPoGInit in deep GCNs on the large-scale OGBN-Arxiv dataset. Specifically, using SPoGInit alone outperforms DGN alone by over 1% in deeper networks. Moreover, integrating SPoGInit with DGN further enhances performance, achieving an improvement of more than 5% compared to either method individually. This effect is particularly pronounced on large graphs, likely due to the greater difficulty of achieving stable optimization in such settings.

**Comparison and integration of SPoGInit with CO-GNN**

We evaluate the performance of SPoGInit in comparison with CO-GNN—a method based on a dynamic message-passing framework—and also investigate the benefits of integrating SPoGInit into CO-GNN. We choose ResGCN as the backbone architecture of our SPoGInit. Our experiments are conducted on three benchmark datasets: Cora and PubMed, which exhibit high homophily, and Amazon-ratings, which is

characterized by heterophily. For consistency with our experimental setup, we use the dataset splits detailed in Table 1 (instead of those in Finkelshtein et al. (2023)). Detailed performance results for Cora and PubMed are reported in Table 14, while those for Amazon-ratings are provided in Table 15.

CO-GNN comprises two key components: an action network and an environment network. We define the overall layer count of CO-GNN as the sum of the layers in these two networks. Specifically, for the Cora and PubMed datasets a one-layer action network is employed, whereas for Amazon-ratings a two-layer action network is utilized.

Experimental results indicate that CO-GNN exhibits a pattern of performance variation as network depth increases. On datasets such as Cora and Amazon-ratings, performance initially improves with increasing depth; however, beyond 16 layers, further deepening leads to significant degradation. We hypothesize that this behavior arises from the need to jointly train the action and environment networks—a process that requires greater training effort to fully converge in deeper architectures compared to conventional GCNs.

Table 14 demonstrates that as network depth increases, SPoGInit outperforms CO-GNN on homophilic datasets such as Cora and PubMed. This indicates that SPoGInit more effectively mitigates the performance degradation typically observed in deeper architectures. On the heterophilic Amazon-ratings dataset, Table 15 shows that while CO-GNN alone attains higher performance than SPoGInit alone, integrating SPoGInit into CO-GNN further improves its performance, enabling deeper networks to achieve superior results while reducing degradation. We hypothesize that this improvement stems from SPoGInit's ability to enhance signal propagation in deep CO-GNN architectures, thereby facilitating more effective training. Notably, our experiments show that the training accuracy of a 32-layer CO-GNN on Amazon-ratings increases from 81.1% to 86.7% after integration with SPoGInit. (Training accuracies are not reported in the main result tables.)

Table 14: Test accuracies of CO-GNN model and ResGCN with SPoGInit on the Cora and PubMed datasets. The bold figure highlights the best performance among different initializations. "Deg" refers to the test accuracy degradation as the depth increases from 4 to 32 layers. The smallest performance drops are highlighted in orange. The results demonstrate that SPoGInit + ResGCN significantly reduces performance degradation compared to CO-GNN and enhances the performance of deep ResGCN.

| Method. | Cora | | | | | PubMed | | | | |
|---|---|---|---|---|---|---|---|---|---|---|
| | 4 | 8 | 16 | 32 | Deg. | 4 | 8 | 16 | 32 | Deg. |
| CO-GNN | **78.2** | **78.7** | **80.2** | 73.7 | ↓ 4.5 | **77.2** | **76.2** | 76.0 | 75.7 | ↓ 1.5 |
| SPoGInit+ResGCN | 75.7 | 77.9 | 78.5 | **78.5** | ↑ 2.8 | 75.4 | **76.2** | **76.4** | **77.0** | ↑ 1.6 |

Table 15: Test accuracies of CO-GNN, SPoGInit+ResGCN, and SPoGInit+CO-GNN on the Amazon-ratings dataset. The bold figure highlights the best performance among different initializations. "Deg" refers to the test accuracy degradation as the depth increases from 4 to 32 layers. The smallest performance drops are highlighted in orange. The results demonstrate that SPoGInit combines effectively with CO-GNN and significantly reduces its performance degradation.

| Method | Amazon-ratings | | | | |
|---|---|---|---|---|---|
| | 4 | 8 | 16 | 32 | Deg. |
| CO-GNN | 48.1 | **50.3** | **51.1** | 50.6 | ↑ 2.5 |
| SPoGInit+ResGCN | **48.6** | 49.1 | 49.3 | 49.5 | ↑ 0.9 |
| SPoGInit+CO-GNN | 47.9 | 49.4 | 50.4 | **51.0** | ↑ 3.1 |

In conclusion, our experiments reveal that SPoGInit surpasses these methods or combines effectively with them, leading to enhanced performance. This demonstrates the versatility of SPoGInit when applied alongside advanced techniques in tackling diverse challenges in GNNs.

### F.3 More experiments on signal propagation

In this subsection, we provide additional insights into the behavior of signal propagation (SP) in deep GCNs by visualizing activation and gradient matrices using heatmaps. Our goal is to illustrate clearly how widely used initialization methods fail to maintain stable signal propagation and to highlight how SPoGInit addresses these issues effectively.

Specifically, we examine a 256-layer GCN using the Tanh activation function. For each initialization, we visualize the activation and gradient matrices at shallow (2-nd), intermediate (127-th), and deep (255-th) layers. Each activation matrix has a shape of $N \times d$, where $N$ denotes the number of graph nodes, and $d = 64$ denotes the network width. For visualization, we select embeddings corresponding to the first 64 nodes.

We first observe that Conventional initialization (Figure 7) and Xavier initialization (Figure 8) lead to degraded signal propagation. Specifically, activation values at deeper layers and gradients at shallower layers become excessively small, indicative of forward and backward signal vanishing issues. By contrast, VirgoFor initialization (Figure 9) and VirgoBack initialization (Figure 10) maintain relatively stable activation matrices but result in gradients at shallower layers significantly exceeding acceptable thresholds, indicating slight backward SP explosion. In comparison, as depicted in Figure 11, SPoGInit successfully maintains stable activation and gradient matrices across all depths, effectively mitigating both SP vanishing and explosion issues in deep GCNs.

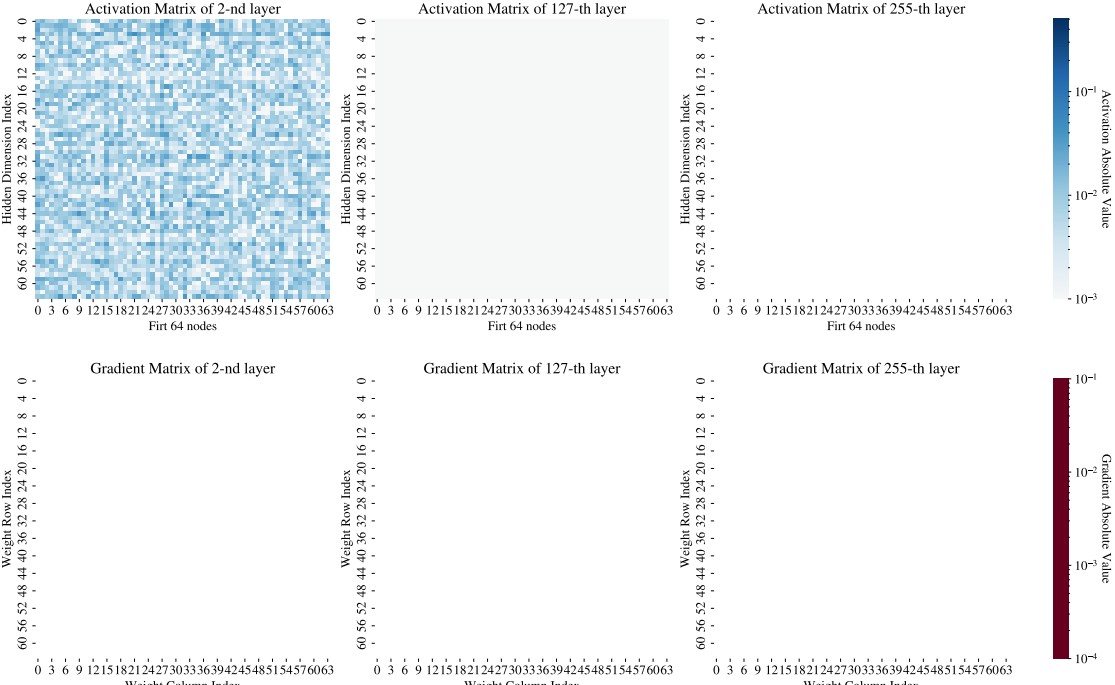

Figure 7: The heatmap of the activation and gradient matrix of a 256-layer tanh-activated GCN with Conventional initialization.

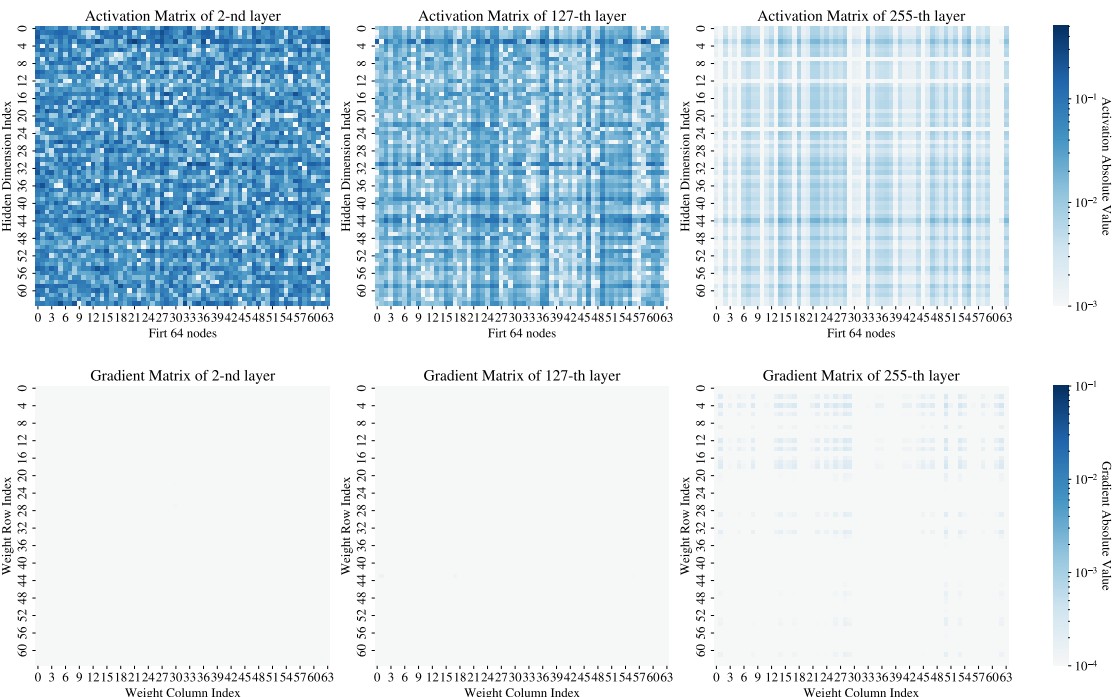

Figure 8: The heatmap of the activation and gradient matrix of a 256-layer tanh-activated GCN with Xavier initialization.

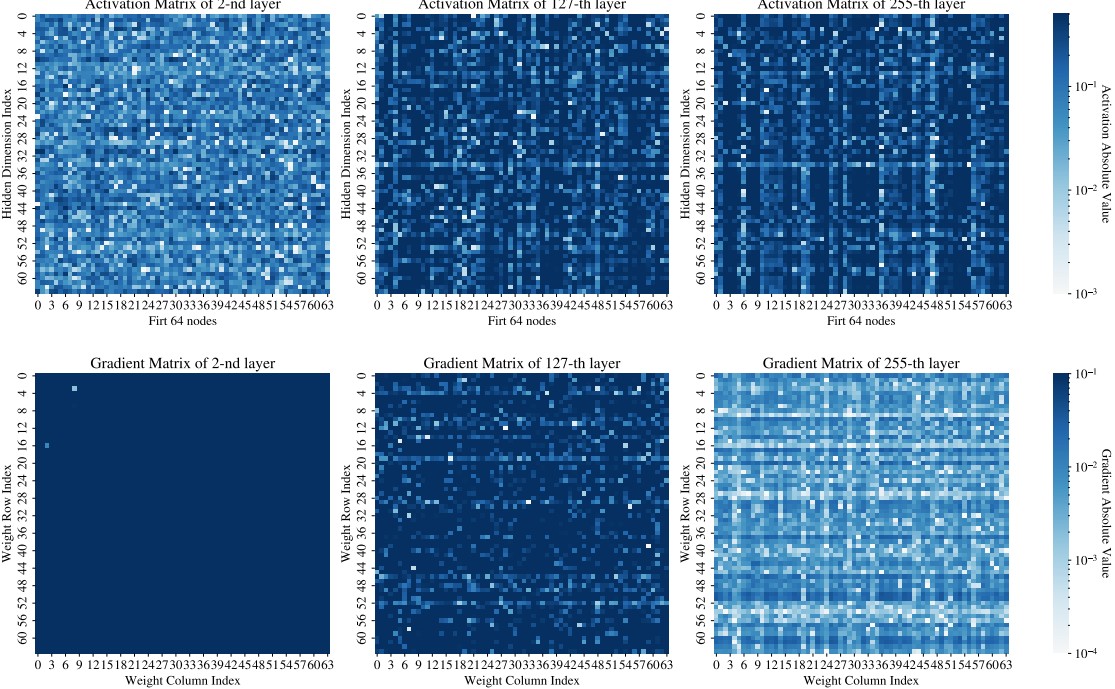

Figure 9: The heatmap of the activation and gradient matrix of a 256-layer tanh-activated GCN with VirgoFor initialization.

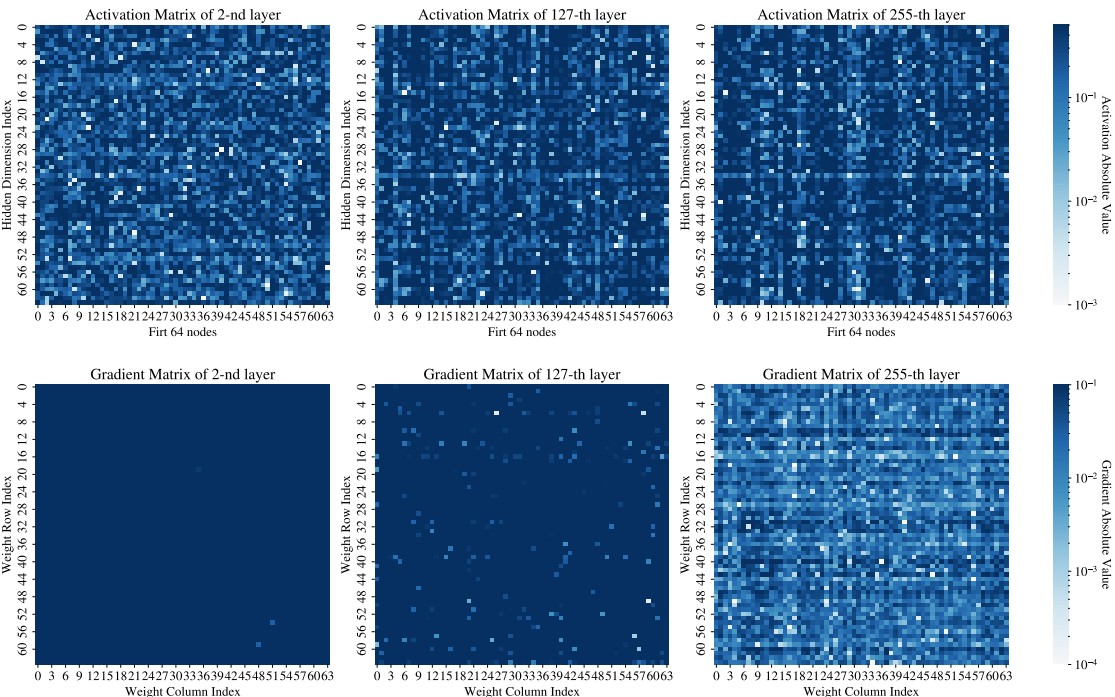

Figure 10: The heatmap of the activation and gradient matrix of a 256-layer tanh-activated GCN with VirgoBack initialization.

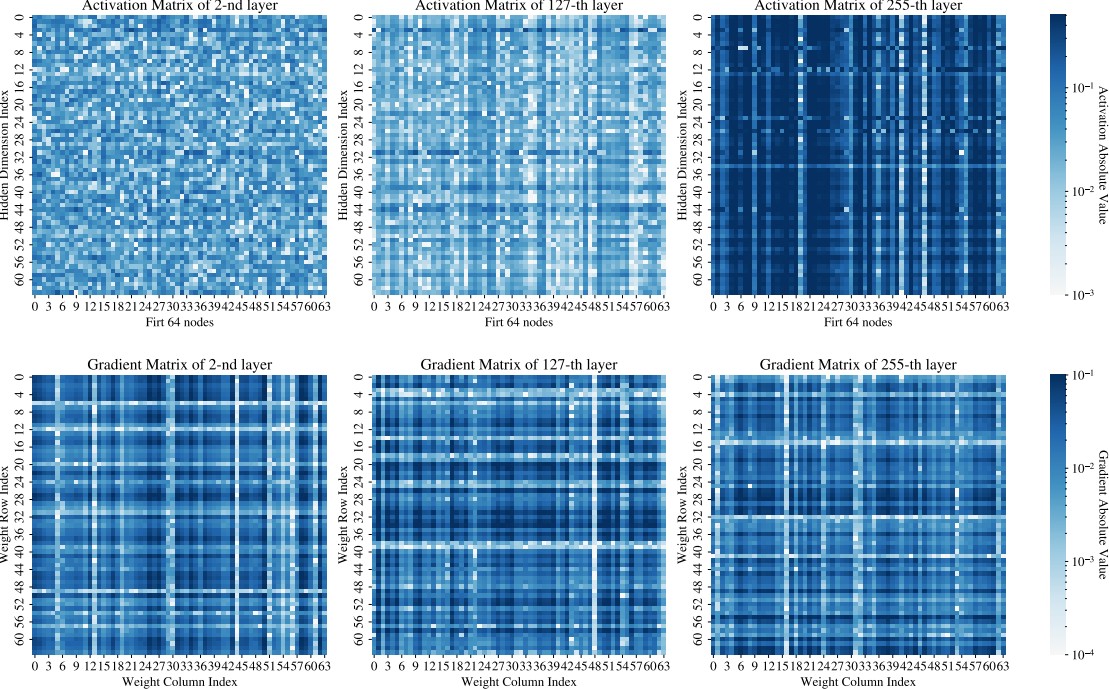

Figure 11: The heatmap of the activation and gradient matrix of a 256-layer tanh-activated GCN with SPoGInit.

### F.4 Computation cost and scalability of SPoGInit

We provide a detailed discussion on the computational cost and scalability of SPoGInit in this subsection.

**Computational cost analysis**

SPoGInit is utilized as a weight initialization search method prior to the main model training process (as described in Algorithm 1 in Appendix E). We identify two primary sources of additional computational cost: **(a) searching steps**: in practice, SpoGInit typically needs 20–40 search steps, which is small relative to the roughly 800-1500 training epochs of GNNs; **(b) optimization method**: SPoGInit employs a zeroth order optimization method, which only relies on only forward propagation computations, not requiring more computationally expensive backward passes. This analysis indicates that the computational overhead introduced by SPoGInit is relatively minor.

Besides, we conduct an empirical study to evaluate the computation overhead brought by SPoGInit. In our experiments, we compare the training time of a vanilla GCN architecture (without SPoGInit) against the same architecture incorporated with SPoGInit, using 40 searching steps, on both the Amazon-ratings and OGBN-Arxiv datasets. As shown in Table 16, SPoGInit consistently adds approximately 18% to the training time on these two datasets. In general, incorporating SPoGInit increases the overall training computational cost by roughly 10%–20%. Given the performance improvements observed in GNNs, we consider this additional cost to be an acceptable trade-off.

Table 16: Comparison of total training time for 64-layer tanh-activated GCN with Xaiver initialization and SPoGInit on the small dataset Amazon-ratings and the large dataset OGBN-Arxiv. All experiments are run for 1000 training epochs. The searching steps of SPoGInit are set as 40.

| Dataset | Nodes | Xavier | SPoGInit | Additional Time |
|---|---|---|---|---|
| Amazon-ratings | 24,492 | 185.7s | 220.5s | 18.7% |
| OGBN-Arxiv | 169,343 | 901.3s | 1064.1s | 18.1% |

**Scalability analysis**

Notably, Table 16 shows that as the graph size grows larger, the proportion of SPoGInit computation (relative to total training computation) would not increase. This implies that SPoGInit can scale effectively to large graph datasets.

### F.5 More experiments on graph attention networks

We conduct comparative experiments between SPoGInit and other baseline initialization methods on Graph Attention Networks (GAT) (Veličković et al., 2017). Building upon the original GAT architecture, we further investigate the performance on a residual-enhanced variant (ResGAT) by incorporating skip connections. The results are presented in Table 17.

The experimental results show that SPoGInit significantly mitigates performance degradation on GAT and ResGAT. Moreover, on ResGAT, while other baseline methods still suffer from severe performance drops as the network depth increases, SPoGInit instead leads to improved performance with greater depth, demonstrating its effectiveness in deep architectures.

### F.6 Experimental comparison with G-Init

This subsection presents a comparison between SPoGInit and G-Init (Kelesis et al., 2024), an initialization method specifically designed to alleviate over-smoothing in deep GNNs. G-Init incorporates the graph topology into the initialization process. Specifically, G-Init scales the weight variance of the base initialization by a factor of $d_i$.[4] Following the experimental setup in the original G-Init paper, we set $d_i = 1.6$ for

---

[4]While the original paper (Kelesis et al., 2024) adopts ReLU as the activation function—typically paired with Kaiming initialization—we use tanh activation, which is commonly initialized using Xavier.

Table 17: Test accuracies of GAT and ResGAT with varying depths and initialization methods. The bold figure highlights the best performance among different initializations. "Deg" refers to the test accuracy degradation as the depth increases from 4 to 64 layers. The smallest performance drops are highlighted in orange. We set the number of attention head to be 1. The results demonstrate that SPoGInit reduces performance degradation compared to baseline initializations and enhances the performance of deep GAT and ResGAT.

| Model | Init. | Cora | | | | | | Arxiv | | | | | |
|---|---|---|---|---|---|---|---|---|---|---|---|---|---|
| | | 4 | 8 | 16 | 32 | 64 | Deg. | 4 | 8 | 16 | 32 | 64 | Deg. |
| GAT | Conventional | 79.4 | 73.2 | 50.3 | 37.4 | 31.9 | ↓47.5 | 68.2 | 66.2 | 5.9 | 5.9 | 5.9 | ↓62.3 |
| | Xavier | **79.7** | **79.5** | 74.1 | 71.2 | 68.8 | ↓10.9 | 68.4 | 68.1 | 61.1 | 43.7 | 14.9 | ↓53.5 |
| | VirgoFor | 79.0 | 78.9 | 74.9 | 74.7 | 74.0 | ↓5.0 | **68.5** | **68.4** | 65.8 | 63.7 | **50.5** | ↓18.0 |
| | VirgoBack | 78.1 | 76.2 | 74.7 | 71.6 | 71.8 | ↓6.3 | **68.5** | **68.4** | 65.8 | 63.7 | 49.5 | ↓19.0 |
| | SPoGInit | 79.4 | 78.8 | **76.2** | **76.3** | **74.5** | ↓4.9 | 68.3 | 66.0 | **66.7** | **64.0** | 47.1 | ↓21.2 |
| ResGAT | Conventional | 78.3 | 78.5 | 78.1 | **77.5** | 32.5 | ↓45.8 | 70.6 | **71.4** | 71.8 | 69.7 | 21.6 | ↓49.0 |
| | Xavier | 77.6 | **79.3** | 77.8 | 32.3 | 32.0 | ↓45.6 | **70.8** | **71.4** | 69.7 | 21.7 | 21.7 | ↓49.1 |
| | VirgoFor | 78.4 | 77.9 | 75.4 | 31.3 | 31.5 | ↓46.9 | 70.5 | 70.5 | 21.8 | 21.6 | 21.7 | ↓48.8 |
| | VirgoBack | **79.4** | 77.6 | 76.1 | 26.1 | 33.2 | ↓46.2 | 70.7 | 70.4 | 21.6 | 21.6 | 16.3 | ↓54.5 |
| | SPoGInit | 77.1 | 77.4 | **78.6** | 76.0 | **78.3** | ↑1.2 | 70.7 | 71.3 | **71.9** | **71.8** | **71.4** | ↑0.7 |

OGBN-Arxiv and Arxiv-year, and $d_i = 2.0$ for all other benchmark datasets. We systematically evaluate the performance of tanh-activated GCN architectures ranging from 4 to 64 layers under different initialization schemes.

The results, summarized in Table 18, show that while G-Init can alleviate performance degradation in deep GCNs to some extent, SPoGInit consistently outperforms it in most cases. For instance, on the PubMed dataset, G-Init suffers from a 4% performance drop as depth increases, whereas SPoGInit achieves stable or improved performance. These findings highlight the effectiveness of SPoGInit's initialization strategy, which explicitly searches for stable signal propagation patterns, in mitigating depth-induced degradation in GCN training.

Table 18: Test accuracies of GCN models initialized with G-Init and SPoGInit across varying depths. The bold figure highlights the best performance among different initializations. "Deg" refers to the test accuracy degradation as the depth increases from 4 to 64 layers. We choose the starting point of SPoGInit as VirgoFor on the Arxiv dataset. The results show that while both initializations help alleviate degradation in deep GCNs, SPoGInit achieves more consistent performance and smaller accuracy drops across different datasets.

| Model | Init. | Cora | | | | | | Arxiv | | | | | |
|---|---|---|---|---|---|---|---|---|---|---|---|---|---|
| | | 4 | 8 | 16 | 32 | 64 | Deg. | 4 | 8 | 16 | 32 | 64 | Deg. |
| GCN | G-Init | 79.7 | 78.3 | 76.7 | 75.2 | 73.4 | ↓6.3 | **69.6** | 69.5 | **68.8** | 66.5 | 58.9 | ↓10.7 |
| | SPoGInit | **79.8** | **79.6** | **78.2** | **75.8** | **73.7** | ↓6.1 | **69.6** | **69.6** | 68.7 | **67.2** | **61.4** | ↓8.2 |

| Model | Init. | PubMed | | | | | | Arxiv-year | | | | | |
|---|---|---|---|---|---|---|---|---|---|---|---|---|---|
| | | 4 | 8 | 16 | 32 | 64 | Deg. | 4 | 8 | 16 | 32 | 64 | Deg. |
| GCN | G-init | **78.5** | 76.8 | **78.0** | 77.7 | 74.5 | ↓4.0 | 43.8 | 29.9 | **45.5** | **45.6** | 43.4 | ↓0.4 |
| | SPoGInit | 77.4 | **77.5** | 77.0 | **78.4** | **78.1** | ↑0.7 | **43.9** | **41.9** | 45.2 | 44.9 | **43.9** | 0 |

## G  Supplemental experiment results

### G.1  Experimental settings and hyperpameters

**Settings for the experiments on mainstream datasets.**

We set $w_1 = 1$, $w_2 = 10$, $w_3 = 1$ for the vanilla GCN and $w_1 = 1$, $w_2 = 1$, $w_3 = 1$ for other architectures in these experiments. An early stopping criterion is also implemented for SPoGInit: if the metric fails to decrease over $\delta$ consecutive steps, the search is terminated. Unless otherwise specified, SPoGInit is initialized with Xavier initialization in these experiments. To guarantee that models with the same depth have the same receptive field, we set the number of hops to 1 in the MixHop layer for the MixHop architecture.

In our experiments on the Cora and PubMed datasets,

- We perform grid searches over learning rates of 1e-3, 1e-4, 5e-5, and 1e-5.

- The training epochs and early stopping patience are listed in Table 19.

- For vanilla GCN and MixHop, we evaluate two configurations: (1) weight decay set to 5e-3 and dropout rate set to 0.5, and (2) both weight decay and dropout rate set to 0. We report the best performance achieved between these settings.

- For ResGCN and gatResGCN, we set the dropout rate to 0.5.

- In Table 2, the learning rate for SPoGInit is set to 0.1. The early stopping step $\delta$ is set to 20 for ResGCN and gatResGCN, and to 10 for vanilla GCN and MixHop. In the experiments with ResGCN and gatResGCN, SPoGInit starts with conventional initialization.

- In Figure 3, we use PyTorch's autograd functionality to compute the gradient of the scaling factors. In Figure 4 and 5, the learning rate for SPoGInit is set to 0.05 for ResGCN and MixHop, with the early stopping step $\delta$ set to 10 for ResGCN.

Table 19: Hyperparameter configurations of experiments on Cora and PubMed.

| GCN layers | training epoch | early stop patience |
|---|---|---|
| 4/8/16 layers | 800 | 200 |
| 32 layers | 1200 | 300 |
| 64 layers | 1500 | 375 |

In the experiments on the OGBN-Arxiv and Arxiv-year datasets,

- All models are trained for 1000 epochs.

- The learning rate is set to 5e-3 for ResGCN and gatResGCN, and 5e-4 for MixHop. For the vanilla GCN, the learning rates are configured as follows: 5e-3 for the 4-layer and 8-layer models, 5e-4 for the 16-layer and 32-layer models, and 5e-5 for the 64-layer model.

- For ResGCN and gatResGCN, we set the dropout rate to 0.5.

- For SPoGInit, the learning rate is set to 0.2 for ResGCN and gatResGCN, 0.1 for MixHop, 0.07 for vanilla GCN on the Arxiv-year dataset, and 0.05 for vanilla GCN on the OGBN-Arxiv dataset. The parameter $\delta$ in SPoGInit is set to 10 for ResGCN, gatResGCN, and MixHop, and 20 for vanilla GCN.

In the experiments on the Amazon-photo and Amazon-computers datasets,

- We perform grid searches over learning rates of 1e-3, 1e-4, 5e-5, and 1e-5.

- All model are trained with 64 hidden units.

- The training epochs and early stopping patience are listed in Table 19. One difference is that for 32-layer model, we used 1000 epochs for training and 200 epochs for early stopping patience.

- For ResGCN, gatResGCN and vanilla GCN, we set the dropout rate to 0.5. For vanilla GCN, we set the weight decay to be 0.00005. For MixHop, we tried two configurations: (1) dropout 0.5, weight decay 0.00005 (2) dropout 0, weight decay 0, and we choose the best performance amoug these configurations.

- We use the same configurations of SPoGInit in Cora and PubMed datasets. For the Amazon-computers dataset, the starting point is set as VirgoFor.

In the experiments on the Amazon-ratings and Roman-Empire datasets. We use VirgoFor as the starting point of SPoGInit in the experiments with vanilla GCN, and learning rate of SPoGInit is set to 0.1. The other configurations are the same with those in the OGBN-Arxiv and Arxiv-year datasets.

In the experiments with DGN,

- In the Cora and PubMed datasets, we perform two configurations: (1) following the setting in Zhou et al. (2020b), the dropout as 0.6, weight decay is set as 0.0005. (2) dropout and weight decay are both set to be zero. We report the best performance amoug these two configureations. The other configurations are the same with those in Cora and PubMed experiments.

- In the OGBN-Arixv dataset, we use the vanilla GCN with Conventional Initialization as base model. The other configurations are the same with those in OGBN-Arxiv experiments.

- For the group number $G$ in DGN, we set it as 10 in Cora, and 5 for other datasets.

In the experiments with CO-GNN,

- To align with the experiment settings in Section 5, we fix the width of environment network to be 64, and we set the width of action network to be 16 in Cora and PubMed dataset, and 32 in the Amazon-ratings. And we use mean GNN as the action network.

- In the Cora and PubMed datasets, we follow the settings in Finkelshtein et al. (2023): We perform grid searches over learning rates of 0.01, 0.05, 0.005, 0.0005. The dropout is set as 0.5 and weight decay is set as 5e-4, and $\tau_0$ is set as 0.1. We use mean GNN as the environment network in Cora and GCN as the environment network in PubMed. We train the model for 1000 epochs and we adopt the skip connection in the model. The other configurations are the same with those in Cora and PubMed experiments in our setting.

- In the Amazon-ratings datasets, we follow the settings in Finkelshtein et al. (2023): we train the model for 3000 epochs, the learning rate is set as 3e-4. And we use mean GNN as the environment network. We adopt the layernorm and skip connection.

**Settings for the exploration of starting points in SPoGInit**

In these experiments, the settings for SPoGInit are slightly different from the previous ones. To be more specific,

- For ResGCN, the learning rate of SPoGInit is set to 0.1 when starting from VirgoFor or VirgoBack initialization, and 0.2 in all other cases.

- For vanilla GCN, the learning rate of SPoGInit is set to 0.1 when starting from Conventional initialization and 0.05 in all other cases.

**Settings for the missing feature experiments**

To enhance the dataset's dependence on long-range relationships, we reduce the proportion of training split and set the train/validation/test split as 10%/25%/65%, which is generated through the implementation in Lim et al. (2021). In this experiment, we set SPoGInit to use Conventional initialization as the starting point when it is adopted to gatResGCN, while SPoGInit begins with Xavier initialization as the starting point in other cases.

**Settings for MILP experiment**

In the MILP experiment, we train the models for 500 epochs. The batch size is set as 40 and the hidden dimension is set as 64.

## G.2   Ablation study of SPoGInit

In this subsection, we conduct the ablation study of SPoGInit by analyzing various combinations of its SP metric components. Specifically, we reformulate the optimization problem (2) to include either one or two of the three SP metrics. Table 20 presents the performance of a 32-layer vanilla GCN with these modified SPoGInit variants on the Cora dataset. For this experiment, both dropout rate and weight decay are set to 0, while all other hyperparameters follow the settings for the experiments on mainstream datasets in Appendix G.1.

The results indicate that incorporating a single SP metric into the optimization problem slightly improves the performance of vanilla GCNs, with the most notable enhancement observed when SPoGInit includes the FSP metric. Incorporating two SP metrics leads to a more substantial improvement, although it still falls short of the performance achieved by SPoGInit using all three SP measures.

These experimental results highlight the importance of combining all three SP metrics to maximize the performance of SPoGInit.

Table 20:   Test accuracies of a 32-layer vanilla GCN with modified SPoGInit variants: A "✓" indicates that a specific metric is included in the signal propagation optimization of SPoGInit, while a "-" denotes its exclusion. The results show that incorporating one or two SP metrics improves the performance of the deep vanilla GCN. However, the best performance is achieved when all three SP metrics are used in SPoGInit.

| Variants | FSP | BSP | GEV | Test Accuracy |
|---|---|---|---|---|
| GCN (Xavier) | | | | $72.83 \pm 0.45$ |
| +SPoGInit | | | | |
| | ✓ | - | - | $73.83 \pm 1.72$ |
| | - | ✓ | - | $73.50 \pm 0.36$ |
| | - | - | ✓ | $73.90 \pm 1.56$ |
| | ✓ | ✓ | - | $75.60 \pm 0.16$ |
| | ✓ | - | ✓ | $73.60 \pm 1.72$ |
| | - | ✓ | ✓ | $73.77 \pm 1.43$ |
| | ✓ | ✓ | ✓ | $\mathbf{75.80 \pm 0.16}$ |

## H   Broader impact statement

In this paper, we employ signal propagation theory to analyze the performance degradations in deep GCNs. Additionally, we propose a solution (SPoGInit) to address signal propagation issues and alleviate this problem. This paper is a theoretical and algorithmic paper on graph neural nets, and does not seem to pose negative social impact.

