# OpenReview forum: "Exploring and Improving Initialization for Deep Graph Neural Networks: A Signal Propagation Perspective"
_TMLR — Accepted by TMLR_

### Review · Reviewer_nwvu · 2025-02-09

**Summary Of Contributions:**

The authors attempt to address the performance degradation of deep GCNs from the lens of signal propagation (SP). Both theoretical and empirical findings underscore the importance of stabilizing three SP metrics proposed in this paper for boosting the performance of deep GCNs. As far as I know, this is a completely new perspective. Although some of the theories are obvious, it is a good contribution to unifying them into a new framework that makes sense.

**Audience:**

Yes

**Broader Impact Concerns:**

N/A.

**Claims And Evidence:**

Yes

**Requested Changes:**

See weaknesses above.

**Strengths And Weaknesses:**

There are no major problems with the article as a whole. Here are just some suggestions to improve the presentation of the article.

> Some presentation formats need improvement

- 'Kaiming (usually for ReLU)' in section 2.2, should $W_{ij}^{(l)}$ be $W_{kk'}^{(l)}$?
- All figures are recommended to be redrawn. Multiple sub-figures should be merged directly when drawing with matplotlib, and the legend should be shared and placed uniformly in the upper right area. In addition, it is recommended to change the font or use the matplotlib2tikz package to export the tikz drawing.
- It is recommended to use a restatable environment to restate the theorem when supplementing the proof in the appendix to facilitate reading.

> Recommended additional references

- Han J, Huang W, Rong Y, et al. Structure-Aware DropEdge Toward Deep Graph Convolutional Networks[J]. IEEE Transactions on Neural Networks and Learning Systems, 2023.

- Fang T, Xiao Z, Wang C, et al. Dropmessage: Unifying random dropping for graph neural networks[C]//Proceedings of the AAAI Conference on Artificial Intelligence. 2023, 37(4): 4267-4275.

---

> ### Author Response · Authors · 2025-02-25
> **Response to Reviewer nwvu**
>
> We thank the reviewer for the constructive suggestions. Below, we detail the revisions made in response to your comments:
>
> - **Typo in Section 2.2:**
>     The notation issue has been corrected as suggested.
>
> - **Figures Redrawing:**
>     We have redrawn all the figures in the manuscript. Specifically, multiple sub-figures have been merged into single comprehensive figures. After careful consideration, we have placed the legends at the top center, which we believe offers better visual consistency and clarity compared to the upper right. Moreover, we have refined the fonts in the figures. We hope these adjustments are acceptable and we are happy to make further modifications based on the reviewer’s preference.
>
> - **Theorem Restatement:**
>     A restatement of the theorems have been added at the beginning of the proofs in the appendix sections to enhance clarity and ease of reading.
>
> - **Additional References:**
>     The recommended additional references have been incorporated into the Related Works section (Section 6).
>
>
> We hope these revisions effectively address your concerns and improve the overall presentation of our work. Thank you again for your insightful feedback.

---

### Review · Reviewer_zcM9 · 2025-02-28

**Summary Of Contributions:**

This paper introduces a novel initialization method, SPoGInit, for deep Graph Convolutional Networks (GCNs) by leveraging signal propagation theory to address performance degradation caused by over-smoothing and unstable signal propagation in deeper networks.

**Audience:**

Yes

**Claims And Evidence:**

Yes

**Requested Changes:**

1. **Computational Cost and Scalability**
   Include a discussion on the computational cost and scalability of SPoGInit, especially when applied to large graphs.

2. **Experimental Evaluation with Diverse Datasets**
   Extend the experimental evaluation to include more diverse datasets, such as:
   - Amazon (https://arxiv.org/abs/1811.05868)
   - Datasets with non-homophilic graphs
   This would help demonstrate the broader applicability of SPoGInit.

3. **Comparison with State-of-the-Art Methods**
   Provide a more thorough comparison with state-of-the-art methods aimed at mitigating over-smoothing and improving the performance of deep GCNs. This could include:
   - **Normalization Methods**:
     - DGN: Towards Deeper Graph Neural Networks with Differentiable Group Normalization (NIPS 2022) - [Link](https://arxiv.org/abs/2006.06972)
   - **Architectural Modifications**:
     - Layer-diverse Negative Sampling for Graph Neural Networks (TMLR 2024) - [Link](https://openreview.net/forum?id=WOrdoKbxh6)
   - **Drop Edges**:
     - Cooperative Graph Neural Networks (ICML 2024) - [Link](https://arxiv.org/abs/2310.01267)

4. **Clarification of SPoGInit's Limitations**
   Clarify the limitations of SPoGInit, particularly in scenarios where the method may not perform as expected. Additionally, suggest potential future work to address these limitations.

5. **Visualization with Heatmaps**
   A heatmap to visualize different initializations would be beneficial for providing an intuitive understanding of the differences between methods.

**Strengths And Weaknesses:**

**Strengths**
1. The theoretical analysis is rigorous, providing clear insights into why traditional initialization methods fail in deep GCNs.
2. The proposed SPoGInit method is empirically validated across multiple datasets and architectures, demonstrating significant improvements in performance, especially for tasks involving long-range relationships.
3. The paper is well-structured, with a comprehensive set of experiments and ablation studies that highlight the effectiveness of the proposed approach.

Weaknesses:
1. The paper lacks a deeper discussion on the computational overhead of SPoGInit, particularly in comparison to traditional initialization methods, which could be a concern for large-scale applications.
2. While the method is tested on several datasets (4 datasets), the evaluation could be extended to more diverse and real-world graph datasets to further validate its generalizability.
3. The paper could benefit from a more detailed comparison with other recent techniques addressing over-smoothing and deep GCN performance, such as normalization methods or architectural modifications.

---

> ### Author Response · Authors · 2025-03-24
> **Part 1: Computational Cost and Scalability**
>
> **Comment:** *"Include a discussion on the computational cost and scalability of SPoGInit, especially when applied to large graphs."*
>
> **Response:**
>
> We appreciate the reviewer's insightful comments and suggestions. Below, we provide a detailed discussion on the computational cost and scalability of SPoGInit.
>
> 1. **Computational Cost**
>
> (a). **Cost Breakdown:**
> SPoGInit is a weight initialization search method performed before the main model training process (as outlined in Algorithm 1 in Appendix E). The primary sources of additional computational cost are:
>    - **Searching Steps:** In practice, SpoGInit typically needs 20–40 search steps, which is small relative to the typical 800-1500 training epochs of GNNs.
>    - **Optimization Method:** SPoGInit employs a zeroth order optimization method, which only relies on forward propagation computations, not requiring more computationally expensive backward passes.
>
> This analysis suggests that the additional computational overhead of SPoGInit is modest.
>
> (b). **Empirical Overhead:**
> Besides, we conduct an empirical study to evaluate the computation overhead brought by SPoGInit. In our experiments, we compare the training time of a vanilla GCN architecture (without SPoGInit) against the same architecture incorporated with SPoGInit, using 40 searching steps, on both the Amazon-Ratings and OGBN-Arxiv datasets. As shown in Table 1, SPoGInit consistently adds approximately 18\% to the training time on these two datasets. In general, incorporating SPoGInit increases the overall training computational cost by roughly 10\%–20\%. We consider this overhead acceptable given the significant performance improvements observed in the GNN.
>
> *Table 1: Comparison of total training time for 64-layer tanh-activated GCN with Xavier initialization and SPoGInit on the small dataset Amazon-Ratings and the large dataset OGBN-Arxiv. All experiments are run for 1000 training epochs. The searching steps of SPoGInit are set as 40.*
>
> | **Dataset**   | **Nodes** | **Xavier** | **SPoGInit** | **Additional Time** |
> |---------------|-----------|------------|--------------|------------------------------|
> | Amazon-Ratings          | 24,492     | 185.7s     | 220.5s       | 18.7%                        |
> | OGBN-Arxiv    | 169,343   | 901.3s     | 1064.1s      | 18.1%                        |
>
> 2. **Scalability**
> Notably, Table 1 above shows that as the graph size grows larger, the proportion of SPoGInit computation (relative to total training computation) would not increase. This implies that SPoGInit can scale effectively to large graph datasets.
>
> To maintain the coherence of the paper, we have incorporated the above discussion into Appendix F.4 of the revised manuscript.

---

> ### Author Response · Authors · 2025-03-24
> **Part 2: Experimental Evaluation with Diverse Datasets**
>
> **Comment:** *"Extend the experimental evaluation to include more diverse datasets, such as:*
> - *Amazon (https://arxiv.org/abs/1811.05868)*
> - *Datasets with non-homophilic graphs*
>
> *This would help demonstrate the broader applicability of SPoGInit."*
>
> **Response:**
>
> We thank the reviewer for the suggestion. In response, we extend our experimental evaluation (see Appendix F.1) to include additional datasets—both homophilic (Amazon-photo and Amazon-computers) and non-homophilic (Amazon-ratings and Roman-empire). On these datasets, our results demonstrate that SPoGInit reduces performance degradation in deep GCN models and, in some cases, improves accuracy with increased depth. These improvements are consistent across various architectures, confirming the robustness and broader applicability of SPoGInit. Detailed experimental setups, dataset statistics, and performance comparisons can be found in Appendix F.1.

---

> ### Author Response · Authors · 2025-03-24
> **Part 3: Comparison with State-of-the-Art Methods**
>
> **Comment:** *Provide a more thorough comparison with state-of-the-art methods aimed at mitigating over-smoothing and improving the performance of deep GCNs. This could include:*
> - **Normalization Methods:**
> *DGN: Towards Deeper Graph Neural Networks with Differentiable Group Normalization (NIPS 2022)*
> - **Architectural Modifications:**
> *Layer-diverse Negative Sampling for Graph Neural Networks (TMLR 2024)*
> - **Drop Edges:**
> *Cooperative Graph Neural Networks (ICML 2024)*
>
>
> **Response:**
>
> We thank the reviewer for the valuable suggestions. In Appendix F.2, we have now added experiments that compare and combine SPoGInit with both DGN and CO-GNN. These additional results clearly demonstrate that SPoGInit not only outperforms these methods in reducing performance degradation in deep GCNs, but it also integrates effectively with them to further enhance network's performance.
>
> Regarding the comparison with *Layer-diverse Negative Sampling*, we have carefully examined the official implementation provided by the authors, which currently consists of a single Jupyter notebook. We noticed that the code only supports CPU execution and cannot directly run on GPUs, thus posing significant scalability issues and long runtimes when applied to large-scale graph datasets. Given these practical constraints, conducting a fair and thorough experimental comparison with this method is challenging. Therefore, we temporarily do not include this baseline from our experiments. However, if the reviewer has suggestions or guidance on this matter, we would greatly appreciate their input.

---

> ### Author Response · Authors · 2025-03-24
> **Part 4: Clarification of SPoGInit's Limitations**
>
> **Comment:** *Clarify the limitations of SPoGInit, particularly in scenarios where the method may not perform as expected. Additionally, suggest potential future work to address these limitations.*
>
> **Response:**
>
> We thank the reviewer for the insightful suggestions. We agree that a discussion of SPoGInit's limitations and potential future work is valuable. Below, we provide a more detailed clarification of the limitations and possible extensions:
>
> **Task-Specific Design:**
> SPoGInit has been specifically designed and tested for node classification tasks within graph neural networks (GNNs). While the method has shown promising results in this context, its applicability to other tasks, such as edge prediction or graph classification, remains unexplored. Adapting SPoGInit for a wider range of graph-based learning tasks is as an important direction for future work.
>
> **Architectural Compatibility:**
> Although SPoGInit is designed as a general method for GNNs, our current implementation is not entirely plug-and-play for all GNN architectures. The integration of SPoGInit into novel or non-standard GCN variants (e.g., GCNII [1], GraphTransformer [2]) will require additional engineering efforts. Future work could focus on creating more robust and flexible integration methods for a broader range of GNN models, enhancing SPoGInit’s adaptability.
>
> **Computational Overhead:**
> As highlighted in our analysis, SPoGInit introduces an approximate 10%~20% computational overhead. While we believe this overhead is acceptable in many practical scenarios, it could become a limiting factor in situations where computational resources are severely constrained. Future work could explore methods to reduce the computational cost of SPoGInit, either through more efficient optimization techniques or by leveraging hardware acceleration.
>
> To maintain the coherence of the paper, we have incorporated the above discussion into Section 7 of the revised manuscript.
>
> **Reference:**
>
> [1] Chen, Ming, et al. "Simple and deep graph convolutional networks." International conference on machine learning. PMLR, 2020.
>
> [2] Shehzad, Ahsan, et al. "Graph transformers: A survey." arXiv preprint arXiv:2407.09777 (2024).

---

> ### Author Response · Authors · 2025-03-24
> **Part 5: Visualization with Heatmaps**
>
> **Comment:** *A heatmap to visualize different initializations would be beneficial for providing an intuitive understanding of the differences between methods.*
>
> **Response:**
>
> We thank the reviewer for the helpful suggestion. In Appendix F.3, we have now added heatmap visualizations to intuitively compare different initialization methods. These heatmaps demonstrate that while Conventional and Xavier initializations suffer from forward and backward signal vanishing, and VirgoFor/VirgoBack exhibit gradient explosion issues in shallow layers, our proposed SPoGInit consistently maintains stable activations and gradients across all layers. We believe these visualizations provide valuable insights, allowing readers to better understand the differences and characteristics among the compared methods. We thank the reviewer again for the valuable suggestion, which helps us further improve the clarity and intuitiveness of our presentation.

---

### Review · Reviewer_u83j · 2025-03-19

**Summary Of Contributions:**

The paper makes the following contributions:

- **Signal Propagation Framework:** It introduces a framework for analyzing signal propagation in deep GCNs through three key metrics—forward signal propagation (FSP), backward signal propagation (BSP), and graph embedding variation (GEV). This framework provides insight into the issues of oversmoothing and gradient instability in deep GCNs.

- **SPoGInit Initialization Method:** Building on the SP analysis, the authors propose SPoGInit, an initialization scheme that adaptively adjusts layer-wise weight variances by solving an optimization problem aimed at stabilizing the three SP metrics, mainly in the context of deep GCNs.

- **Theoretical Analysis:** The paper offers a theoretical examination under the Neural Network Gaussian Process (NNGP) approximation, demonstrating that traditional initialization methods fail to control all three SP metrics simultaneously as the network depth increases.

These contributions collectively advance the understanding of initialization in deep GNNs and propose a solution that enhances the training of deeper architectures.

**Audience:**

Yes

**Claims And Evidence:**

Yes

**Requested Changes:**

Please kindly refer to the Weaknesses.

**Strengths And Weaknesses:**

**Strengths**

- Introduces a novel signal propagation framework with three metrics (FSP, BSP, and GEV) that address oversmoothing and gradient issues in deep GCNs.
- Proposes the SPoGInit initialization method, which adapts layer-wise weight variances and shows promising improvements in performance as network depth increases.
- Provides both theoretical analysis (using the NNGP approximation) and extensive empirical validation across multiple architectures and datasets.

**Weaknesses**

- The analysis relies on an infinite-width assumption, which raises concerns about its applicability to practical finite-width networks and its relationship with GNTK.
- There is insufficient discussion on computational efficiency, including time and hardware requirements compared to other initialization methods.
- Key experimental parameters (e.g., the range of $V_{FSP}$ and $V_{BSP}$ values and the emphasis on weight factors like $w_2$) are not thoroughly justified.
- The added value of incorporating the GEV metric appears marginal relative to the extra computational cost.
- Inclusion of additional datasets and models (e.g., graph classification tasks, attentive models like GAT), and recent GNN initialization methods such as Jaiswal et al., Kelesis et al. Han et al. (or discuss them in the related work) would provide a fairer and more comprehensive evaluation.
- Comparison of SpoGInit with other methods that use different activation functions rises concerns about the source of the observed improvements. Are these improvements attributed to the proposed method or to the use of different activation function?
- Including the resulting values of $\sigma_i$ for each layer would shed more light on what values does the proposed method learn and if and how they differ from other initialization methods.

---

> ### Author Response · Authors · 2025-04-11
> **Part 1: Infinite‑Width Assumption**
>
> **Comment:** *The analysis relies on an infinite-width assumption, which raises concerns about its applicability to practical finite-width networks and its relationship with GNTK.*
>
> **Response:**
>
> Thanks for your comments. Below are our responses.
>
> **Applicability to practical finite-width networks**
> Our analysis relies on an infinite-width assumption—a common approach in traditional theoretical analyses of neural networks (see, e.g., [1][2][3]). This assumption offers the primary benefit of ensuring that the feature embeddings at each layer follow a Gaussian distribution under random initialization, thereby simplifying the theoretical treatment. Such a Gaussian property is not strictly guaranteed for practical, finite-width networks.
>
> Nonetheless, we empirically demonstrate that the embeddings of a finite-width network are approximately Gaussian, implying that the infinite-width approximation remains reasonable in practice. Specifically, we randomly selected three node embeddings from the final convolutional layer of a 4-layer, 64-width GCN network trained on the OGBN-Arxiv dataset. We then applied the Kolmogorov-Smirnov hypothesis test to evaluate their distribution. In this test, a p-value below 0.05 would indicate a significant deviation from a Gaussian distribution. The observed p-values—0.75, 0.90, and 0.73—suggest that the embeddings closely follow a Gaussian distribution. Thus, despite the theoretical limitation, our empirical findings support the validity of using the infinite-width approximation.
>
> [1] Lee, Jaehoon, et al. "Deep neural networks as gaussian processes." arXiv preprint arXiv:1711.00165 (2017).
>
> [2] Sohl-Dickstein, Jascha, et al. "On the infinite width limit of neural networks with a standard parameterization." arXiv preprint arXiv:2001.07301 (2020).
>
> [3] Yang, Greg. "Scaling limits of wide neural networks with weight sharing: Gaussian process behavior, gradient independence, and neural tangent kernel derivation." arXiv preprint arXiv:1902.04760 (2019).
>
>
> **Relationship with GNTK**
>
> NNGP (Neural Network Gaussian Process) for GNNs and GNTK (Graph Neural Tangent Kernel) are two theoretical tools used to analyze the behavior of graph neural networks. While they share the common goal of characterizing GNNs in the infinite-width limit, they differ in focus: NNGP primarily describes the properties of feature embeddings at initialization, whereas GNTK captures the training dynamics of GNNs, particularly how the model converges under gradient descent.
>
> Our work focuses on improving GNN initialization. As a future direction, it would be interesting to explore how GNTK can be leveraged to optimize the training process of GNNs.
>
> **We have incorporated these discussions into Section 6 (Related Work) of the revised manuscript.**

---

> ### Author Response · Authors · 2025-04-11
> **Part 2: Discussion on Computational Efficiency**
>
> **Comment:** *There is insufficient discussion on computational efficiency, including time and hardware requirements compared to other initialization methods.*
>
>
> **Response:**
>
> We appreciate the reviewer's comments and suggestions. Below are our responses.
>
> **(A). Computation time:** (We note that this part overlaps with our reply to Reviewer zcM9.)
>
> **1). Cost Breakdown:**
> SPoGInit is a weight initialization search method performed before the main model training process (as outlined in Algorithm 1 in Appendix E). The primary sources of additional computational cost are:
>    - **Searching Steps:** In practice, SpoGInit typically needs 20–40 search steps, which is small relative to the typical 800-1500 training epochs of GNNs.
>    - **Optimization Method:** SPoGInit employs a zeroth order optimization method, which only relies on forward propagation computations, not requiring more computationally expensive backward passes.
>
> This analysis suggests that the additional computational overhead of SPoGInit is modest.
>
> **2). Empirical Overhead:**
> Besides, we conduct an empirical study to evaluate the computation overhead brought by SPoGInit. In our experiments, we compare the training time of a vanilla GCN architecture (without SPoGInit) against the same architecture incorporated with SPoGInit, using 40 searching steps, on both the Amazon-Ratings and OGBN-Arxiv datasets. As shown in Table 1, SPoGInit consistently adds approximately 18\% to the training time on these two datasets. In general, incorporating SPoGInit increases the overall training computational cost by roughly 10\%–20\%. We consider this overhead acceptable given the significant performance improvements observed in the GNN.
>
> *Table 1: Comparison of total training time for 64-layer tanh-activated GCN with Xavier initialization and SPoGInit on the small dataset Cora and the large dataset OGBN-Arxiv. All experiments are run for 1000 training epochs. The searching steps of SPoGInit are set as 40.*
>
> | **Dataset**   | **Nodes** | **Xavier** | **SPoGInit** | **Additional Time** |
> |---------------|-----------|------------|--------------|------------------------------|
> | Amazon-Ratings          | 24,492     | 185.7s     | 220.5s       | 18.7%                        |
> | OGBN-Arxiv    | 169,343   | 901.3s     | 1064.1s      | 18.1%                        |
>
> The other initialization baseline methods do not involve weight searching prior to network training and therefore incur negligible additional computation time. However, considering the overall performance improvement, we still recommend using SPoGInit despite the extra computational cost.
>
>
> **Remark 1 (Scalability):** Notably, Table 1 above shows that as the graph size grows larger, the proportion of SPoGInit computation (relative to total training computation) would not increase. This implies that SPoGInit can scale effectively to large graph datasets.
>
> To maintain the coherence of the paper, we have incorporated the above discussion into Appendix F.4 of the revised manuscript.
>
> **Remark 2:** In our previous response to Reviewer zcM9, we mistakenly listed the datasets in Table 1 as *Cora* and *OGBN-Arxiv*. The correct datasets are *Amazon-Ratings* and *OGBN-Arxiv*. We have now updated both responses to reflect the correct information.
>
>
> **(B). Hardware cost**
>
> As described in Appendix E, the objective function of SPoGInit only relies on the network's feature embeddings and gradients, which are already computed during the forward and backward propagation in training. Therefore, SPoGInit does not incur extra memory overhead.

---

> ### Author Response · Authors · 2025-04-11
> **Part 3: Justification of Key Experimental Parameters (e.g., Range of V_FSP and V_BSP, Weight Factor $w_2$)**
>
> **Comment:** *Key experimental parameters (e.g., the range of $V_{FSP}$ and $V_{BSP}$ values and the emphasis on weight factors like $w_2$) are not thoroughly justified.*
>
> **Response:**
>
> Thanks for your comment. We clarify that $V_{FSP}$ and $V_{BSP}$ are not tunable parameters but rather two variable terms within the objective function of SPoGInit. For details, please refer to Equation (49) in Appendix E.
>
> We analyze the effect of the coefficient $w_2$ in SPoGInit, which controls the contribution of the backward signal propagation (BSP) term $\hat{V_{BSP}}$ in the objective function. Empirically, we observe that the initial value of $\hat{V_{BSP}}$ is substantially lower than that of $\hat{V_{FSP}}$, leading to a potential imbalance between the two objectives. To mitigate this discrepancy, in practice, we set a larger value for $w_2$ than $w_1$ in order to amplify the role of $\hat{V_{BSP}}$ during optimization.
>
> We conduct experiments on a 32-layer GCN with tanh activation, trained on the Cora dataset. Table 1 reports the values of $\hat{V_{FSP}}$ and $\hat{V_{BSP}}$ under different settings of $w_2 \in \{1,10,100\}$. We fix the learning rate at 0.1, use 100 optimization iterations, and set $w_1 = w_3 = 1$. We report the average result over 20 runs. The results reveal two key trends regarding the impact of $w_2$:
>
> - **Minimum $\hat{V}_{BSP}$ at $w_2 = 10$**: As $w_2$ increases from 1 to 10, $\hat{V_{BSP}}$ decreases significantly, indicating improved backward signal propagation. However, further increasing $w_2$ to 100 leads to a slight increase in $\hat{V_{BSP}}$, suggesting that excessive emphasis on the BSP term may destabilize its optimization.
> - **Forward signal propagation deteriorates at large $w_2$**: While $\hat{V}_{FSP}$ remains stable between $w_2 = 1$ and $w_2 = 10$, it increases sharply when $w_2$ reaches 100, indicating that an overly large $w_2$ harms forward signal preservation.
>
> These findings suggest that $w_2 = 10$ achieves a favorable trade-off between forward and backward signal preservation, yielding low values for both $\hat{V_{FSP}}$ and $\hat{V_{BSP}}$. Based on this analysis, we adopt $w_2 = 10$ as the default setting in the subsequent experiments.
>
> Table 1: The $\hat{V_{FSP}}$ and $\hat{V}_{BSP}$ of 32-layer tanh-activated GCN using SPoGInit with different $w_2$ on Cora dataset.
>
> |                       | **w/o SPoG** | **SPoG using $w_2=1$** | **SPoG using $w_2=10$** | **SPoG using $w_2=100$** |
> |-----------------------|--------------|-------------------------|--------------------------|---------------------------|
> | $\hat{V}_{FSP}$       | 31.2         | $4.1 \times 10^{-3}$    | $4.4 \times 10^{-3}$     | $6.8 \times 10^{-2}$      |
> | $\hat{V}_{BSP}$       | $1.3 \times 10^{-1}$ | $5.8 \times 10^{-3}$    | $6 \times 10^{-4}$       | $1 \times 10^{-3}$        |
>
>
> The above analysis has been included in Appendix E.2 of the revised manuscript.

---

> ### Author Response · Authors · 2025-04-11
> **Part 4: Incorporating the GEV Metric**
>
> **Comment:** *The added value of incorporating the GEV metric appears marginal relative to the extra computational cost.*
>
> **Response:**
> We thank the reviewer for raising this concern. We would like to clarify that the additional cost introduced by adding the GEV term to SPoGInit’s objective is minimal. Therefore, we believe the performance gains justify the inclusion of GEV.
>
> **Technical explanation:**
>
> As defined in Appendix E, the GEV metric is computed as
> $$
> \hat{M}^{(L)}_{GEV} := \frac{Dir(H^{(L)})}{\Vert H^{(L)} \Vert^2_F},
> $$
> where the Dirichlet energy is given by $Dir(H) = \text{tr}(H^\top \hat{L} H)$, and $\hat{L}$ denotes the normalized Laplacian of the input graph $\mathcal{G}$ (see Appendix B for mathematical details). Importantly, the GEV metric depends solely on the final-layer node embeddings $H^{(L)}$, which are computed in a single forward pass of the network. The additional cost of computing the norm and Dirichlet energy is negligible compared to the forward pass itself.
>
> We note that the computation of FSP metric already requires a forward pass. Thus, the final-layer embeddings for GEV computation can be directly obtained from this same forward pass, meaning no additional computation is needed.
>
> **Empirical evaluation:**
>
> To evaluate the computational efficiency of incorporating GEV, we conduct experiments using a 64-layer GCN with tanh activation on the Cora and PubMed datasets.
> As shown in Table 1, the runtime of the SPoGInit algorithm with GEV is only slightly higher than that without GEV, with an increase of just about 1\%–3\%. In practice, this additional cost is minimal and acceptable, given the consistent performance improvements observed when GEV is used alongside the FSP and BSP metrics.
>
> *Table 1: Initialization time (in seconds) of a 64-layer tanh-activated GCN using SPoGInit with and without GEV on the Cora and PubMed datasets. For each case, the total number of initialization iterations is set to 40, and the results are averaged over 3 runs.*
>
> |                                  | **Cora** | **PubMed** |
> |----------------------------------|----------|------------|
> | SPoGInit without GEV             | 6.46     | 9.90       |
> | SPoGInit with GEV                | 6.66     | 10.0       |
> | Relative Overhead of GEV         | 3.1%     | 1.1%       |
>
>
> In summary, since the GEV computation utilizes existing outputs from the forward pass with minimal overhead, its integration into SPoGInit’s objective is both computationally efficient and beneficial in practice.
>
> The above analysis has been included in Appendix E.3 of the revised manuscript.

---

> ### Author Response · Authors · 2025-04-11
> **Part 5: Additional Datasets, models, and GNN initialization methods**
>
> **Comment:** *Inclusion of additional datasets and models (e.g., graph classification tasks, attentive models like GAT), and recent GNN initialization methods such as Jaiswal et al., Kelesis et al. Han et al. (or discuss them in the related work) would provide a fairer and more comprehensive evaluation.*
>
> **Response:**
>
> We thank the reviewer for the suggestion. Below are our responses.
>
> **Additional Datasets:** (We note that this part overlaps with our reply to Reviewer zcM9.)
>
> We extend our experimental evaluation (see Appendix F.1) to include additional datasets—both homophilic (Amazon-photo and Amazon-computers) and non-homophilic (Amazon-ratings and Roman-empire). On these datasets, our results demonstrate that SPoGInit reduces performance degradation in deep GCN models and, in some cases, improves accuracy with increased depth. These improvements are consistent across various architectures, confirming the robustness and broader applicability of SPoGInit. Detailed experimental setups, dataset statistics, and performance comparisons can be found in Appendix F.1.
>
> Although the reviewer suggests including graph classification tasks, we note that our method is specifically designed for node classification tasks. Extending SPoGInit to graph-level tasks involves additional considerations and is a promising direction for future work.
>
> **Additional Models:**
>
> We conduct comparative experiments between SPoGInit and other baseline initialization methods on Graph Attention Networks (GAT). Building upon the original GAT architecture, we further investigate the performance on a residual-enhanced variant (ResGAT) by incorporating skip connections.
>
> The experimental results show that SPoGInit significantly mitigates performance degradation on GAT and ResGAT. Moreover, on ResGAT, while other baseline methods still suffer from severe performance drops as the network depth increases, SPoGInit instead leads to improved performance with greater depth, demonstrating its effectiveness in deep architectures.
>
> Detailed results and analyses are included in Appendix F.5 of the revised manuscript.
>
> **Recent GNN initialization methods:**
>
> **Kelesis et al. [1]** propose G-Init, which incorporates the graph topology into the initialization process. We include in Appendix F.6 a comparison experiment between G-Init and SPoGInit on vanilla GCN. The results show that while both G-Init and SPoGInit alleviate performance degradation in deep GCNs, SPoGInit provides more consistent results and smaller accuracy drops across datasets.
>
> **Jaiswal et al. [2]** also approach initialization from a signal propagation perspective. For the forward pass, they focus on Topology-Aware Isometry, which differs from our focus on stabilizing the output-input norm ratio. For the backward pass, they rely on gradient-guided dynamic rewiring, modifying the model architecture itself. In contrast, we control backward signal propagation through the BSP metric. Moreover, their method design does not address over-smoothing, while we explicitly incorporate the Graph Embedding Variation (GEV) metric to do so.
>
> **Han et al. [3]** introduce MLPInit, which is not aimed at improving signal propagation or reducing over-smoothing. Instead, it seeks to accelerate training by initializing GNNs with the weights of a fully trained, equivalent MLP. This is a fundamentally different motivation from ours.
>
> We have incorporated these comparisons and discussions into Section 6 (Related Work) of the revised manuscript.
>
> **References:**
>
> [1] Kelesis et al., Reducing Oversmoothing through Informed Weight Initialization in Graph Neural Networks, arXiv:2410.23830 (2024)
>
> [2] Jaiswal et al., Old can be gold: Better gradient flow can make vanilla-GCNs great again, NeurIPS 2022
>
> [3] Han et al., MLPInit: Embarrassingly Simple GNN Training Acceleration with MLP Initialization, arXiv:2210.00102 (2022)

---

> ### Author Response · Authors · 2025-04-11
> **Part 6: Comparisons Across Activation Functions and Additional Models**
>
> **Comment:** *Comparison of SPoGInit with other methods that use different activation functions rises concerns about the source of the observed improvements. Are these improvements attributed to the proposed method or to the use of different activation function?*
>
> **Response:**
>
> We respectfully clarify a potential misunderstanding. In our experiments, for a fixed GNN architecture, **SPoGInit and all compared methods use the same activation function**. As stated in Section 5.1 (Experimental Settings):
>
> > "In our experiments, we use the tanh activation function for vanilla GCNs, as Theorem 3.2 shows that the graph variation embedding of vanilla GCNs with ReLU activation does not benefit from further optimization. For other GCN architectures, we use the ReLU activation function."
>
> Therefore, the observed performance improvements are not attributed to differences in activation functions but rather to the effectiveness of the proposed SPoGInit.

---

> ### Author Response · Authors · 2025-04-11
> **Part 7: Reporting the Learned Per-Layer Scaling Factor $\sigma_{w,i}$**
>
> **Comment:** *Including the resulting values of $\sigma_i$ for each layer would shed more light on what values does the proposed method learn and if and how they differ from other initialization methods.*
>
> **Response:**
>
> Thank you for the valuable suggestion. We have plotted the values of $\sigma_{w,i}$ for each layer of our SPoGInit, as well as for several baseline initialization methods, including Conventional initialization, Xavier, VirgoFor, and VirgoBack. As shown in Figure 6 in Appendix E.4, $\sigma_{w,i}$ differs significantly across different initialization methods. Notably, SPoGInit performs an adaptive variance search, which leads to a fluctuating pattern of $\sigma_{w,i}$ across layers.

---

> > ### Comment · Reviewer_u83j · 2025-04-13
> > **Thank you for the detailed response**
> >
> > I appreciate the authors’ responses to the concerns raised and the corresponding revisions made to improve the manuscript. It is clear that significant effort was put into addressing the feedback, and I find the clarifications and updates to be constructive and helpful.
> >
> > One point I would like to revisit briefly is the behavior of the BSP term. While I understand that an excessive emphasis on this term may destabilize optimization, I would be curious if the authors have any further intuition or insights into why this happens, even if a detailed investigation is beyond the scope of the current work.
> >
> > Regarding the omission of graph classification tasks, I acknowledge the authors’ rationale. That said, incorporating such tasks, perhaps even in a limited scope, could help demonstrate the generality of the proposed method across different GNN applications and task types (even though I do not consider this as mandatory).
> >
> > Lastly, in the related work section where the manuscript discusses "other architectural modifications", I suggest including [1] and [2], as they are relevant and may help situate the proposed method more clearly within the existing literature.
> >
> > Overall, I believe the manuscript has improved, and I appreciate the careful attention the authors have given to all points raised.
> >
> > [1] Zhang et al., ‘Graph-adaptive Rectified Linear Unit for Graph Neural Networks', WWW 2022
> >
> > [2] Kelesis et al., 'Reducing Oversmoothing in Graph Neural Networks by Changing the Activation Function', ECAI 2023

---

> > > ### Author Response · Authors · 2025-04-16
> > >
> > > We sincerely thank the reviewer for the positive feedback and thoughtful suggestions. We address the three remaining points below:
> > >
> > > **1. On the effect of over-emphasizing the BSP term**
> > >
> > > We appreciate your insightful question regarding the behavior of the BSP term. We also find this phenomenon puzzling and thus increase the number of experimental runs from 20 to 50 to mitigate the effect of randomness. The updated results (see Table 1 below) show that when the weight coefficient $w_2$ is increased to 100, the value of the BSP term becomes slightly lower than that under $w_2 = 10$. This suggests that the previously observed increase might have been due to stochastic fluctuations.
> > >
> > > *Table 1: The $\hat{V_{FSP}}$ and $\hat{V}_{BSP}$ of 32-layer tanh-activated GCN using SPoGInit with different $w_2$ on Cora dataset. In SPoGInit, we set the learning rate as 0.1, total iteration as 100, $w_1$ and $w_3$ as 1. We report the average result over 50 runs.*
> > >
> > > |                           | **w/o SPoG**         | **SPoG using $w_2=1$** | **SPoG using $w_2=10$** | **SPoG using $w_2=100$** |
> > > |---------------------------|----------------------|-------------------------|--------------------------|---------------------------|
> > > | $\hat{V}_{FSP}$           | 31.2                 | $5.4 \times 10^{-3}$    | $6.4 \times 10^{-3}$     | $7.5 \times 10^{-2}$      |
> > > | $\hat{V}_{BSP}$           | $1.3 \times 10^{-1}$ | $3.0 \times 10^{-3}$    | $1.1 \times 10^{-3}$     | $1.0 \times 10^{-3}$      |
> > >
> > >
> > >
> > > Importantly, the updated results still support the conclusion that $w_2 = 10$ achieves a favorable trade-off between forward and backward signal preservation, as it yields low values for both $\hat{V_{FSP}}$ and $\hat{V}_{BSP}$. We have revised the corresponding experimental description and analysis in Appendix E.2.
> > >
> > > **2. On generalizing SPoGInit to other GNN tasks**
> > >
> > > Thank you for highlighting the importance of evaluating the generality of SPoGInit across different GNN tasks. While our current work focuses on node classification, we believe the proposed SP framework has the potential to be extended to other settings such as edge prediction and graph classification.
> > >
> > > To be more specific, SPoGInit might be adapted to these tasks by modifying the SP metrics accordingly. For example:
> > > - In edge prediction, the graph embedding variation (GEV) metric could be redefined to capture differences in edge-level representations.
> > > - For graph classification, one could aggregate node-level GEV statistics or design a variant of the metric that reflects variations in the graph-level embedding across layers.
> > >
> > > We believe these are promising directions for follow-up research and plan to explore them in future work.
> > >
> > > **3. More related works on other architectural modifications**
> > > We thank the reviewer for pointing out these relevant works. We have included both references—Zhang et al. (WWW 2022) and Kelesis et al. (ECAI 2023)—into the Related Work section (Section 6).

---

### Author Response · Authors · 2025-06-08
**Camera-ready Version Submitted**

We sincerely thank the editor and reviewers for their insightful feedback, which has significantly improved our paper. We have carefully addressed all comments and incorporated additional experiments and analyses as suggested.

---

### Decision · Action_Editor_fpxk · 2025-05-02

**Recommendation:** Accept with minor revision

**Comment:**

Three experts reviewed the paper. The authors responded appropriately to the three review comments' suggestions, questions, and concerns and revised the paper. The reviewers agreed to accept the paper unanimously.

As stated above, the paper meets the acceptance criteria in terms of both *Claim and Evidence* and *Audience*. Therefore, I am in favor of Accepting the paper.

I suggest fixing one point: In P.35 L.1, part 2 seems to be a typo of part 1.

**Audience:**

GNN is one of the main areas of deep learning research, and GNN performance degradation by stacking many layers is one of the main topics that has been studied for a long time and is still under active research. This paper proposes a new initialization method for GNNs in terms of signal propagation to solve this problem.

Therefore, this paper is within the scope of interest of TMLR readers and meets the Audience criteria.

**Claims And Evidence:**

This paper analyzes the performance degradation of GNNs caused by deepening, including over-smoothing, from the perspective of signal propagation. This paper claims two things:
- First, this paper introduces three signal processing metrics (forward signal propagation, backward signal propagation, and graph embedding variation). It shows that existing GCNs and Residual GCNs cannot simultaneously control these three metrics.
- Second, this paper introduces SPoGInit, which initializes the model parameters by optimizing these metrics, and demonstrates its effectiveness.

Both claims are supported by appropriate evidence and satisfy the criteria for claims and evidence:
- For the first claim, this paper shows that these metrics are either zero or divergent under infinite width, assuming that the activation function of Residual GCN is linear and using the Neural Network Gaussian process approximation. Also, numerical experiments demonstrate that as the number of layers increases, one of the metrics either vanishes or explodes.
- For the second claim, this paper verifies SPoGInit's effectiveness by applying four models and four baseline initialization methods to node classification tasks and long-range dependence tasks on four types of datasets. In addition, in response to the reviewer's comments, the authors conducted additional experiments that change models, datasets, baselines, and ablation studies.